# Challenges in developing a global gradient-based groundwater model (G³M v1.0) for the integration into a global hydrological model

Robert Reinecke[1], Laura Foglia[3], Steffen Mehl[4], Tim Trautmann[1], Denise Cáceres[1], Petra Döll[1,2]

[1]Institute of Physical Geography, Goethe University Frankfurt, Frankfurt am Main, Germany
[2]Senckenberg Leibniz Biodiversity and Climate Research Centre Frankfurt (SBiK-F), Frankfurt am Main, Germany
[3]Department of Land, Air and Water Resources, University of California, Davis, USA
[4]Department of Civil Engineering, California State University, Chico, USA

*Correspondence to*: Robert Reinecke (reinecke@em.uni-frankfurt.de)

**Abstract.** In global hydrological models, groundwater (GW) is typically represented by a bucket-like linear groundwater
reservoir. Reservoir models, however, can (1) only simulate GW discharge to surface water (SW) bodies but not recharge from
SW to GW, (2) provide no information on the location of the GW table and (3) assume that there is no GW flow among grid
cells. This may lead, for example, to an underestimation of groundwater resources in semi-arid areas where GW is often
replenished by SW or to an underestimation of evapotranspiration where the GW table is close to the land surface. To overcome
these limitations, it is necessary to replace the reservoir model in global hydrological models with a hydraulic head gradient-
based GW flow model.

We present G³M, a new global gradient-based GW model with a spatial resolution of 5', which is to be integrated in
the 0.5° WaterGAP Global Hydrology Model (WGHM). The newly developed model framework enables in-memory coupling
to WGHM while keeping overall runtime relatively low, which allows sensitivity analyses, calibration and data assimilation.
This paper presents the G³M concept and model design decisions that are specific to the large grid size required for a global
scale model. Model results under steady-state naturalized conditions, i.e. neglecting GW abstractions, are shown. Simulated
hydraulic heads show better agreement to observations around the world than model output of de Graaf et al. (2015). Locations
of simulated SW recharge to GW are found, as is expected, in dry and mountainous regions but areal extent of SW recharge
may be underestimated. Globally, GW discharge to rivers is by far the dominant flow component such that lateral GW flows
only become a large fraction of total diffuse and focused recharge in case of losing rivers, some mountainous areas and some
areas with very low GW recharge. A strong sensitivity of simulated hydraulic heads to the spatial resolution of the model and
the related choice of the water table elevation of surface water bodies was found. We suggest to investigate how global-scale
groundwater modelling at 5' spatial resolution can benefit from more highly resolved land surface elevation data.

## 1 Introduction

Groundwater (GW) is the source of about 40% of all human water abstractions (Döll et al., 2014) and is also an essential source
of water for freshwater biota in rivers, lakes and wetlands. GW strongly affects river flow regimes and supplies the majority
of river water during ecologically and economically critical periods with little precipitation. GW storage and flow dynamics
have been altered by human GW abstractions as well as climate change and will continue to change in the future (Taylor et
al., 2012). Around the globe, GW abstractions have led to lowered water tables and, in some regions, even GW depletion (Döll
et al., 2014; Scanlon et al., 2012; Wada et al., 2012; Konikow, 2011). This has resulted in reduced base flows to rivers and
wetlands (with negative impacts on water quality and freshwater ecosystems), land subsidence and increased pumping costs
(Wada, 2016; Döll et al., 2014; Gleeson et al., 2012; 2016). The strategic importance of GW for global water and food security
will probably intensify under climate change as more frequent and intense climate extremes increase variability of SW flows
(Taylor et al., 2012). International efforts have been made to promote sustainable GW management and knowledge exchange
among countries, e.g., UNESCO's program on International Shared Aquifer Resources Management (ISARM)

(http://isarm.org) and the ongoing GW component of the Transboundary Waters Assessment Program (TWAP) (http://www.geftwap.org). To support prioritization for investment among transboundary aquifers as well as identification of strategies for sustainable GW management, information on current conditions and possible trends of the GW systems is required (UNESCO-IHP, IGRAC, WWAP, 2012). In a globalized world, an improved understanding of GW systems and their interaction with SW and soil is needed not only at the local and regional but also at the global scale.

To assess GW at the global scale, global hydrological models (GHMs) are used e.g. (Wada et al., 2012; 2016; Döll et al., 2012; 2014). In particular, they serve to quantify GW recharge (Döll and Fiedler, 2008). Like typical hydrological models at any scale, GHMs simulate GW dynamics by a linear reservoir model. In such a model, the temporal change of GW storage in each grid cell is computed from the balance of prescribed inflows and an outflow that is a linear function of GW storage. Linear reservoir models can only simulate GW discharge to SW bodies but not a reversal of this flow, even though losing streams may provide focused GW recharge that allows the aquifer to support ecosystems alongside the GW flow path (Stonestrom et al., 2007) as well as human GW abstractions. Losing streams typically occur in semi-arid and arid but seasonally also in humid regions. In addition, such linear reservoir models provide no information on the location of the GW table, and assume that GW flow among grid cells is negligible. To simulate the dynamics of water flow between SW bodies and GW in both directions as well as the effect of capillary rise on evapotranspiration, it is necessary to compute lateral GW flows among grid cells as function of hydraulic head gradients and thus the dynamic location of the GW table. To achieve an improved understanding of GW systems at the global scale, and in particular of the interactions of GW with SW and soil, it is therefore necessary to replace the linear GW reservoir model in GHMs by a hydraulic gradient-based GW flow model.

Large-scale gradient-based GW flow models are still rare and mainly available for data-rich regions, e.g. for the Death Valley (Belcher and Sweetkind, 2010) and the Central Valley (Belcher and Sweetkind, 2010; Faunt, 2009; Dogrul et al., 2016) in the USA, but also for large fossil groundwater bodies in arid regions (e.g. the Nubian Aquifer System in North Africa, (Gossel et al., 2004)). However, they are in most cases not integrated within hydrological models that quantify GW recharge based on climate data and provide information on the condition of SW (e.g. streamflow and storage). For North America, Fan et al. (2007) and Miguez-Macho et al. (2007) linked a land surface model with a two-dimensional gradient-based GW model and computed, with a daily time step, GW flow, water table elevation, GW–SW interaction, and capillary rise, using a spatial resolution of 1.25 km. One challenge was the determination of the river conductance that affects the degree of GW–SW interaction. A computationally very expensive integrated simulation of dynamic SW, soil and GW flow using Richards' equation for variably saturated flow was achieved at a spatial resolution of 1 km for the continental US by applying the ParFlow model (Maxwell et al., 2015). In both studies, GW abstractions were not taken into account.

A first simulation of the steady-state GW table for the whole globe at the very high resolution of 30" was presented by Fan et al. (2013) and compared to an extensive compilation of observed hydraulic heads. However, there was no head-based interactions with SW; GW above the land surface was simply discarded. Global GW flow modeling is strongly hampered by data availability, including the geometry of aquifers and aquitards as well as parameters like hydraulic conductivity (de Graaf et al., 2017), and by computational restrictions on spatial resolution leading to conceptual problems, e.g., regarding SW-GW interactions (Morel-Seytoux et al., 2017). Recently, some GW flow models that are in principle applicable for the global scale were developed but were applied only regionally in data-rich regions (Rhine basin: Sutanudjaja et al., 2011; France: Vergnes et al., 2012; 2014). The first global gradient-based GW model that was run for both steady-state (de Graaf et al., 2015) and transient conditions (de Graaf et al., 2017) was driven by GW recharge and SW data of the GHM PCR-GLOBWB (van Beek et al., 2011). However, to achieve plausible discharge performance, they found it necessary to increase drainage from GW to rivers beyond the drainage driven by the hydraulic head difference between GW and river. This additional drainage, which accounts for about 50% of global GW flow into SW, is simulated as a function of GW storage above the floodplain.

In this study, we present the Global Gradient-based Groundwater Model (G$^3$M) that is to be integrated into the GHM WaterGAP 2 to improve estimation of flows between SW and GW (affecting both streamflow and groundwater recharge and thus water availability for humans and ecosystems) and implement capillary rise (affecting evapotranspiration). Table 1

provides a comparative summary of G³M as well as the global groundwater models of Fan et al. (2013), de Graaf et al. (2015; 2017), and the continental scale model ParFlow (Maxwell et al., 2015).

The objective of this paper is to learn from a steady-state model, a well-established first step in groundwater model development, to (1) understand the basic model behaviour by limiting model complexity and degrees of freedom, and thus (2) providing insights into dominant processes and uncovering potential model-inherent characteristics difficult to observe in a fully coupled transient model. A transient model might obfuscate model inherent trends due to the slow changing nature of groundwater processes e.g. trends towards large over/under-estimation due to wrong parameterisation. A fully coupled model furthermore adds complexity and uncertainty to the model outcome. The presented steady-state model is furthermore used to (3) investigate parameter sensitivity and sensitivity to spatial resolution. In addition, the steady-state solution can be used as (4) initial condition for future fully coupled transient runs.

Model concept and equations as well as applied data and parameter values are presented in section 2. In section 3, we show steady-state results of G³M driven by WGHM data. Simulated hydraulic heads are compared to observations world-wide and to the output of existing large-scale GW models (Table 1). Furthermore, sensitivity to parameters and grid size is shown for the example of New Zealand. Finally, the implications of modeling decisions and grid size are discussed (section 4) and conclusions are drawn (section 5).

## 2 Model description

### 2.1 G³M model concept

Although G³M is based on principles of the well-known GW flow modelling software MODFLOW (Harbaugh, 2005), G³M differs in its parameterization from traditional local and regional GW models. These models are generally based on rather detailed information on hydrogeology (including aquifer geometry and properties such as hydraulic conductivity derived from pumping tests), topography, pumping wells, location and shape of SW bodies as well as on observations of hydraulic head in GW and SW. Local observations guide the developer in constructing the model such that local conditions and processes can be properly represented. The lateral extent of individual grid cells of such GW flow models is generally smaller or similar to the depth of the aquifer(s) and the size of the SW bodies that interact with the GW. The global GW flow model G³M, however, covers all continents of the Earth except Greenland and Antarctica. At this scale, information listed above is poor or non-existing, and the lateral extent of grid cells needs to be relatively large due to computational (and data) constraints. We selected a grid cell size of 5' by 5' (approx. 9 km by 9 km at the equator), as this size fits well to WaterGAP and is smaller than the suggested 6' of Krakauer et al. (2014). WaterGAP 3 (Eisner, 2016) has the same cell size, and 36 of such cells fit to into one 0.5° WaterGAP 2 cell. Global climate data are only available for 0.5° grid cells. The landmask of G³M, i.e. location and size of 5' grid cells, is that of WaterGAP 3 end encompasses 2.2 million 5' grid cells on each layer.

Due to the lack of the spatial distribution of hydrogeological properties, we chose to use, in the current version of G³M, two GW layers with a vertical size of 100 m each (Fig. 1). We performed a sensitivity analysis that confirmed the findings of others (de Graaf et al., 2015) that the aquifer thickness has a relatively small impact on the model results. Therefore, selecting a uniform thickness of 100 m (motivated by the assumed depth of validity of the lithology data) (Fig. 1) worldwide for the first layer and also for the second layer is expected to lead to less uncertainties as compared to hydraulic conductivities and the surface water table elevation.

G³M focuses on a plausible simulation of water flows between GW and SW, and we deemed it suitable to have an upper GW layer that interacts with SW and soil (the soil layer of WaterGAP is described in detail in S1) and a lower one in which GW may flow laterally without such interactions. As land surface elevation within each 5' grid cell, with an area of approximately 80 km², may vary by more than 200 m (Fig. S4.1), neighbouring cells in G³M may not be adjacent anymore (Fig. 1), in contrast to (regional) GW models with smaller grid cells. This makes G³M a rather conceptual model in which

water exchange between groundwater cells is driven by hydraulic head gradients but flow can no longer be conceptualized as occuring through continuous pore space. In addition, due to the coarse spatial scale and the possible large variations of land surface elevations within each grid cell, the upper model layers should not be considered to be aligned with an average land surface elevation. The model layers can be rather thought to be vertically aligned with the elevation of the surface water body

table, as this prescribed elevation is, together with the sea level, the only elevation included in the groundwater flow equation (Eq. (1)).

The simulation of aquifers that contain dry cells and/or cells that oscillate between wet and dry states poses great challenges to solving Eq. (1) (Niswonger et al., 2011). G³M-f (the framework code used to implement G³M) implements the traditional wetting approach from Harbaugh (2005) as well as the approach proposed by Niswonger et al. (2011) along with

the proposed damping scheme. Both approaches have proven to be insufficient to simulate head-based transmissivities (unconfined contions) on the global scale. Large mountainous areas would be excluded if unconfined conditions are assumed from the beginning of the solution step, as the head is often far below the deepest model layer, resulting in a no-flow condition and imposing convergence issues to the matrix solver. We choose to simulate both layers with a specific saturated thickness even though the upper layer can be expected to decrease in water level and thus in transmissivity (hydraulic conductivity times

saturated depth). The large uncertainties regarding hydraulic conductivities (possibly an order of magnitude), further justifies using the computationally more efficient assumption of specified saturated thickness. This approach is consistent with findings that this is accurate for large, complex groundwater models (Sheets et al., 2015). Furthermore, it is consistent with recent presented large scale studies e.g. for the Rhine Meuse basin of Sutanudjaja et al. (2011) (using one confined layer), the Death Valley Regional Flow Model (Belcher, 2004; Faunt et al., 2011), and the global groundwater model of de Graaf et al. (2017)

(two layers and partially unconfined conditions are simulated by parametrization of the model input and not by a head-dependant transmissivity).

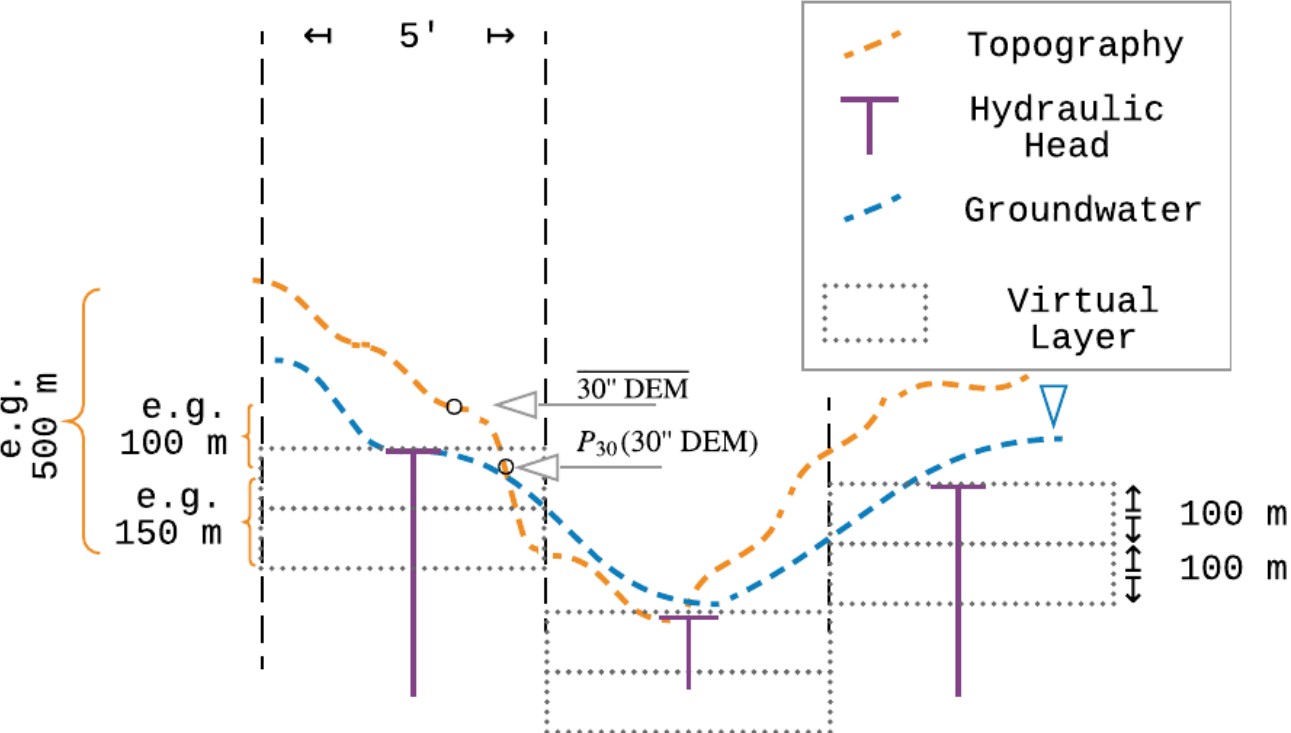

**Figure 1** Schematic of G³M's spatial structure, with 5' grid cells, hydraulic head per cell, and the conceptual virtual layers (virtual because
at this stage only confined conditions are computed). The underlying variability of the topography changes the perception of simulated depth to groundwater depending on what metrics are used to represent it on a coarser resolution. Layers in G³M are of a conceptual nature and describe the saturated flow between locations of head laterally and vertically. The $P_{30}$ is used in the presented steady-state model as SW elevation instead of an average or minimum per grid cell.

Three-dimensional groundwater flow is described by a partial differential equation (approximated in the model implementation by using the finite differences method (Sect. 2.4))

$$\frac{dGWS}{dt} = \left(\frac{\partial}{\partial x}\left(K_x \frac{\partial h}{\partial x}\right) + \frac{\partial}{\partial y}\left(K_y \frac{\partial h}{\partial y}\right) + \frac{\partial}{\partial z}\left(K_z \frac{\partial h}{\partial z}\right) + W\right) \Delta x \Delta y \Delta z = S_s \frac{\partial h}{\partial t} \Delta x \Delta y \Delta z \tag{1}$$

where $K_{x,y,z}$ is the hydraulic conductivity $[LT^{-1}]$ along the x, y, and z axis between the cells (harmonic mean of grid cell conductivity values), $S_s$ the specific storage $[L^{-1}]$, $\Delta x \Delta y \Delta z$ $[L^3]$ the volume of the cell, and $h$ the hydraulic head $[L]$. In- and out-flows in the groundwater are accounted for as

$$W \, \Delta x \Delta y \Delta z = R_g + Q_{swb} - NA_g - Q_{cr} + Q_{ocean} \tag{2}$$

where $Q_{swb}$ is flow between the SW bodies (rivers, lakes, reservoirs and wetlands) and GW $[L^3T^{-1}]$, $Q_{cr}$ is capillary rise, i.e. the flow from GW to the soil, and $Q_{ocean}$ is the flow between ocean and GW $[L^3T^{-1}]$, representing the boundary condition. In case of $Q_{swb}$ and $Q_{ocean}$, a positive value represents a flow into the groundwater.

$Q_{swb}$ in Eq. (3) replaces $k_g \, GWS$ and $R_{g\_swb}$ in the linear storage equation of WaterGAP (Eq. (S1)), such that losing conditions of all types of SW bodies can be simulated dynamically. It is calculated as a function of the difference between the elevation of the water table in the SW bodies $h_{swb}$ $[L]$ and $h_{aq}$ as

$$Q_{swb} = \begin{cases} c_{swb}(h_{swb} - h_{aq}) & h_{aq} > B_{swb} \\ c_{swb}(h_{swb} - B_{swb}) & h_{aq} \le B_{swb} \end{cases} \tag{3}$$

where $c_{swb}$ is the conductance $[L^2T^{-1}]$ of the SW body bed (river, lake, reservoir or wetland) and $B_{swb}$ the SW body bottom elevation $[L]$.

Conductance of SW bodies is often a calibration parameter in traditional GW models (Morel-Seytoux et al., 2017). Following Harbaugh (2005), it can be estimated by

$$c_{swb} = \frac{K \, L \, W}{h_{swb} - B_{swb}} \tag{4}$$

where $K$ is hydraulic conductivity, $L$ is length and $W$ is width of the SW body per grid cell. For lakes (including reservoirs) and wetlands, $c_{swb}$ is estimated based on hydraulic conductivity of the aquifer $K_{aq}$ and SW body area (Table 2). For gaining rivers, conductance is quantified individually for each grid cell following an approach proposed by Miguez-Macho et al. (2007). The value of river conductance $c_{riv}$, according to Miguez-Macho et al. (2007), in a GW flow model needs to be set to such a values that, for steady-state conditions, the river is the sink for all the inflow to the grid cell (GW recharge and inflow from neighbouring cells) that is not transported laterally to neighbouring cells such that

$$c_{riv} = \frac{R_g + Q_{eq_{lateral}}}{h_{eq} - h_{riv}} \qquad h_{aq} > h_{riv} \tag{5}$$

For G³M, we computed the equilibrium head $h_{eq}$ as the 5' average of the 30" steady-state heads calculated by Fan et al. (2013). Using WGHM diffuse GW recharge lateral equilibrium flow $Q_{eq_{lateral}}$ $[L^3T^{-1}]$ is net lateral inflow into the cell computed based on the $h_{eq}$ distribution as well as G³M $K_{aq}$ and cell thickness (Table 2). Elevation of the river water table $h_{riv}$ [L] is to be provided by WGHM. Using a fully dynamic approach, i.e. utilizing the hydraulic head and lateral flows from the current iteration to re-calculate $c_{riv}$ in each iteration towards the steady-state solution, has proven to be too unstable due to its non-linearity affecting convergence. We limit $c_{riv}$ to a maximum of $10^7 \, m^2 day^{-1}$; this would be approximately the value for a 10 km long and 1 km wide river with a head difference between GW and river of 1 m and hydraulic conductivity of the river bed of $10^{-5}$ m/s.

If the river recharges the GW (losing river), Eq. (5) cannot be used as the Fan et al. (2013) high-resolution equilibrium model only models groundwater outflows but not inflows from SW bodies. If $h_{aq}$ drops below $h_{riv}$, Eq. (4) is used to compute $c_{riv}$, with $K$ equals to $K_{aq}$.

The flux across the model domain boundary $Q_{ocean}$ is modeled as a head-dependent flow based on a static head boundary.

$$Q_{ocean} = c_{ocean}(h_{ocean} - h_{aq}) \tag{6}$$

where $h_{ocean}$ is the elevation of the ocean water table $[L]$, $h_{aq}$ the hydraulic head in the aquifer $[L]$ and $c_{ocean}$ the conductance of the boundary condition $[L^2 T^{-1}]$ (Table 2). We assume that density difference to sea-water is negligible at this scale. $Q_{cr}$ is not yet implemented in G³M.

## 2.2 The steady-state uncoupled model version

In a first implementation stage, G³M was developed as a steady-state (right-hand side of Eq. (1) is zero) standalone model that represents naturalized conditions (i.e. without taking into account human water use) during 1901-2013. Input data and parameters used are listed in Table 2 and described below.

Gleeson et al. (2014) provided a global subsurface permeability data set from which $K_{aq}$ was computed. The data set was derived by relating permeabilities from a large number of local to regional GW models to the type of hydrolithological units (e.g., "unconsolidated" or "crystalline"). The geometric mean permeability values of nine hydrolithological units were mapped to the high-resolution global lithology map GLiM (Hartmann and Moosdorf, 2012). In continuous permafrost areas, a very low permeability value was assumed by Gleeson et al. (2014). The estimated values represent the shallow surface on the scale of 100 m depth. The unique dataset has three inherent problems when used as input for a GW model: (1) At this scale, important heterogeneities such as discrete fractures or connected zones of high hydraulic conductivity controlling the GW flow are not visible. (2) Jurisdictional boundaries due to different data sources in the global lithological map lead to artifacts. (3) The differentiation between coarse and fine-grained unconsolidated deposits is only available in some regions resulting in $10^{-4}\ m\ s^{-1}$ as hydraulic conductivity for coarse-grained unconsolidated deposits. If the distinction is not available, a rather low value of $10^{-6}\ m\ s^{-1}$ is set for unconsolidated porous media (Fig. S4.3). The original data was gridded to 5' by using an area-weighted average and used as hydraulic conductivity of the upper model layer. For the second layer, hydraulic conductivity of the first layer is reduced assuming that conductivity decreases exponentially with depth. Based on the e-folding factor $f$ used by Fan et al. (2013) (a calibrated parameter based on terrain slope), conductivity of the lower layer is calculated by multiplying the upper layer value by $\exp(-50\ m\ f^{-1})^{-1}$ (Fan et al., 2007).

Mean annual GW recharge computed by WaterGAP 2.2c for the period 1901-2013 is used as input (Fig. S4.4), while no net abstraction from GW was taken into account. It would not be meaningful to try to derive a steady-state solution under existing net groundwater abstractions that in some regions cause GW depletion with continuously dropping water tables. The 0.5° data of WaterGAP was equally distributed to the pertaining cells. Regarding the ocean boundary condition, $h_{ocean}$ is set to 0 m and $c_{ocean}$ to $10\ m^2\ day^{-1}$ (Table 2), reflecting a global average conductance based on hydraulic conductivity and lateral surface area.

It is assumed that there is exchange of water between GW and one river stretch in each 5' grid cell, and in addition where lakes and wetlands exist according to WaterGAP 3, which provides, for each grid cell, the area of "local" and "global" lakes and wetlands. In WaterGAP, "local" SW bodies are only recharged by runoff produced within the grid cell, while "global" SW bodies also obtain inflow from the upstream cell. In an uncoupled model, it is difficult to prescribe the, in reality temporal variable, area of lakes and wetlands that affect the flow exchange between SW body and GW. Maps generally show the maximum spatial extent of SW bodies. This maximum extent is seldom reached, in particular in case of wetlands in dry areas. For global wetlands (wetlands greater than one 5' cell), it is therefore assumed in this model version that only 80% of their maximum extent is reached. In the transient model SW body areas change over time. A further difficulty in an uncoupled model run is that the water table elevation of SW bodies does not react to the GW-SW exchange flows $Q_{swb}$ and that water supply from SW is not limited by availability. A losing river may in reality dry out and therefore cease to lose any more water. For rivers $B_{swb}$ is set to $h_{riv} - 0.349 \times Q_{bankfull}^{0.341}$ (Allen et al., 1994), where $Q_{bankfull}$ is the bankfull river discharge in

the 5' grid cell (Verzano et al., 2012). Globally constant values are used for $B_{swb}$ for wetlands, local lakes and global lakes (Table 2).

For the steady-state model, all SW bodies in a grid cell are assumed to have the same head i.e $h_{riv} = h_{swb}$. We found that for both gaining and losing conditions, $Q_{swb}$ and thus computed hydraulic heads are highly sensitive to $h_{swb}$. The overall
best agreement with the hydraulic head observations of Fan et al. (2013) was achieved if $h_{swb}$ (Eq. (3), (4) and (5)) was set to the $30^{th}$ percentile ($P_{30}$) of the 30" land surface elevation values of Fan et al. (2013) per 5' cell, e.g. the 30" elevation that is exceeded by 70% of the 100 30" elevation values within one 5' cell. To decrease convergence time we used $h_{eq}$ derived from the high-resolution steady-state hydraulic head distribution of Fan et al. (2013) as initial guess of $h_{aq}$. In each outer iteration (Sect. 2.4) gaining and losing conditions may change depending on the current head solution.

**2.3 Integration into WGHM**

We intend to integrate G³M into WaterGAP 2, i.e. the 0.5° version of WATERGAP (for details see S1), to keep computation time low enough for performing sensitivity analyses and ensemble-based data assimilation and calibration, instead of integrating it into WaterGAP 3 (Eisner, 2016), which has the same spatial resolution as G³M. However, data from WaterGAP 3 were used to set up G³M. Location and area of the 5' grid cells of G³M are the same as in the landmask of WaterGAP 3. In
addition, the percentage of the 5' grid cell area that is covered by lakes (including reservoirs) and by wetlands, based on Lehner and Döll (2004), is taken from WaterGAP 3, as well as the length and width of the main river within each 5' grid cell as (Table 2).

**2.4 Model implementation**

G³M is implemented using a newly developed open-source model framework G³M-f (Reinecke, 2018). The main motivation
to develop a new model framework is the efficient in-memory coupling to the GHM and flexible adaptation to the specific requirements of global-scale modelling. Written in C++14, the framework allows the implementation of global and regional groundwater models alike while providing an extensible purely object-oriented model environment. It is primarily targeted as extension to WaterGAP but allows an in-memory coupling to any GHM or can be used as a standalone groundwater model. It provides a unit-tested (Dustin, 2006) environment offering different modules that can couple results in-memory to a different
model or write out data flows to different file formats. G³M-f has the following advantages over using an established GW modelling software such as MODFLOW. G³M-f enables an improved coupling capability. Unlike MODFLOW it provides a clear development interface to the programmer coupling a model to G³M-f. It can be easily compiled as a library, and provides a clearly separated logic between computation and data read-in/write-out). It is written in the same language as the target GHM enabling a straight-forward in-memory access to arrays without the need to write data to disk, required when coupling with
MODFLOW (a very expensive operation even if that disk is a RAM-disk). Even though it is possible to call FORTRAN functions from C++, it is very complicated to pass file pointers properly, as the I/O implementation of both languages differ substantially and it is widely considered bad practice to handle I/O in two different languages at once. As MODFLOW was never designed to be coupled/integrated to/into other models, it is not possible to separate the I/O logic fully from the computational logic without substantial code changes that are hard to test. To this end, G³M-f provides a highly modularized
framework that is written with extensibility as design goal while implementing all required groundwater mechanisms.

Eq. (1) is reformulated as finite-difference equation and solved using a conjugate gradient approach and an Incomplete LUT preconditioner (Saad, 1994). In order to keep the memory footprint low, the conjugate gradient method makes use of the sparse matrix. Furthermore, it solves the equations in parallel (preconditioner currently non-parallel). As internal numerical library G³M uses Eigen3 (eigen.tuxfamily.org). G³M can compute the presented steady-state solution (with the right-hand side of Eq. (1) being zero and the heads of Fan et al. (2013) as initial guess (Table 1,2)) on a commodity computer with four
computational cores and a standard SSD in about 30 minutes while occupying 6 GB of RAM.

Similar to MODFLOW, G³M-f solves Eq. (1) in two nested loops using a Picard iteration (Mehl, 2006): (1) the outer iteration checks the head and residual convergence criterion (if the maximum head change between iterations is below a given value in three consecutive iterations and/or the norm of the residual vector of the conjugate gradient (Harbaugh, 2005; Niswonger et al., 2011) is below a given value). It adjusts head-dependant values e.g. from gaining to losing conditions and updates the system of linear equations if flows are no longer head dependent. (2) The inner loop primarily consists of the conjugate gradient solver, which runs for a number of iterations defined by the user or until the residual convergence criterion is reached (Table 2), solving the current system of linear equations.

Because switching between Eq. (4) and Eq. (5), which occurs if e.g. $h_{aq}$ drops below $h_{riv}$ from one iteration to the next causes an abrupt change of $c_{riv}$ inducing a nonlinearity that affects convergence we introduced an $\epsilon = 1$ m interval around $h_{riv}$ and interpolate $c_{riv}$ between the two Eq. (4) and (5) by a cubic hermite spline polynomial over that interval. This allows for a smoother transition between both states, reducing the changes in the solution if a river is in a gaining condition in one iteration and in a losing condition in the next or vice versa.

Different from Vergnes et al. (2014), G³M's computations are not based on spherical coordinates directly but on an irregular grid of quadratic cells of different size depending on the latitude. Cell sizes are provided by WaterGAP3 and are derived from their spherical coordinates maintaining their correct area and centre location. The model code will be adapted in the future to account for the different length in x and y direction per cell correctly.

## 3 Results

### 3.1 Global hydraulic head and water table depth distribution under natural steady-state conditions

As expected, the computed global distribution of steady-state hydraulic head (in the upper model layer) under natural conditions (Fig. 2a) follows largely the land surface elevation (Fig. S4.2), albeit with a lower range and locally different ratios between the hydraulic head and land surface gradients (Fig. S4.6). Water table depth (WTD), i.e. the distance between the groundwater table and the land surface, can be computed by subtracting the hydraulic head computed by G³M for the upper layer of each 5' grid cell from the arithmetic mean of the land surface elevations of the 100 30" grid cells within each 5' cells (Fig. S4.2). The global map of steady-state WTD (Fig. 2b) clearly resembles the map of differences between surface elevation and $P_{30}$ , the assumed water level of SW bodies $h_{swb}$, shown in Fig. S4.1**,** which indicates that simulated WTD is strongly governed by the assumed water level in SW bodies.

Deep GW, i.e. a large WTD, occurs mainly in mountainous regions (Fig. 2b). These high values of WTD are mainly a reflection of the steep relief in these areas as quantified either by the differences of mean land surface elevations between neighbouring grid cells (Fig. S4.7) or the difference between mean land surface elevation and $P_{30}$, the 30[th] percentile of the 30" land surface elevations (Fig. S4.1). When computed hydraulic head is subtracted not from average land surface elevation but from $P_{30}$, the assumed water table elevation of SW bodies, the resulting map shows that the groundwater table is mostly above $P_{30}$, in both flat and steep terrain (Fig. 2c). Thus, high WTD values at the 5' resolution do not indicate deep unsaturated zones and losing rivers but just high land surface elevation variations within a grid cell. In steep terrain, 5' water tables are higher above water level in the surface water bodies than in flat terrain (Fig. 2c). Deep GW tables that are not only far below the mean land surface elevation but also below the water table of surface water bodies are simulated to occur in some (steep or flat) desert area with very low GW recharge. Negative WTD only occurs in places were the $P_{30}$ is above the mean surface elevation e.g. parts of the Netherlands (Fig. 2b). Less than 10 cells experience WTD smaller than -10 m, which is very likely due to a not fully converged head solution.

In 2.1 % of all cells, GW head is simulated to be above the average land surface elevation, by more than 1 m in 0.3 % and by more than 100 m in 0.004 % of the cells. The shallow water table in large parts of the Sahara is caused by losing rivers (and some wetlands) that cannot run dry in the model, causing an overestimation of the GW table (section 2.2). Please

note that the computed steady-state WTD certainly underestimates the steady WTD in GW depletion areas such as the High Plains Aquifer and the Central Valley in the USA (section S2), North-western India, North China Plain and parts of Saudi Arabia and Iran (Döll et al., 2014) as groundwater withdrawals are not taken into account in the presented steady-state simulation of G³M.

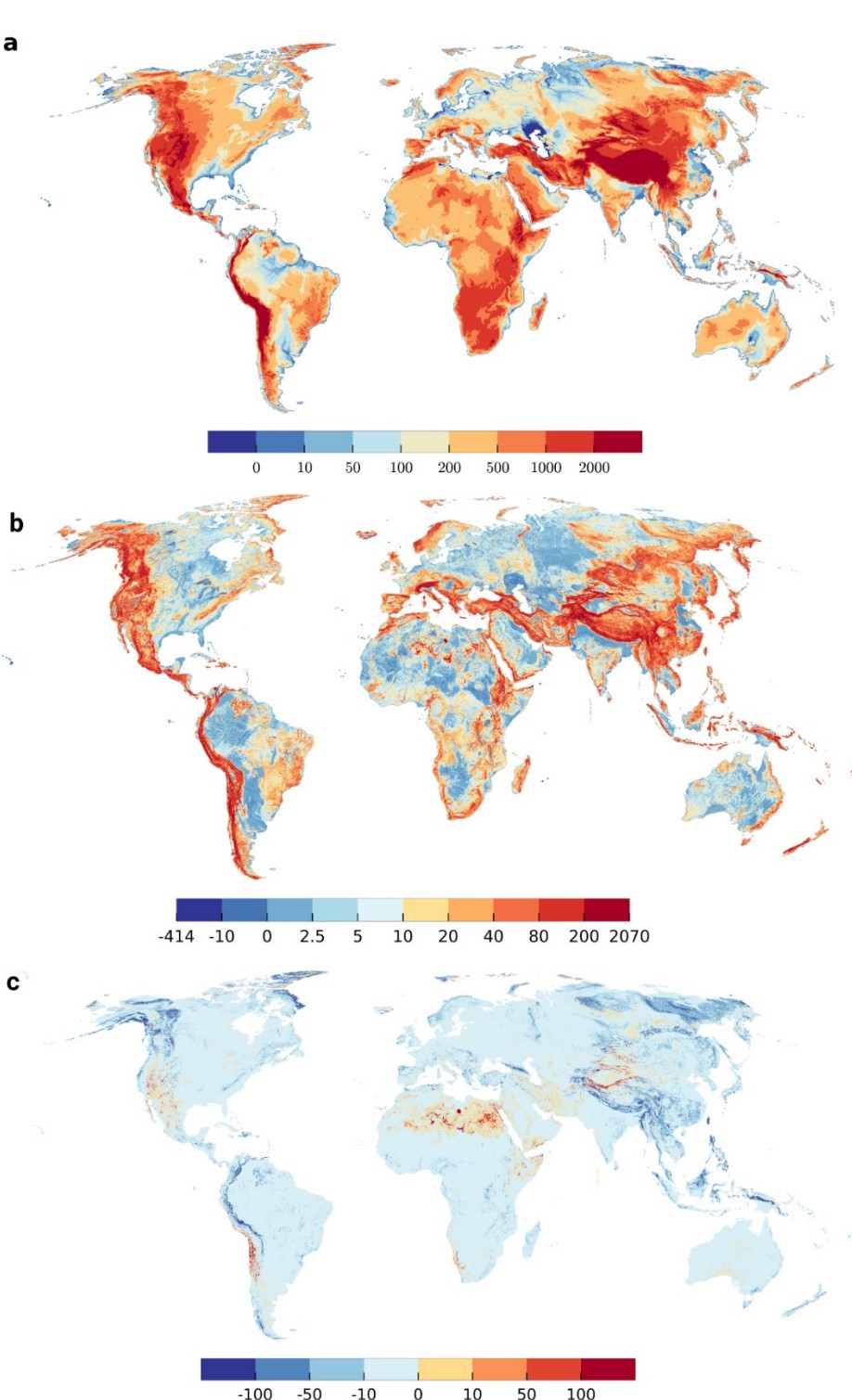

**Figure 2** (a) Simulated steady-state hydraulic head of groundwater above sea level [**m**]. Maximum value 6375 m, minimum -414 m (Extremes included in dark blue and dark red). (b) Water table depth [**m**]. (c) Difference between 30th percentile of the 30" land surface elevation per 5' grid cell (chosen elevation for surface water bodies $h_{swb}$) and simulated groundwater head [**m**]. Maximum value 1723 m, 10 minimum value -1340 m (Extremes included in dark blue and dark red).

**3.2 Global water budget**

Inflows to and outflows from GW of all G³M grid cells were aggregated according to the compartments ocean, river, lake, wetland, and diffuse GW recharge from soil (Fig. 3). The difference between the global sum of inflows and outflows is less than $10^{-6}$ %. This small volume balance error indicates the correctness of the numerical solution.

Total diffuse GW recharge, model input from WaterGAP, from soil is $10^4 \ km^3 \ year^{-1}$ and approximately equal to the simulated flow of GW to rivers (Fig. 3). Rivers are the ubiquitous drainage component of the model, followed by wetlands, lakes and the ocean boundary. According to G³M, the amount of river water that recharges GW is more than one order of magnitude smaller than GW flow to rivers (Fig. 3). Possibly, flow from SW bodies to GW is overestimated, as outflow from SW is not limited by water availability in the SW, and depending on the hydraulic conductivity, Eqs. (4) and (5) can lead to

rather large flows. Inflow from the ocean, which is more than two magnitudes smaller than outflow to ocean, occurs in regions where $h_{swb}$ = P$_{30}$ is below $h_{ocean}$ e.g. the Netherlands. Globally, lakes and wetlands are computed to receive up to $10^3 \ km^3 \ year^{-1}$ of water from GW, and lose 1-2 orders of magnitude less.

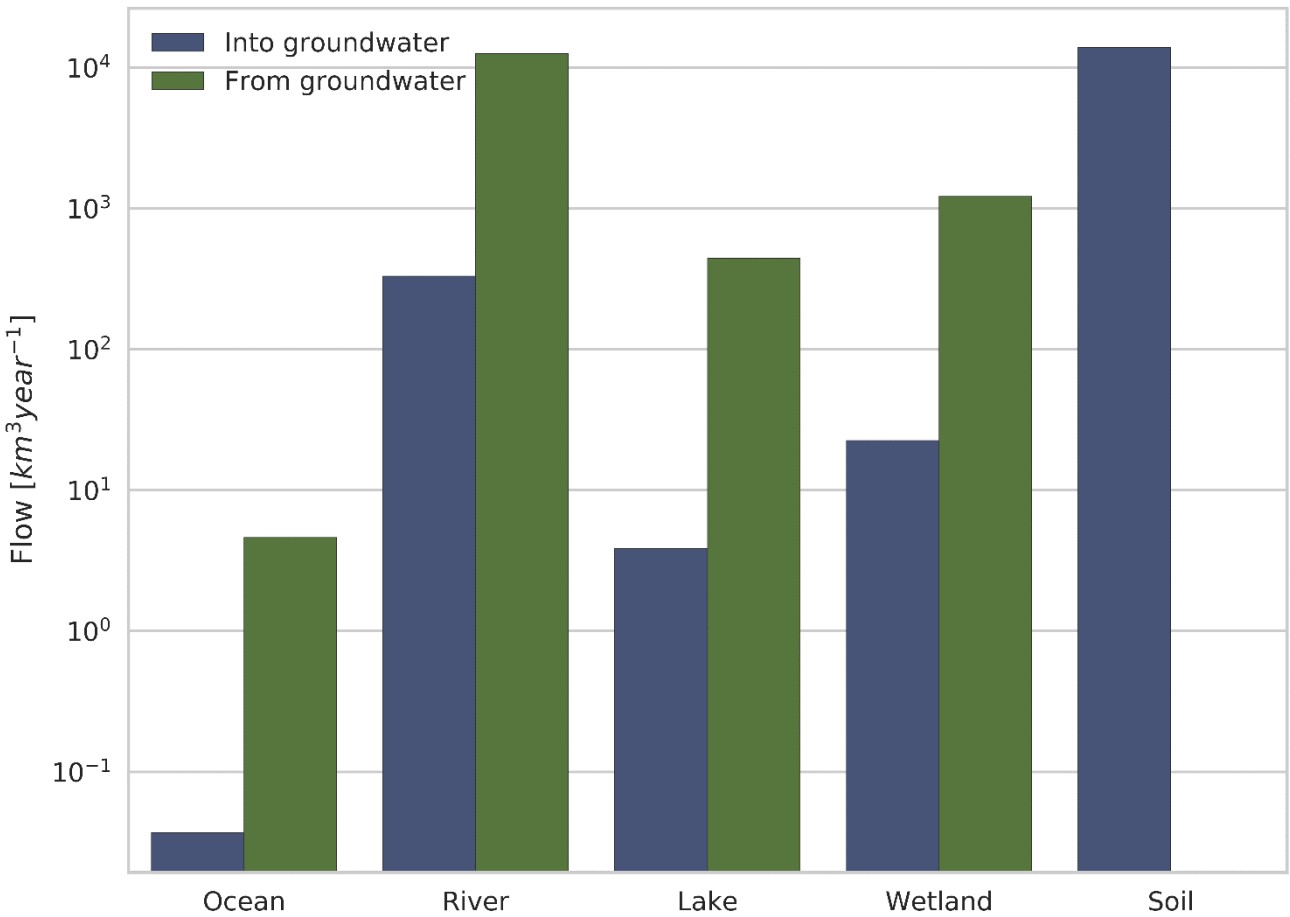

**Figure 3** Global sums of flows from different compartments into or from GW at steady state. Flows into the GW are denoted by the color blue, flows out of the GW into the different compartments by green. The compartment soil is the diffuse GW recharge from soil calculated by WaterGAP.

**3.3 GW-SW interactions**

Figure 4 plots the spatial distribution of simulated flows from and to lakes and wetlands (Fig. 4a) as well as from and to rivers (Fig. 4b). Parallel to the overall budget (Fig. 3), the map reveals the globally large but locally strongly varying influence of lakes and wetlands (Fig. 4a). Rivers with riparian wetlands such as the Amazon River receive comparably small amounts of

GW as most of the GW is drained by the wetland (compare Figs. 4a and 4b). Similarly, areas dominated by wetlands and lakes (e.g. parts of Canada and Scandinavia) show less inflow for rivers (Fig. 4b). In G$^3$M, all SW bodies (rivers, lakes and wetlands) in a grid cell either loose or gain water. Consistent with negative or positive differences between $h_{swb}$ and $h_{aq}$ (Fig. 2c). 93 % of all grid cells contain gaining rivers, and only 7% losing rivers. Gaining lakes and wetlands are found in 12 % and 11 % of

the cells, respectively, whereas only 0.2 % contain a losing lake or wetland.

Gaining rivers, lakes and wetlands with very high absolute $Q_{swb}$ values over 500 mm year$^{-1}$ (averaged over the grid cell area of approximately 80 km$^2$) can be found in the Amazon and Congo basin as well in Bangladesh and Indonesia, where GW recharge in very high (Fig. S4.4). Values below 1 mm year$^{-1}$ occur in dry and in permafrost areas where groundwater recharge is small.

Losing SW bodies are caused by a combination of low GW recharge from soil (Fig. S4.4) and steep mountainous terrain (Fig. S4.7). While the steep Himalayas receive enough GW recharge to have gaining SW bodies, this is not the case for the much dryer mountain ranges around the Taklamakan desert in Central Asia, or mountainous Iran where SW bodies are losing. In the Sahara, GW recharge is so low that SW bodies are losing even in relatively high terrain.

Rivers lose more than 100 mm year$^{-1}$ in Ethiopia and Somalia, West Asia, Northern Russia, the Rocky Mountains

and the Andes whereas lower values can be observed in Australia and in the Sahara. High values of outflow from wetlands and lakes are found in Tibet, the Andes and northern Russia, lower values in the Sahara and Kazakhstan. The river Nile in the Northern Sudan and Egypt is correctly simulated to be a losing river (Fig. 4b), being an allogenic river that is mainly sourced from the upstream humid areas, including the man-made Lake Nasser (Elsawwaf et al., 2014) (Fig. 4a). Furthermore, the following lakes and riparian wetlands are simulated to recharge GW: parts of the Congo River, Lake Victoria, the Ijsselmeer,

Lake Ladoga, the Aral Sea, parts of the Mekong Delta, the Great Lakes of North America. On the other hand, no losing stretches are simulated along the Niger River and its wetlands and almost none in the North-eastern Brazil even though that losing conditions are known to occur there (Costa et al., 2013; FAO, 1997). This is also true when the minimum elevation for SW bodies is assumed (compare Fig. S4.10) leading to the conclusion that the misrepresentation might be linked to an inadequate representation of the local geology.

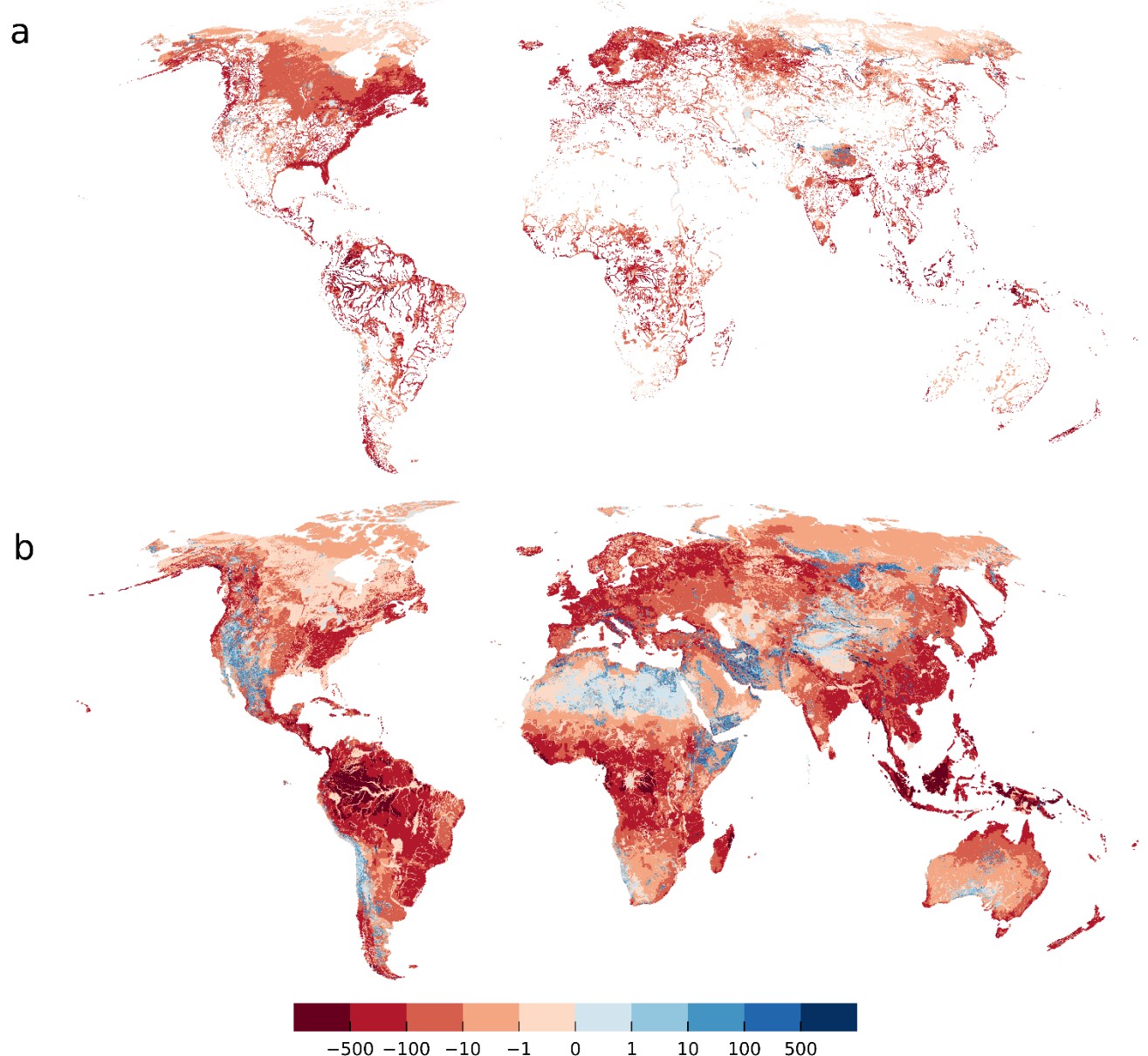

**Figure 4** Flow $Q_{swb}$ [$mm\ year^{-1}$] from/to wetlands, lakes (a) and losing/gaining streams (b) with respect to the 5' grid cell area. Gaining surface water bodies are shown in red, surface water bodies recharging the aquifer in blue. Focused aquifer recharge occurs in arid regions, e.g. alongside the river Nile, and in mountainous regions where the average water table is well below the land surface elevation.

5    Simulated flows between GW and SW depend on assumed conductances for both rivers and lakes/wetlands (Eqs. (3), (4), (5)) shown in Fig. 5. $Q_{swb}$ (Fig. 4) correlates positively with conductance. Conductance for gaining rivers correlates positively with GW recharge (Eq. (5) and Fig. S4.4). High river conductance values are reached in the tropical zone due to a high GW recharge but are capped at a plausible maximum value of $10^7\ m^2 day^{-1}$s  in case of a river (section 2.1) (Fig. 5b). Lakes and wetlands, can have larger values of conductance due to their large areas, e.g. in Canada or Florida.

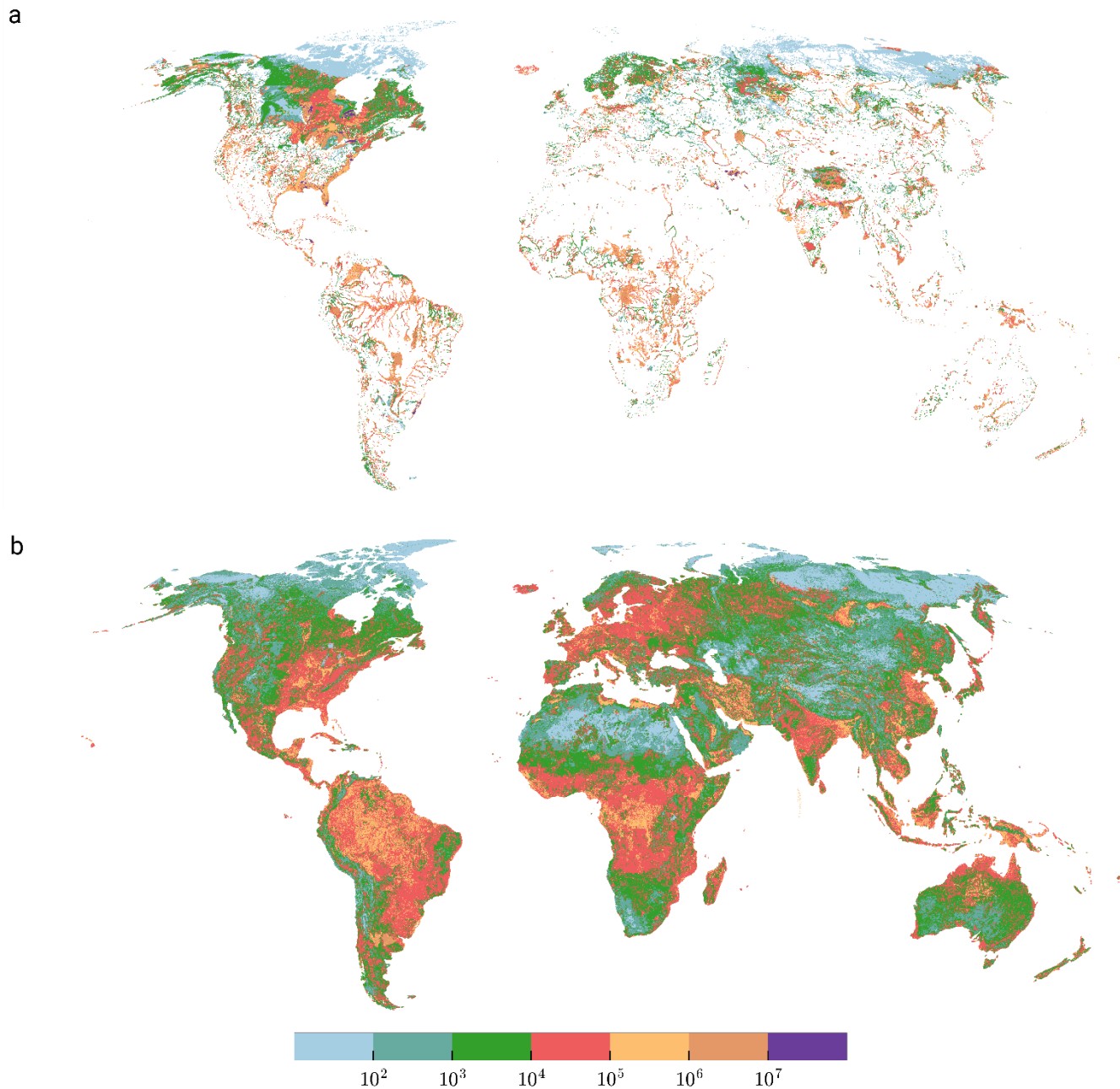

**Figure 5** Conductance [$m^2day^{-1}$] of lakes and wetlands (a) and rivers (b). In regions close to the pole conductance is in general lower due to the influence of the low aquifer conductivity (losing conditions), and relatively small GW recharge due to permafrost conditions (only applies for gaining conditions). Max conductance of wetlands is $10^8$ $m^2day^{-1}$.

## 3.4 Lateral flows

Figure 6 shows lateral GW flow (between grid cells, summing up over all model layers) in percent of the sum of diffuse GW recharge from soil and GW recharge from SW bodies. The percentage of recharge that is transported through lateral flow to neighbouring cells depends on five main factors: (1) hydraulic conductivity (Fig. S4.3), (2) diffuse GW recharge (Fig. S4.4), (3) losing or gaining SW bodies (Fig. 4), (4) their conductance (Fig. 5) and (5) the head gradients (Fig. 2a).

On large areas of the globe, where GW discharges to SW bodies, the lateral flow percentage is less than 0.5% of the total GW recharge to the grid cell, as most of the GW recharge in a grid cells is simulated to leave the grid cell by discharge to SW bodies. For example, in the permafrost regions, the assumed very low hydraulic conductivity limits the outflow to neighbouring cells of the occurring recharge, leading to these very low percent values. Such values also occur in regions with high SW conductances and rather low hydraulic conductivities, e.g. in the Amazon Basin. Values of more than 5% occur where hydraulic conductivity is high even if the terrain in rather flat, such as in Denmark. Higher values may occur in case of gaining

SW bodies in dry areas like Australia or in the Taklamakan desert. They can also be observed in mountainous regions where large hydraulic gradients can develop. In mountains with gaining surface water bodies, lateral outflows may even exceed GW recharge of the cell. In grid cells where SW bodies recharge the GW, outflow tends to be a large percentage of total GW recharge as there is no outflow from GW other than in lateral direction, and values often exceed 100% (Fig. 6).

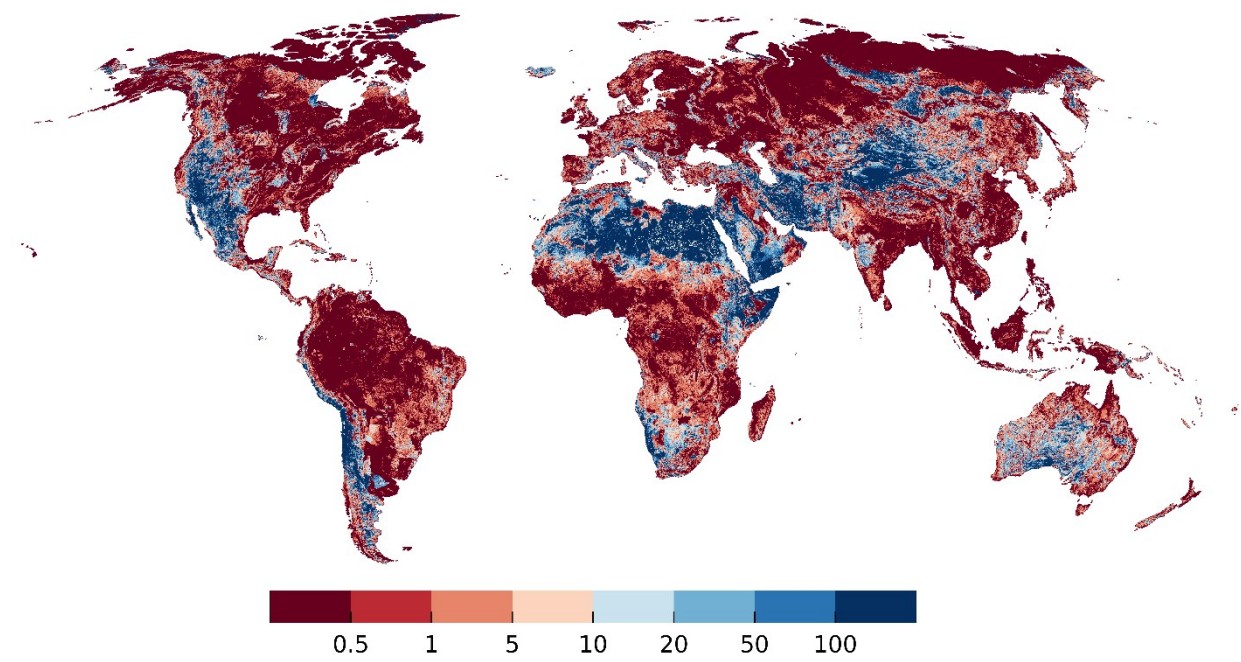

**Figure 6** Percentage of GW recharge from soil and surface water inflow that is transferred to neighboring cells through lateral out flow (sum of both layers). Grid cells with zero total GW recharge are shown in white (a few cells in the Sahara and the Andes).

### 3.5 Comparison to groundwater well observations and the output of two higher-resolution models

Global observations of WTD were assembled by Fan et al. (2007; 2013). We selected only observations with known land surface elevation and removed observations where a comparison to local studies suggested a unit conversion error. This left total of 1,070,402 WTD observations. An "observed head" per 5' model cell was then calculated by first computing hydraulic head of each observation by subtracting WTD from the 5' average of the 30" land surface elevation and then calculating the arithmetic mean of all observations within the 5' model cell. This resulted in 78,664 grid cells with observations out of a total of 2.2 million G³M top-layer grid cells. Multiple obstacles limit the comparability of observations to simulated values. (1) Observations were recorded at a certain moment in time influenced by seasonal effects and abstraction from GW, whereas the simulated heads represent a natural steady-state condition. (2) Observation locations are biased towards river valleys and productive aquifers. (3) Observations may be located in valleys with shallow local water tables too small to be captured by a coarse resolution of 5'.

Simulated steady-state hydraulic heads in the upper model layer are compared to observations in Fig. 7. Shallow GW is generally better represented by the model than deeper GW. Especially the water table in mountainous areas is underestimated, which may be related to observations in perched aquifers caused by low permeability layers (Fan et al., 2013) that are not represented in G³M due to lacking information. Because the steady-state model cannot take into account the impact of GW abstraction, the computed WTD values are considerably smaller than currently observed values in GW depletion areas like the Central Valley in California (where once wetlands existed before excessive GW use depleted the aquifer) and the High Plains Aquifer in the Midwest of the USA. Still, the elevation of the GW table in the non-depleted Rhine valley in Germany is overestimated, too. Overestimates in the Netherlands may partially be due to artificial draining. Figure 8a shows the hydraulic head comparison as scatter plot. Overall, the simulation results tend to underestimate observed hydraulic head but much less than the steady-state model presented by de Graaf et al. (2015) (their Fig. 6).

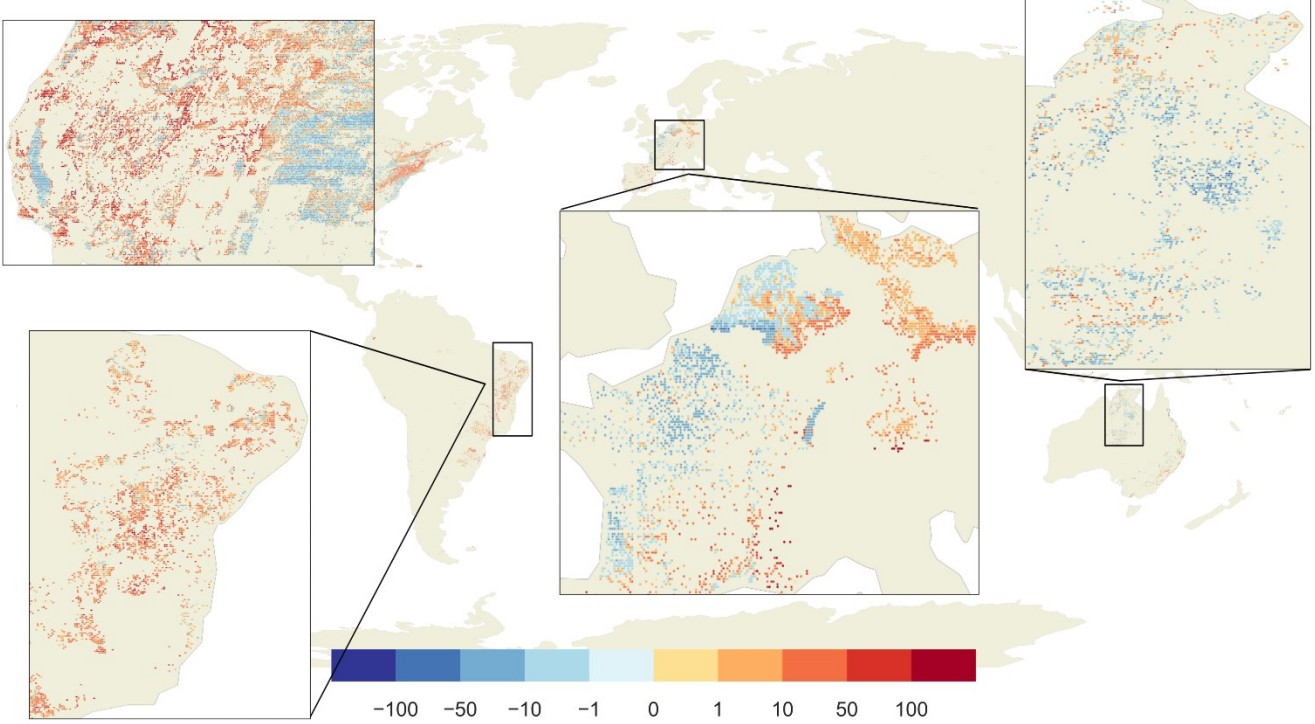

**Figure 7** Differences between observed and simulated hydraulic head [$m$]. Red dots show areas where the model simulated deeper GW as observed, blue shallower GW. In the grey areas, no observations are available.

To compare performance of G³M to the steady-state results of two high-resolution model of Fan et al. (2013) and ParFlow (Maxwell et al., 2015) (Table 1), heads in 30" (Fan et al., 2013) and 1 km (ParFlow) grid cells were averaged to the G³M 5' grid cells. The comparison of 5' observations to the 5' average of ParFlow seems to be consistent with the 1 km model comparison in Maxwell et al. (2015) (their Fig. 5), even though over/under -estimates in the original resolution seem to be smoothed out by averaging to 5' (not shown). The heads of Fan et al. (2013) fit better to observations than G³M heads, with less underestimation (Fig. 8b) and a RMSE (Root Mean Square Error) of 26.0 m compared to the 32.4 m RMSE of G³M. The comparison of G³M heads to Fan et al. (2013) values for all 5' grid cells, which are also the initial heads of G³M and the basis to compute river conductances, show that heads computed with the G³M are mostly much lower except in regions with a shallow GW (Fig. 8c), RMSE is 46.7 m. This cannot be attributed to the 100 times lower spatial resolution per se but to the selection of the 30th percentile of the 30'' as the SW drainage level. Outliers in the upper half of the scatter plot, with much larger G³M heads than the initial values (Fan et al., 2013), are mainly occurring in steep mountain areas like the Himalayas where the 5' model is not representing smaller valleys with a lower head. For the continental US, the computationally expensive 1-km integrated hydrological model ParFlow (Maxwell et al., 2015) fits much better to observations than G³M (Figs. 8d, e), with a RMSE of 14.3 m (ParFlow) compared to 34.2 m (G³M). G³M produces a generally lower water table (Fig. 8f), a main reason being that ParFlow assumes an impermeable bedrock at a depth of 100 m below the land surface elevation.

The global map of head comparison (Fig. 7) suggests that G³M performs resonably well in flat areas compared to mountainous regions. This is corroborated by Fig. 9 that shows the difference between observed and simulated hydraulic heads for five land surface elevation categories. It is evident that model performance deteriorates with increasing land surface elevation and positively correlates with variations of land surface elevation within each grid cell (Fig. S4.7).

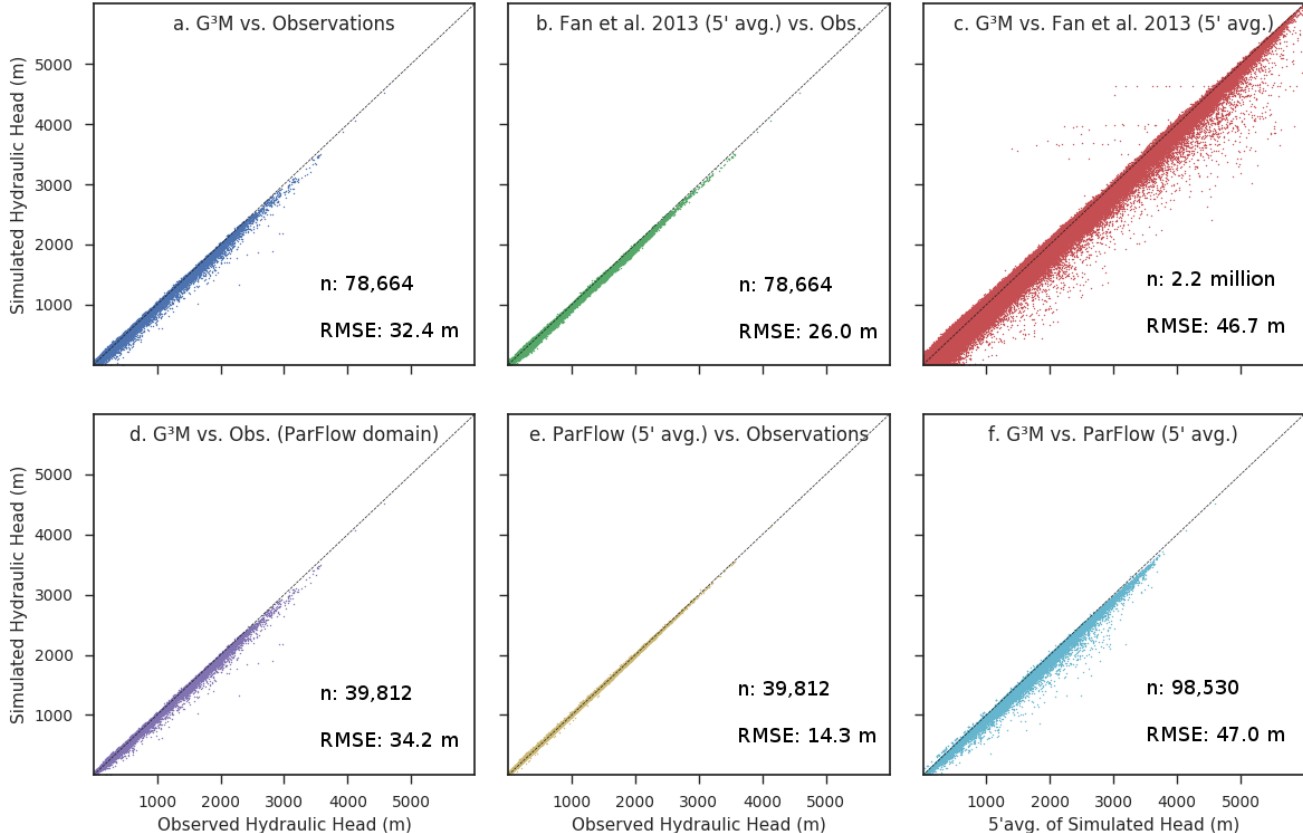

**Figure 8** Scatterplots of simulated vs. observed hydraulic head and inter-model comparison of heads. (Upper panel) The steady-state run of G³M vs. observations (a), the 5' average of the equilibrium head of Fan et al. (2013) vs. observations (b) and the avg. equilibrium vs. G³M (c). (Lower panel) The steady-state run of G³M vs. observations only for the ParFlow domain (d), the 5' average of the ParFlow average annual GW table (Maxwell et al., 2015) vs. observations (e) and the steady-state run of G³M vs. 5' average of the ParFlow average annual GW table (f).

Plotting hydraulic head instead of WTD has the disadvantage that the goodness of fit is dominated by the topography as the observed heads are calculated based on the surface elevation of the model. Well observations provide WTD and only sometimes contain complementary data specifying the elevation at which the measurements were taken. Even though hydraulic heads are a direct result of the model and are forcing lateral GW flows, WTD is more relevant for processes like capillary rise. For G³M, there is almost no correlation between WTD observations and simulated values. To our knowledge, no publication on large-scale GW modeling has presented correlations of simulated with observed WTD.

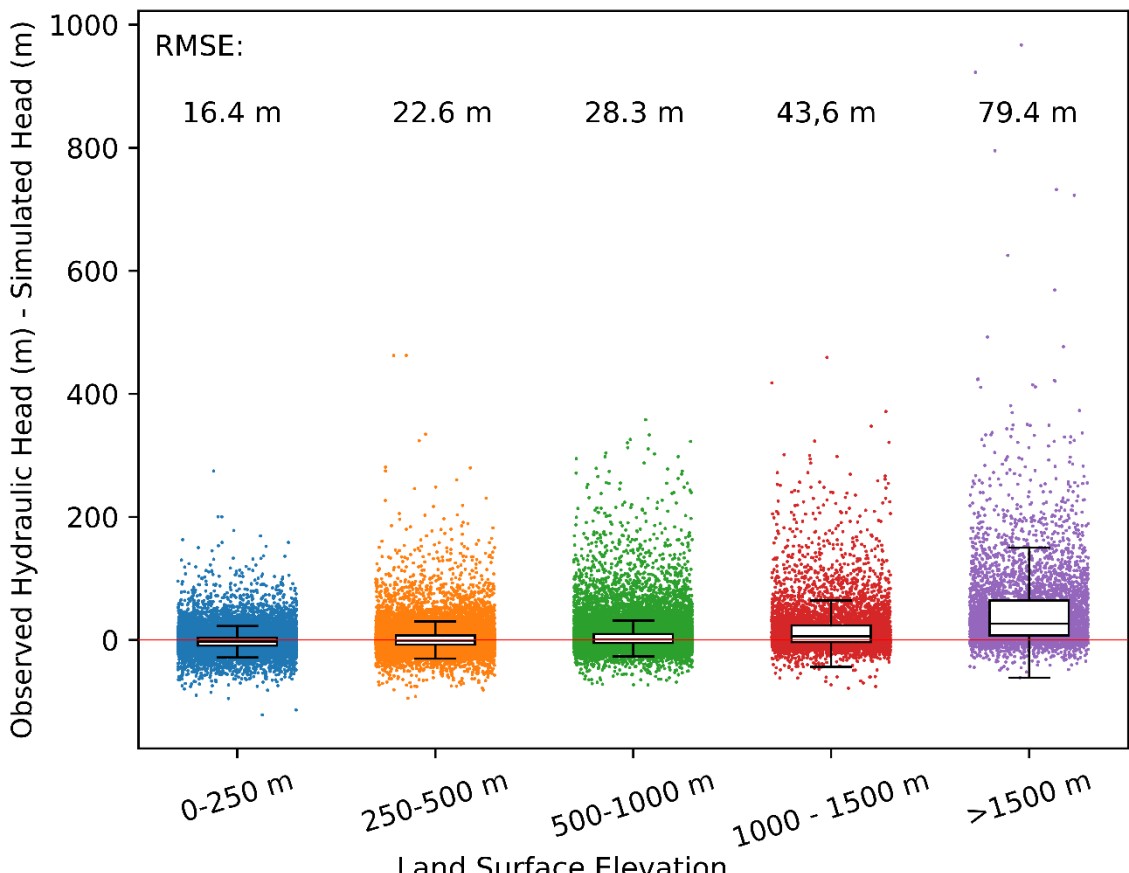

**Figure 9** Observed minus simulated hydraulic head for different land surface elevation categories. The whiskers of the boxplots show the interquartile range.

### 3.6 Testing sensitivity of computed steady-state hydraulic heads to parameter values and spatial resolution

To limit the computational effort for assessing model sensitivity to both parameters and grid size, we selected New Zealand as a representative "small world" that includes a complex topography and the ocean as a clear boundary condition. All inputs and parameters are the same as in the global 5' model.

### 3.6.1 Parameter sensitivity

To determine to which parameters simulated hydraulic heads are most sensitive to, we used the established sensitivity tool UCODE 2005 (Hill and Tiedeman, 2007) to compute composite scaled sensitivity (CSS) values for seven model parameters (S3). CSS of $h_{swb}$ is orders of magnitude larger than the CSS of the other parameters. This confirms our observations during model development when an appropriate value for had to be found (section 2.2). The second-most important parameter is $K_{aq}$, the third most important $R_g$. CSS of the conductance of lakes is one magnitude less than CSS of $R_g$ but as only few cells contain lakes, the CSS value that averages over all grid cells indicates a large sensitivity to $c_{Lakes}$ for grid cells with lakes. Simulated hydraulic heads were found to be rather insensitive to changes in the conductance of rivers, wetlands, and ocean boundary.

### 3.6.2 Sensitivity to spatial resolution

The extremely high sensitivity of simulated hydraulic heads to the choice of $h_{swb}$ (section 3.6.1) and the better agreement of the continental models with a higher spatial resolution of approx. 30" (the Fan et al. (2013) model and ParFlow (section 3.5)) motivated us to run G³M for New Zealand with a spatial resolution of 30", to understand the impact of spatial resolution on simulated hydraulic heads. The 30" G³M model uses the same input as the 5' model except for the land surface elevation, $h_{swb}$

and the location of rivers. While the total length and width of the rivers is equal in both models, a river is assumed to exist in all 5' grid cells, the river is concentrated, in the 30" model, to a few grid cells with each 5' grid cell. The river cell locations at 30" are determined based on 30" HydroSHEDS (hydrosheds.org) information on flow accumulation. Starting with the 30" cell with the highest number of upstream cells per 5' cell, a river is added to this 30" cell using the length and information of HydroSHEDS till the size of the river of the 5' model is reached for all 30" cells within a 5' cell. The areal fractions of all other SW bodies from 5' grid data where used for all 30" grid cells within the 5' grid cell. $h_{swb}$ is set to the land surface elevation.

Figure 10 compares the performance of the two model versions. The comparison of simulated hydraulic head to observations for the Canterburry region (Westerhoff et al., 2018) shows that the overall performance of the 30" model is better, with a smaller RMSE of 26.7 m as compared to a RMSE of 53.8 m in case of the original spatial resolution of 5'. The 30" model results in generally lower simulated hydraulic heads leading to a closer fit to the observed values. This is likely caused by the improved estimation of SW body elevation, which generally leads to lower estimates of $h_{swb}$. On the other hand, overestimates of observed hydraulic heads prevails in the 30" model, even though $h_{swb}$ was set to the land surface elevation, indicating that further investigation is necessary. The underestimates are likely due to large GW abstractions for irrigation in the particular region (Westerhoff et al., 2018).

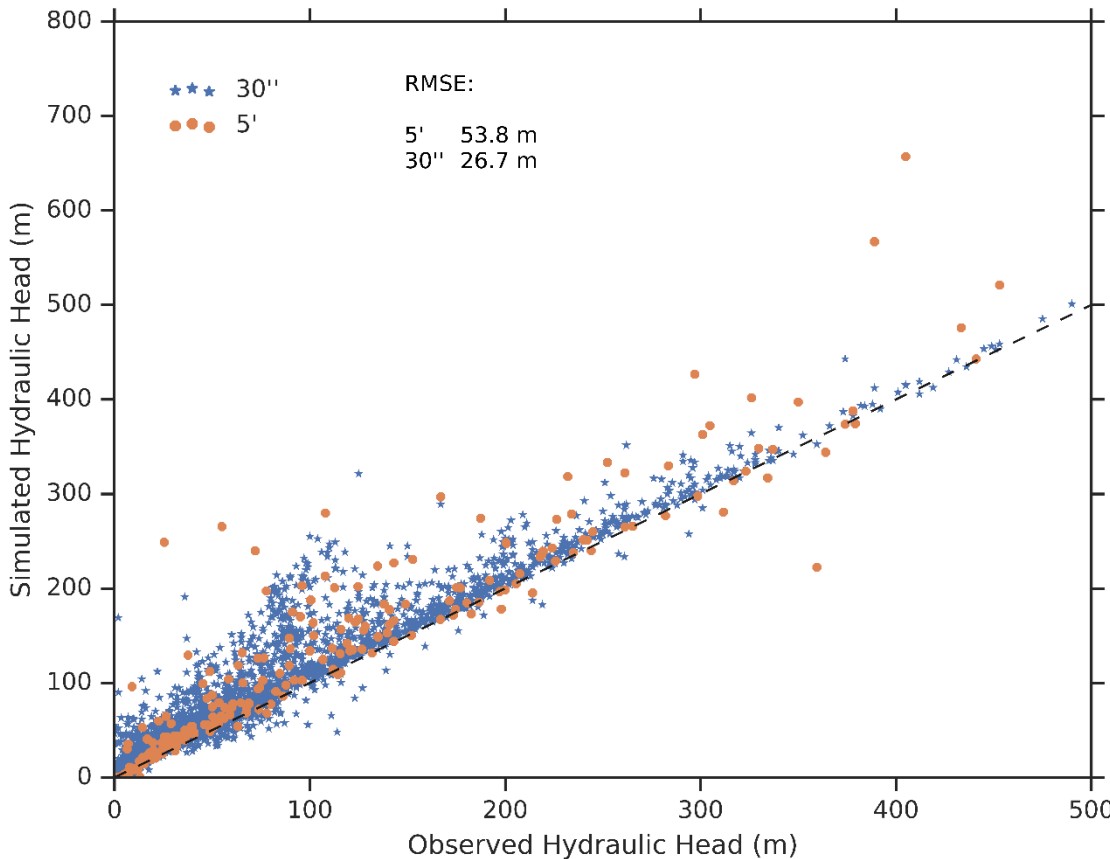

**Figure 10** Low (5') vs. high (30") spatial resolution for the Canterburry region in New Zealand, comparison of observed vs. simulated hydraulic head for both resolutions. The observed head is the geometric mean per 5' and 30" respectively.

## 4 Discussion

The objective of global gradient-based groundwater flow modelling with G³M is to better simulate water exchange between SW and GW in the GHM WaterGAP, for example for an improved estimation of GW resources in dry regions of the globe that are augmented by focused recharge from SW bodies. We assume that the fully coupled model will lead to an improved WaterGAP performance during droughts with an increased drop in streamflow due to the now possible switch from gaining to losing conditions. Presumable a calibration of $c_{swb}$ and $c_{riv}$ is necessary to achieve a good discharge performance. The presented steady-state model is a first step in this direction. It helped to understand basic model behaviour, e.g. the sensitivity

to SW body elevation, and the necessary improvement of its parameterization, before moving to the more complex integrated transient model. The reduced runtime of the steady-state model in comparison to a fully integrated transient run supported the investigation of parameter sensitivity and sensitivity to spatial resolution. Additionally, the presented steady-state model can be used in future fully integrated transient runs as initial condition.

A major challenge for simulating GW-SW interactions (but also capillary rise) at the global scale is the large size of grid cells that is required due to computational constraints. Within the 5' grid cells, land surface elevation at the scale of 30" very often varies by more than 20 m, and often by 200 m and more (Fig. S4.1), while the vertical position of the cell and the hydraulic head are approximated in the model by just one value. The question is whether head-dependent flows between grid cells, between GW and SW and from GW to soil (capillary rise) can be simulated successfully at the global scale, i.e. whether

an improved quantification of these flows as compared to the simple linear reservoir model currently used in most GHMs can be achieved by this approach. This question cannot be answered before a dynamic coupling of G³M with a global hydrological model has been achieved but one may speculate that some innovative approach to take into account the elevation variations within the grid cells is needed.

It is difficult to assess quality of the presented steady-state G³M results. Model performance is assessment is hindered

by data availability and the coarse model resolution. (1) To our knowledge the data collection of depth to groundwater by Fan et al. (2013) is unique. However, they do not represent steady-state values. Apart from depth to groundwater observations, hardly any relevant data is available at the global scale. Especially exchange between surface water and groundwater is difficult to measure even at the local scale. Therefore, we compared G³M results with the results from other large-scale models. Comparison to the results of catchment-scale groundwater flow models is planned for transient runs that will be possible after

integration into WaterGAP. (2) Scale differences make the comparison to point observations of depth to groundwater difficult. Often, observations are biased towards alluvial aquifers in valleys. The calculated hydraulic head of the grid cell may represent the average groundwater level per grid cell correctly but can be still far off the local observations of depth to groundwater. As the current model only represents an uncalibrated natural steady-state, a comparison to observations only provides a first indicator where the model and the performance measurements needs to be improved as we move to a fully transient model.

The presented development of the uncoupled steady-state global GW flow model enabled us to better understand how the spatial hydraulic head pattern relates to the fundamental drivers topography, climate and geology (Fan et al., 2007) and how the interaction to SW bodies governs the global head distribution. Simulated depth to groundwater is particularly affected by the assumed hydraulic head in SW bodies, the major GW drainage component in the model. As rivers represent a natural occurring drainage at the lowest point in a given topography, one would assume that the minimum elevation 30" land surface

elevation per 5' grid cell is a reasonable choice. Experiments have shown that this will induce a head distribution well below the average 5' elevation that is much below observations of Fan et al. (2013). We also tested setting $h_{swb}$ to the average elevation of all "blue" cells (with a WTD of less than 0.25 m) of the steady-state 30" water table results of Fan et al. (2013) that indicate the locations were GW discharges to the surface or SW bodies. This leads to an overall underestimation of the observed hydraulic heads (Fig. S4.9) as the assumed SW elevation is too low. Furthermore, it leads to an increase in losing

SW bodies (comp. Fig. S4.10 with Fig. 4). However, it is difficult to judge whether this improves the simulation. More stretches of the Nile and its adjacent wetlands and also of the Niger wetlands and rivers in North-eastern Brazil are losing in case of lower $h_{swb}$, which appears to reasonable. Additionally, choosing the average as SW elevation provides on the on hand a better fit to observations (Fig. S4.9 right) but leads to a world- wide flooding (Fig. S4.9) and a much longer convergence time due to an increased oscillation between gaining and losing conditions.

The problem is very likely one of scale. All three models (Fan et al., ParFlow, and G³M 30") (Table 1) high-resolution models, even the simple one of Fan et al. (2013), fit better to observations than the 5' model G³M (Fig. 8,10). In case of high resolution, there are a number of grid cells at an elevation above the average 5' land surface elevation, leading to higher hydraulic heads in parts of the 5' area that drain towards the SW body in a lower 30" grid cell. In case of the low spatial

resolution of 5' in which $h_{swb}$ is set to the elevation of the fine-resolution drainage cell, the 5' hydraulic head is rather close to this (low) elevation (Fig. S4.8 center), resulting in an underestimation of hydraulic head and thus an overestimation of WTD. While it is plausible and necessary to assume that there is SW-GW interaction within each of in the approximately 80 km², this is not the case for the two orders of magnitude smaller 30" grid cells. Thus, with the high resolution, heads are not strongly

controlled everywhere by the head in SW bodies. Selecting the 30$^{th}$ percentile of the 30" land surface elevation as $h_{swb}$ was found, by trial-and-error, to lead to a hydraulic head distribution that fits reasonably well to observed head. It avoids that the simulated GW table drops too low while avoiding the excessive flooding that occurs if $h_{swb}$ is set to the average of 30" land surface elevations, i.e. the 5' land surface elevation (Fig. S4.9).

    The constraint that the selected $h_{swb}$ value puts on simulated hydraulic heads is also linked to the conductance of the

SW bodies. A higher conductance will lead to aquifer heads closer to $h_{swb}$. If the hydraulic head drops below the bottom level of the SW body, the hydraulic gradient is assumed to become 1 and the SW body recharges the GW with a rate of $K_{aq}$ per unit SW body area. In case of a $K_{aq}$ value of 10$^{-5}$ m s$^{-1}$, the SW body would lose approximately 1 m of water each day. Further investigations are needed regarding the appropriate choice of SW body elevation and conductance. The simple conductance approach applied in G³M could possibly be improved by the approach by Morel-Seytoux et al. (2017) who proposes an

analytical, and physically based, estimate of the leakance coefficient for coarse scale models based on river and aquifer properties.

    De Graaf et al. (2015) set their SW head ($h_{swb}$) to the mean land surface elevation (Table 1) of the 6' grid cells minus river depth at bankfull conditions plus water depth at average river discharge as compared to P$_{30}$ in the 5' G³M. Together with the missing interaction between lakes and wetlands and a different approach to river conductance, this might be a reason for

the additional drainage above the floodplain that was necessary to improve the discharge to rivers (Sutanudjaja et al., 2014). On the other hand, the additional drainage leads to drainage of water even if the hydraulic head is below the SW elevation, which might have led to the global underestimation of hydraulic heads. Thus, the difference in model heads seems to be closely related to the sensitivity to SW body elevation.

    Due to the spatial resolution and lack of data G³M does not capture the actual variability of topography, aquifer depth

(Richey et al., 2015) and (vertical) heterogeneity of subsurface properties. The lack of information about the three-dimensional distribution of hydraulic conductivity is expected to negatively impact the quality of simulated GW flow. For example, the lateral conductivity and connectivity of groundwater along thousands of kms from e.g. the Rocky Mountains in the Central USA to the coast as well as the vertical connectivity is likely to be overestimated by G³M, as vertical faults and interspersed aquitards are not represented; this is expected to lead to an underestimation of hydraulic head in those mountainous areas.

**5 Conclusions**

We have presented the concept and first results of the new global gradient-based 5' GW flow model G³M that is to be integrated into the 0.5° GHM WaterGAP. The uncoupled steady-state model has provided important insights into challenges of global GW flow modelling mainly related to the necessarily large grid cells size (5' by 5'). In addition, first global maps of SW-GW interactions were generated. Simulated heads were found to be strongly impacted by assumptions regarding the interaction

with SW bodies, in particular the selected elevation of the SW table. We have demonstrated that simulated G³M hydraulic heads fit better to observed heads than the heads of the comparable steady-state GW model of de Graaf et al. (2015), without requiring additional drainage. Furthermore, we provided insights into how the choice of surface water body elevation $h_{swb}$ affects model outcome. In a next step, approaches for utilizing high-resolution topographic data to improve the selection of $h_{swb}$ will be investigated.

The presented results are the first step towards a fully coupled model in which SW heads are jointly computed, also taking into account the impact of SW and GW abstraction. Especially the interaction with SW bodies that can run dry will

make the G$^3$M behavior more realistic. The fully coupled model will simulate transient behaviour reflecting climate variability and change. Simulated hydraulic head dynamics will be compared to observed head time series as well as to the output of large-scale regional models, while total water storage variations will be compared to GRACE satellite data. However, it will be challenging to judge the dynamics of GW and the quality of simulated GW-SW interactions due to a scarcity of observations.

## 6 Code and data availability

The model-framework code is available at globalgroundwatermodel.org or at DOI: 10.5281/zenodo.1175540 with a description on how to compile and run a basic GW model. The code is available under the GNU General Public License 3. Model output is available at DOI: 10.5281/zenodo.1315471.

**Author contribution**

RR led conceptualization, formal analysis, methodology, software, visualization, and writing of the original draft. LF and SM supported review and editing as well as the development of the methodology. TT, CD supported review. PD supervised the work of RR and made suggestions regarding analysis, structure, and wording of the text and design of tables and figures.

**Acknowledgments**

Part of this study was funded by a Friedrich-Ebert foundation Ph.D. fellowship. We are very grateful to Ying Fan and Gonzalo Miguez-Macho for fruitful discussions and data provisioning. We thank the reviewers for their thoughtful comments that helped to improve the manuscript. Furthermore, we like to thank Alexander Wachholz for contributing to the New Zealand figures and Rogier Westerhof for providing the observations for New Zealand.

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

**Table 1** Comparison of global- and continental-scale groundwater models

| Aspect | G³M | de Graaf et al. (2015; 2017) | Fan et al. (2013) | ParFlow |
|---|---|---|---|---|
| Extent | Global | Global | Global | Continental USA |
| Resolution | 5' | 6' | 30" | 1 km |
| Software | G³M-f | MODFLOW | Unnamed | ParFlow |
| Computational expense | Medium | Medium | High | Very high |
| Flow representation | 3D saturated | 3D saturated | 2D saturated | 3D saturated/unsaturated |
| Time scale | Steady-state/(transient) | Steady-state/transient | Steady-state | Steady-state |
| Vertical layers | 2 | 2 | 1 | 5 |
| Full coupling possible | Yes | No (Conceptual issue) | No | Yes (already coupled) |
| In-memory coupling | Yes | No | N/A | Yes |
| Constant saturated thickness | Yes | Yes | No | No |
| Impermeable bottom | No | No | No | Yes |
| Surface water body location | In every cell | In almost every cell | No surface water | Created during simulation |
| Surface water body elevation | $P_{30}$ of 30" DEM | Avg. of 30" DEM | N/A (outflow if WTD < 0.25 m) | N/A |
| Deviation from observations | Large | Very large | Medium | Medium |

**Table 2** Model parameter values, input data sources and other information about the steady-state simulation.

| Parameter | Symbol | Units | Description | Eq. No. |
|---|---|---|---|---|
| Landmask | - | - | Location and area of 2161074 cells at 5' resolution based on WaterGAP (Eisner, 2016)) | - |
| GW recharge | $R_g$ | $L^3 T^{-1}$ | Mean annual diffuse GW recharge 1901–2013 of WaterGAP 2.2c (Müller Schmied et al., 2014) forced with EWEMBI (Lange, 2016), spatial resolution 0.5° (Fig. S4.4) | 2,5,S1 |
| Hydraulic conductivity | $K_{aq}$ | $LT^{-1}$ | Derived from Gleeson et al., 2014 (Fig. S4.3) | 1,3 |
| Hydraulic head | $h_{(aq)}$ | $L$ | Head of the aquifer in a computational cell, initial estimate based on 5' average of 30" head of Fan et al. (2013) | 1,6,5 |
| Ocean boundary conductivity | $c_{ocean}$ | $L^2 T^{-1}$ | $10\ m^2\ day^{-1}$ | 2,6 |
| Ocean boundary head | $h_{ocean}$ | $L$ | Global mean sea-level of 0 m | 6 |
| SW head | $h_{swb}$ | $L$ | 30 % quantile ($P_{30}$) of 30" land surface elevation of Fan et al. (2013) per 5' grid cell | 3 |
| SW bottom elevation | $B_{swb}$ | $L$ | 2 m (wetlands), 10 m (local lakes), 100 m (global lakes) below $P_{30}$ | 4 |
| Area of global and local lakes and global and local wetlands | $WL$ | $L^2$ | Per 5' grid cell, based on WaterGAP 3 (Eisner, 2016), | 4 |
| Length of the river | $L$ | $L$ | Per 5' grid cell, based on WaterGAP 3 (Eisner, 2016) | 4 |
| Width of the river | $W$ | $L$ | Per 5' grid cell, based d on WaterGAP 3 (Eisner, 2016) | 4 |
| River head | $h_{riv}$ | $L$ | $h_{swb}$ | 4,5 |
| River bottom elevation | $B_{riv}$ | $L$ | $h_{riv} - 0.349 \times Q_{bankfull}{}^{0.341}$ (Allen et al., 1994) | 5 |
| Equilibrium hydraulic head | $h_{eq}$ | $L$ | Steady-state hydraulic head of Fan et al. (2013) (averaged to 5' from original spatial resolution of 30") | 5 |
| Layers | - | - | 2 confined, 100 m thick each | - |
| Land surface elevation | - | $L$ | 5' average of 30" digital elevation map of Fan et al. (2013) (Fig. S4.2) | - |
| E-folding factor | - | - | Applied only to lower layer for 150 m depth, based on area-weighted average of Fan et al. (2013) | - |
| Timestep | $t$ | $T$ | Daily timestep | - |
| Head convergence criterion (outer loop) | - | $L$ | Max head change globally < 10 m in three consecutive iterations | - |
| Residual convergence criterion (inner loop) | - | - | $\|$conjugate gradient residuals $\|_{inf} < 10^{-100}$ | - |
| Maximum number of inner iterations | - | - | Maximum 50 inner iterations between outer Picard iterations (Naff, Richard L., and Edward R. Banta, 2008) | - |