# Peer review of "Challenges in developing a global gradient-based groundwater model (G3M v1.0) for the integration into a global hydrological model"

_Geoscientific Model Development, 2018_

## Referee Comment (RC1) · Anonymous Referee #1 · 10 Jun 2018

The paper describes the development and application of a global gradient based hydrogeological model. I can clearly see the use of developing large-scale models. However, there are a range of significant issues related to this paper. I do not support publication at this stage. The reasons are as follows. A fundamental problem with these models is that they are difficult to verify, but this is not at all reflected in the discussion of the results. The authors state on line 24 page that these models are useful in areas with little or no data, as they allow to generate robust information. How can anything robust be generated (and how do we know its true?) in the absence of data. The hydrogeological literature is full of examples where even in the data -rich regions different models produce different outcomes.

[Figure]

We are presented with plots, numbers and graphs and some interpretation, but there is no credible discussion on the reliability of the result obtained. The only indication of model performance is that there is essentially no correlation between simulated and observed depth to groundwater. To me this means simply that the model cannot be used to make these types of predictions.

It is not useful to plot observed and simulated hydraulic heads over such large scales, even if its just for the sake of model comparison. It is true that other authors have also presented simulated vs observed hydraulic heads over such large scales, but this is simply misleading. Depth to groundwater is the variable that counts for calculating exchanges with surface, amongst many other processes. In this sense none of the available models on a global scale is ready yet. This must not necessarily be a problem, as long as the results are not oversold, as is unfortunately rather often the case.

The formulation of the equation 2 is for a confined aquifer. The authors justify this conceptually wrong choice on line 20, page 6: "Flow equations are for confined aquifer because it reduces convergence time. "This is a very poor argument, purely based on convenience. To what extent the model should capture the relevant physics should cannot be a question on how difficult it is to solve equations. The goal of this modelling approach is to advance the interaction between the surface and the subsurface across very large scales. Given that the direct interaction with the surface always happens with unconfined aquifers the fundamental basis of the approach is flawed on the most basic level. While for steady state simulations the term falls out of the equation it is still very concerning that that a model is developed with inadequate flow equations.

The authors will probably argue that this is a first step in model development and that unconfined conditions can be added later. However, a large part of the model is presenting model- simulations. For example, we learn how much water if flowing from aquifers to rivers on a global scale. Given the formulation of the model, these results should not be presented.
[Figure]

I did not understand why the authors develop a new model in the first place. They rightfully acknowledge that models such as MODFLOW exist, and these model could potentially do the job. Their argument is that MODFLOW models typically integrate geological data that is not available on a global scale. Therefore, a simplified model is developed. But this is a strange way of reasoning, as with MODFLOW one is not obliged to integrate all the geological complexity. It would have been perfectly possible to use MODFLOW for this project , with several significant advantages: for example, an unconfined aquifer (see below) could have been simulated. In this sense the novelty of the aspects concerning model development is questionable.

There are many other problems working on a global scale which are not even mentioned here but will even further undermine the credibility of the model. The three most important ones are: (1) Elevation is the wrong parameter for such a model. The data that should be used is not an ellipsoid- DEM but rather a geoid as the geoundulation is significant. (2) The density of sea-water is different, therefore there should be a density correction. (3) Steady-state conditions are inappropriate assumption that is not justified sufficiently well.

Validation is done with other macro-scale models. This is a not an ideal strategy, as these large-scale models suffer from similar deficiencies (even though on less fundamental level). For a solid assessment of model performance a detailed, catchment scale hydrogeological model should be used for a benchmark comparison. On line 28, page 7 the authors highlight that this is ok –"...without losing important model behavior. " Transient and steady state is significantly different in both spatial and temporal dynamics.

The description of the conductance is confusing. In MODFLOW L is not the length of the river itself, but the length of the river within a grid cell. But this might just be an imprecise formulation.

Other aspects also require more justification and discussion. Why only 8 % of wetland

surfaces? Where does this number come from? What are the numerical convergence criteria, as well as a wide range of additional model parameters?

---

## Short Comment (SC1) · 22 Jun 2018

As explained in https://www.geoscientific-model-development.net/about/manuscript_types.html the preferred reference to code release is through the use of a DOI which is then cited in the paper. As the model version is already published on GitHub a DOI can easily be created using for instance Zenodo, see https://guides.github.com/activities/citable-code/ for details. As also stated in the guide lines E-mail contact to obtain access is not preferred and simulations and data should be made available as supplement, as DOI or as part of the release.

Lutz Gross GMD Executive Editor

---

## Referee Comment (RC2) · Anonymous Referee #2 · 29 Jun 2018

Review of "Beyond the bucket . . ." by Reinecke et al

This work presents a global groundwater model, coded based on the MODFLOW formulation to couple to WaterGAP. Model formulation is presented. Steady state simulations driven by recharge were conducted and are compared to observations and other models (which I applaud). This work is interesting, as global groundwater stores are of tremendous scientific importance to the hydrologic community and the paper is generally clearly written. I have listed several points below that I think need to be addressed before suitability for publication can be assessed. Hopefully these points help the authors focus this work and improve the clarity and quality.

Bigger picture, conceptual questions:

1. Is 5' an appropriate resolution at which to simulate groundwater flow? The analysis by Krakauer et al may be useful in determining the appropriate resolution.

2. The work is coupled to WaterGap at 0.5deg, this is a really large scale discrepancy — how do you think this might alter the model results?

3. The comparisons between this study and Fan et al and Maxwell et al are interesting. While pressure head is important, I think the bias from these scatterplots, basically water table depth, is more meaningful (as plotted in Fan et al / Maxwell et al too). The statistics will really be driven by topography which can occlude model performance and differences.

4. The diagram for how the model handles topographic breaks (Fig 1) is super confusing. Basically is water moved between cells even if there is a disconnect?

5. The assumption of confined conditions really seems hard to justify. This is effectively what de Graaf et al (2015, 2017) do with their two layer MODFLOW model with a stream package connection to PCRGLOB. There are so many assumptions present I think more careful discussion of how sensitivities in these assumptions (e.g. parameters in what amounts to the stream package used here) and feedback back to the WaterGap (which I think is just one-way at this point) would be really important.

6. From Figure 2 it appears that not all the features are implemented in this model, or perhaps not all the features are activated except for recharge. Since the abstract discusses capillary subsides for plant water use but this feature is not described (nor is it entirely clear how that would be implemented as a simple flux), I think a thorough re-working of this discussion and assumptions are needed. Unfortunately, this figure begs the question why is a methods paper in GMD incomplete and not presenting all the model features?

7. The maps of water table depth seem to have a tremendous shallow bias. It is

hard to say because of low figure resolution, but perhaps most of Eastern N America, most of Australia, half of Europe and all of Tropical Africa are under water. I think additional discussion is needed here at least. Could this be due to the steady state assumptions? Confined conditions? The stream aquifer package? Resolution and slope? ET feedbacks?

8. It's hard to tell what the difference is here between the PRCGlob-MODFLOW model and this current model. More discussion is needed to clarify this distinction. I actually feel it's okay if there are many similar models out there (and both can be good models or bad models, it's not a competition), I would like more dissection of the differences in approach.

9. The current model is also completely different from the Central Valley model. This strikes me as odd too. Is it water use? Boundary conditions?

References Krakauer et al Groundwater flow across spatial scales: importance for climate modeling, ERL (9), 2014
* * *

---

## Author Comment (AC1) · 24 Jul 2018

We thank both reviewers for the thoughtful comments and questions. They helped us in particular to improve our explanation of the conceptual approach to gradient-based groundwater modeling that is necessary for global-scale groundwater modeling with a coarse spatial resolution, including the choice of simulating unconfined conditions and the conceptual difficulties in defining depth to groundwater table.

Referee #1

**1.1 A fundamental problem with these models is that they are difficult to verify, but this**

is not at all reflected in the discussion of the results.

Reply: We have added a new paragraph to section 4 Discussion.

Changes to manuscript: We added the following paragraph to section 4 Discussion (second paragraph) "It is difficult to assess performance of the presented steady-state G$^3$M results. Model performance assessment is hindered by data availability and the coarse model resolution. (1) To our knowledge the data collection of depth to groundwater by Fan. et al (2013) is unique. However, they do not represent steady-state values. Apart from depth to groundwater observations, hardly any relevant data is available at the global scale. Especially exchange between surface water and groundwater is difficult to measure even at the local scale. Therefore, we compared G3M results with the results from other large-scale models. Comparison to the results of catchment-scale groundwater flow models is planned for transient runs that will be possible after the integration into WaterGAP. (2) Scale differences make the comparison to point observations of depth to groundwater difficult. Multiple local observations within a 5' cell may strongly vary, maybe just due to land surface elevation variations within the approximately 80 km2 large cells (compare Fig. S1 and S8). Often, observations are biased towards alluvial aquifers in valleys. The calculated hydraulic head of the grid cell may represent the average groundwater level per grid cell correctly but can be still far off the local observations of depth to groundwater. As the current model only presents an uncalibrated natural steady-state, a comparison to observations only provides a first indicator where the model and the performance measurements needs to be improved as we move to a fully transient model."

**1.2 The authors state on line 24 page 1 that these models are useful in areas with little or no data, as they allow to generate robust information. How can anything robust be generated (and how do we know its true?) in the absence of data. The hydrogeological literature is full of examples where even in the data -rich regions different models produce different outcomes.**

Reply: In the revised version, we have deleted the statement about robust information, and explain now in more detail (on page 3 lines 24 ff) the purpose of our research effort, i.e. global-scale gradient-based groundwater flow modeling.

Changes to manuscript: Page 3, Line 24 ff now reads: "Our model development approach was to learn from existing large-scale regional models (Faunt, 2009; Vergnes et al., 2014, Maxwell et al., 2015; Dogrul et al., 2016) to gain insights into how the coarse spatial resolution, incomplete data, and conceptual model design affects model outcome. We want to find out whether we can use gradient-based groundwater modelling at the global scale, when later integrated into a global hydrological model, to improve estimation of flows between SW and GW (affecting both e.g. streamflow and groundwater recharge and thus water availability for humans and ecosystems) and capillary rise (affecting evapotranspiration)."

**1.3 We are presented with plots, numbers and graphs and some interpretation, but there is no credible discussion on the reliability of the result obtained. The only indication of model performance is that there is essentially no correlation between simulated and observed depth to groundwater. To me this means simply that the model cannot be used to make these types of predictions.**

Reply: Comparison between depth to GW derived from simulated steady-state hydraulic head and point-scale observations of (non-steady state) depth to GW is not straightforward at all, such that clear conclusions about the model performance are difficult. The model performance assessment is hindered by two factors: data availability and scale. (1) To our knowledge the data collection of depth to groundwater by Fan. 2013 is unique. We try to extend that picture with large scale regional models as base for our comparison. We do acknowledge that comparison to a model is not the same as a comparison to observations. Apart from depth to groundwater observations hardly any relevant data is available. Especially exchange between surface water and groundwater is inherently hard to measure even at the local scale and thus often a calibration parameter in small scale models. (2) Scale differences make the comparison to depth

to groundwater observations difficult. (1) Multiple local observations within a 5 arcmin cell can vary by a large range (2) and they may have been observed at location very different from the average groundwater characteristic (e.g. average hydraulic head) within a grid cell - often biased towards alluvial aquifers in valleys. The calculated hydraulic head of the grid cell may represent the average groundwater level per grid cell but can be still far of the local observation. Furthermore, we observe that the depth to groundwater is highly influenced by the location of the surface water bodies (swb) and perception of depth to groundwater changes if calculated heads are compared to swb elevation and not to the average cell elevation.

Changes to manuscript: To respond to this comment, we added the following paragraph to section 4 Discussion (second paragraph) (refer to comment #1.1) To show the conceptual difficulty of calculating "simulated" depth to GW from the simulated 5' grid cell hydraulic head and an effective or mean land surface elevation at this scale, we revised section 3.1 and added, as Fig. 3b, a map showing the difference between P30 (the 30th percentile of the 30" land surface elevations, the assumed elevation of the surface water body water table) and the computed hydraulic head.

**1.4 It is not useful to plot observed and simulated hydraulic heads over such large scales, even if its just for the sake of model comparison. It is true that other authors have also presented simulated vs observed hydraulic heads over such large scales, but this is simply misleading. Depth to groundwater is the variable that counts for calculating exchanges with surface, amongst many other processes. In this sense none of the available models on a global scale is ready yet. This must not necessarily be a problem, as long as the results are not oversold, as is unfortunately rather often the case.**

Reply: Hydraulic head is the main model output which is a good reason for showing it as such. And while the simulated heads might not match the observed very well in terms of absolute quantities, there are insights to be gained by looking at trends. Even local-scale models often do not match heads very well, but can be useful to understand

the system response (i.e., how/where do aquifer heads change with other changes in the system stresses (pumping increases, recharge decreases, stream flows change, etc.). Depth to groundwater table is only derived from the model output using some estimate of a representative land surface elevation (see response to the above comment and the ensuing revisions of the manuscript). Calculated depth to groundwater highly depends on how a DEM is used to account for inter cell variability. On the other hand, this is also true for the derived head observations. Plots of simulated head vs. observed head are heavily influenced by the DEM signal and deviations due to difference in depth to water table are obfuscated due to the plot scales. Furthermore, the interaction of surface water bodies and the groundwater is driven by the gradients between heads. We do agree, however, that simulation of capillary rise requires a good estimate of local-scale depth to GW. Currently the model outcomes are not suitable to perform such a calculation. We already stated in the original manuscript (p.14 line 19-20) that there is almost no correlation between depth to GW observations and simulated values. So we are transparent about this and think that we do not "oversell" our results.

Changes to manuscript: none

**1.5 The formulation of the equation 2 is for a confined aquifer. The authors justify this conceptually wrong choice on line 20, page 6: "Flow equations are for confined aquifer because it reduces convergence time. "This is a very poor argument, purely based on convenience. To what extent the model should capture the relevant physics should cannot be a question on how difficult it is to solve equations. The goal of this modelling approach is to advance the interaction between the surface and the subsurface across very large scales. Given that the direct interaction with the surface always happens with unconfined aquifers the fundamental basis of the approach is flawed on the most basic level. While for steady state simulations the term falls out of the equation it is still very concerning that that a model is developed with inadequate flow equations.**

Reply: The paper presents a conceptual model that differs in many aspects to traditional regional GW models due to the required coarse spatial resolution. Using the flow equation for unconfined conditions, which is typically done for the upper layer of groundwater models (unless confined by aquitards) is done to represent that in case of the same hydraulic gradient, less water can be transported if the hydraulic head and thus the saturated thickness drops. When looking at depth of GW in Fig. (old)3, one may think that in particular in mountainous terrain, the 100 m thick upper layer of the aquifer has fallen (almost) dry and does therefore in reality transfer no more groundwater. However, as shown in section 3.1, the high depth to GW is mainly related to the large land surface elevation differences within the 5' grid cell. , in almost all cells, the groundwater table is above the elevation of the water table in the surface water bodies (while land surface elevation per se is not part of the flow equation). Thus, given the high uncertainty of assumed hydraulic conductivity values and unknown actual aquifer depth, the assumption of fixed transmissivities seems to be appropriate for our global 5' model. Using the equation of unconfined conditions cannot be expected to improve the simulations significantly. Conceptually, at the applied coarse spatial resolution of the GW model, model layers should not be considered to be fixed to a land surface elevation. The model layers can be rather thought to be vertically (somewhat) aligned with the elevation of the surface water body table, and the flow equation rather governs the lateral and vertical fluxes over a thickness of 200 m.

Changes to manuscript: To clarify the difficult but important aspect of the relation between model layers and surface elevation in steep terrain at the spatial resolution of 5', we revised Figure 1 and added to section 2.1: "In addition, due to the coarse spatial scale and the possible large variations of land surface elevations within each grid cell, the upper model layers should not be considered to be aligned with an average land surface elevation. The model layers can be rather thought to be vertically aligned with the elevation of the surface water body table, as this prescribed elevation is, together with the sea level, the only elevation included in the groundwater flow equation (Eq. 2)." We added to the second paragraph of section 2.3: "We choose to simulate confined flow conditions in both layers even though the upper layer can be expected to decrease

in depth and thus in transmissivity (hydraulic conductivity times saturated depth). Every unconfined aquifer can have an equivalent confined representation assuming a correct saturated thickness (Sheets et al., 2015). However, given the large uncertainties regarding hydraulic conductivities (possibly an order of magnitude) and the lack of knowledge about aquifer thickness, it is appropriate to choose the computationally more efficient assumption of confined conditions."

**1.6 I did not understand why the authors develop a new model in the first place. They rightfully acknowledge that models such as MODFLOW exist, and these model could potentially do the job. Their argument is that MODFLOW models typically integrate geological data that is not available on a global scale. Therefore, a simplified model is developed. But this is a strange way of reasoning, as with MODFLOW one is not obliged to integrate all the geological complexity. It would have been perfectly possible to use MODFLOW for this project, with several significant advantages: for example, an unconfined aquifer (see below) could have been simulated. In this sense the novelty of the aspects concerning model development is questionable.**

Reply: The main reason for not using MODFLOW directly but just implementing the MODFLOW approach is an efficient coupling to the existing global hydrological model WGHM. The structure of MODFLOW does not allow an efficient in memory coupling that also account for the two different scales without too much computational overhead. The new model allows a more flexible extension of new components and adaptions to the conceptual nature of the model like an alternative capillary rise or dynamic recalculation of surface waterbody conductance. Additionally, the model framework is indeed capable of simulating unconfined conditions. Nevertheless, we believe the decision against it is a reasonable assumption - see response to #1.5.

Changes to manuscript: Page 8, Line 25-27 "The main motivation to develop a new model framework is the efficient in-memory coupling to the GHM and more flexible adaptation to the specific requirements of global-scale modelling."

**1.7 There are many other problems working on a global scale which are not even mentioned here but will even further undermine the credibility of the model. The three most important ones are: (1) Elevation is the wrong parameter for such a model. The data that should be used is not an ellipsoid- DEM but rather a geoid as the geoundulation is significant. (2) The density of sea-water is different, therefore there should be a density correction. (3) Steady-state conditions are inappropriate assumption that is not justified sufficiently well.**

Reply: (1) As far as I understand the SRTM-based DEM it is based on a reference ellipsoid (WGS84) and a reference geoid that should already account for geoundulation. We assume that on a 5 arcmin resolution differences in the gravitational field are negligible. Furthermore, other inputs present a much higher uncertainty. (2) As the model is not intended to be used for studying specifically groundwater-ocean interactions, and given the cell size of 9 km, we assume that the difference in density can be neglected at this scale as other parameterization introduce a higher level of uncertainty. (3) Presenting a steady-state model is one of the first steps to understand model behaviour before moving towards a fully transient and fully coupled model. This represents a well-established method in developing groundwater models - regional as well as large-scale models. A steady-state model (1) limits the degrees of freedom and thus model complexity as no time-variation needs to be taken into account and no storage changes need to be tracked. (2) A steady-state clearly uncovers dominant processes and trends that otherwise might have been obfuscated in a transient model due to the slow changing nature of groundwater. It is evident that not all processes can be observed in a steady-state and model behaviour will change as we move towards a fully transient model. It represents a first step in the model development process. Furthermore, (3) generated steady-state hydraulic heads can be used as initial state for a transient model spin-up phase in a fully coupled model. It is true however that surface water bodies do not have a steady-state and that aquifers are ever changing. This is why the presented steady-state represents a first step into the model development as we move towards fully transient and coupled model. We think that it is not meaningful

to move to a transient model directly with a completely new model without looking at the steady-state behaviour first.

Changes to manuscript: We added to the last paragraph of the introduction: "Steady-state simulations are a well-established first step in groundwater model development to understand the basic model behavior limiting model complexity and degrees of freedom, thus providing insights into dominant processes and uncovering possible model-inherent characteristics impossible to observe in a fully coupled transient model. A transient model might obfuscate model inherent trends due to the slow changing nature of groundwater processes. A fully coupled model furthermore adds complexity and uncertainty to the model outcome. In addition, the steady-state solution can be used as initial condition for future fully coupled transient runs. " (2) Page 5, Line 16

**1.8 Validation is done with other macro-scale models. This is a not an ideal strategy, as these large-scale models suffer from similar deficiencies (even though on less fundamental level). For a solid assessment of model performance, a detailed catchment scale hydrogeological model should be used for a benchmark comparison.**

Reply: Validation has been achieved by a comparison to global groundwater observations, assumed naturalized conditions in a well-studied area (Central Valley) and by an additional comparison to other large-scale models (Maxwell et al./Fan et al.). Goal of the model development was not the replication of regional groundwater characteristics - at this scale this is not a reasonable goal. Comparison is furthermore likely to be very challenging or impossible as a catchment might span only a couple of cells of the global model. The comparison to other large-scale models however enable a comparison based on similar input data (and input data deficiencies) uncovering how model decisions at this scale affect model outcome.

Changes to manuscript: See changes in response to comment #1.3 Page 17, Line 29 ff. "The presented comparison to other large-scale models is based on the assumption that same model deficiencies e.g. in available data and scale issues can uncover

differences in model decision. A comparison to catchment scale models is challenging as scales can differ by multiple magnitudes. As the model is further developed towards a transient model the presented comparison to simulations in data-rich regions need to be extended and temporal changes in interactions with surface water investigated."

**1.9 On line 28,page 7 the authors highlight that this is ok –". . .without losing important model behavior. " Transient and steady state is significantly different in both spatial and temporal dynamics.**

Reply: Reviewer refers to line 28 on page 3. We agree with the reviewer

Changes to manuscript: We revised the sentence. See changes in response to comment #1.2 and #1.7.

**1.10 The description of the conductance is confusing. In MODFLOW L is not the length of the river itself, but the length of the river within a grid cell. But this might just be an imprecise formulation.**

Reply: This is correct (See table 1). Manuscript has been changed accordingly.

Changes to manuscript: Page 6, Line 5

**1.11 Other aspects also require more justification and discussion. Why only 8 % of wetland surfaces? Where does this number come from? What are the numerical convergence criteria, as well as a wide range of additional model parameters?**

Reply: Manuscript is describing 80% of wetland area. Available maps of wetland areas show the maximum spatial extent of surface water bodies. As the maximum extent is seldom reached we reduce the extent for the steady-state model to 80% of the area shown in maps. In the fully transient model the wetland area will be adjusted in each time step as a function of wetland water storage.

It is not clear to us what the referee meant by "as well as a wide range of additional parameters". Parameters including convergence criteria are shown in Table 1.

—————————————————————————————————————————

Referee #2

**2.1 Is 5' an appropriate resolution at which to simulate groundwater flow? The analysis by Krakauer et al may be useful in determining the appropriate resolution.**

Reply: Kraukauer et al. (2014) suggests that a grid spacing smaller than $0.1°$ (6') for lateral groundwater processes is favourable for models running at a finer resolution than $1°$. Thus a 5' seems to be reasonable even though our results suggest that the scale properties of surface water elevation need to be investigated further and that information from subgrid scales might need to be accounted for to improve overall results.

Changes to manuscript: Page 4, Line 12,13

**2.2 The work is coupled to WaterGap at 0.5deg, this is a really large scale discrepancy. How do you think this might alter the model results?**

Reply: As groundwater recharge is mainly driven by climate inputs that are only available at coarse scales the presented steady-state model is not affected by the scale differences. Moving towards a fully coupled model scale differences between the two models play an important role especially for surface water body coupling. For example, it is not reasonable to calculate a river head change in the $0.5°$ model and apply that change equally to all 5' grid cells to recalculate the interaction between the surface water and the groundwater. The (future) presentation of a fully transient coupled model needs to discuss this more extensively.

Changes to manuscript: none

**2.3 The comparisons between this study and Fan et al and Maxwell et al are interesting. While pressure head is important, I think the bias from these scatterplots, basically**

water table depth, is more meaningful (as plotted in Fan et al / Maxwell et al too). The statistics will really be driven by topography which can occlude model performance and differences.

Reply: Please refer to our responses and changes to manuscript in response to comments #1.3 and #1.4.

**2.4 The diagram for how the model handles topographic breaks (Fig 1) is super confusing. Basically is water moved between cells even if there is a disconnect?**

Reply: Yes, this is due to the coarse lateral discretization where in a 5' grid cell with approx. 80 km$^2$ area, the elevation differences can be larger than 200 m (as described in the text). Lateral interaction between neighbouring cells is always calculated in the model even if large topographic breaks are present. In order to avoid confusion, we modified Fig. 1 and text in section 2.1 to clarify that the top layer in the model should not be thought of as being located right at the land surface elevation.

Changes to manuscript: To clarify the difficult but important aspect of the relation between model layers and surface elevation in steep terrain at the spatial resolution of 5', we revised Figure 1 and added to section 2.1: "In addition, due to the coarse spatial scale and the possible large variations of land surface elevations within each grid cell, the upper model layers should not be considered to aligned with an average land surface elevation. The model layers can be rather thought to be vertically aligned with the elevation of the surface water body table, as this prescribed elevation is, together with the sea level, the only elevation included in the groundwater flow equation (Eq. 2)."

**2.5 The assumption of confined conditions really seems hard to justify. This is effectively what de Graaf et al (2015, 2017) do with their two layer MODFLOW model with a stream package connection to PCRGLOB. There are so many assumptions present I think more careful discussion of how sensitivities in these assumptions (e.g. parameters in what amounts to the stream package used here) and feedback back to the WaterGap (which I think is just one-way at this point) would be really important.**

Reply: Regarding the assumption of confined conditions, we now explain the rationale for it (see changes to manuscript). A sensitivity analysis is beyond the scope of this paper. We are currently preparing a paper that presents an extensive sensitivity analysis of the steady-state G$^3$M presented here.

Changes to manuscript: Regarding the assumption of confined conditions, we added to the second paragraph of section 2.3: "We choose to simulate confined flow conditions in both layers even though the upper layer can be expected to decrease in depth and thus in transmissivity (hydraulic conductivity times saturated depth). Every unconfined aquifer can have an equivalent confined representation assuming a correct saturated thickness (Sheets et al., 2015). However, given the large uncertainties regarding hydraulic conductivities (possibly an order of magnitude) and the lack of knowledge about aquifer thickness, it is appropriate to choose the computationally more efficient assumption of confined conditions."

**2.6 From Figure 2 it appears that not all the features are implemented in this model, or perhaps not all the features are activated except for recharge. Since the abstract discusses capillary subsides for plant water use but this feature is not described (nor is it entirely clear how that would be implemented as a simple flux), I think a thorough re-working of this discussion and assumptions are needed. Unfortunately, this figure begs the question why is a methods paper in GMD incomplete and not presenting all the model features?**

Reply: The intention of Fig. 2 was to show how the gradient-based groundwater model G3M is planned to be coupled with/integrated into the global hydrological model Water-GAP. This information is necessary to understand the modelling choices made for the steady-state G3M presented in the manuscript, as a first step towards a fully coupled transient model. We think that a steady-state model is an important first step to justify a newly developed groundwater model and needs to be presented to the scientific community before moving further along to a fully coupled transient model. The steady-state model alone shows the difficulties of simulating groundwater flows at the coarse spatial

resolution required for global-scale modelling. The model feature capillary rise is not presented as it cannot work without coupling to the soil compartment of WaterGAP.

Changes to manuscript: We added the following sentence to the last paragraph of the 1 Introduction: "Capillary rise is not included in the presented steady-state simulation as simulation of capillary rise requires information of soil moisture that is only available when G$^3$M is fully integrated into WGHM."

**2.7 The maps of water table depth seem to have a tremendous shallow bias. It is hard to say because of low figure resolution, but perhaps most of Eastern N America, most of Australia, half of Europe and all of Tropical Africa are under water. I think additional discussion is needed here at least. Could this be due to the steady state assumptions? Confined conditions? The stream aquifer package? Resolution and slope? ET feedbacks?**

Reply: The visual impression is wrong, only the darkest blue means "under water", and this happens only in 2.1% of all cells. As we write (already in the first manuscript version) in section 3.1, "In 2.1 % of all cells, GW head is simulated to be above the land surface elevation, by more than 1 m in 0.3 % and by more than 100 m in 0.004 % of the cells.". Still, areas in Eastern N-America, Australia, Europe and tropical Africa present very shallow groundwater tables. This is mainly due to large wetland extends in these areas in connection with the steady-state approach. The extent of all wetlands (global already reduced by 20%) likely is overestimated as the data represents a maximum extend that is rarely reached in reality. Additionally, wetlands don't have a steady-state (or rather no surface water body) thus the interaction with the groundwater is likely overestimated and leads to the observed flooding.

Changes to manuscript: none

**2.8 It's hard to tell what the difference is here between the PRCGlob-MODFLOW model and this current model. More discussion is needed to clarify this distinction. I actually feel it's okay if there are many similar models out there (and both can be**

good models or bad models, it's not a competition), I would like more dissection of the differences in approach.

Reply: Already in the first version, we wrote in the abstract "Together with an appropriate choice for the effective elevation of the SW table within each grid cell, this enables a reasonable simulation of drainage from GW to SW such that, in contrast to the GW model of de Graaf et al. (2015, 2017), no additional drainage based on externally provided values for GW storage above the floodplain is required in G³M. Comparison of simulated hydraulic heads to observations around the world shows better agreement than de Graaf et al. (2015)." More explanation about this additional drainage required by PCR-GLOBWB but not G³M is given in the introduction: "The first global gradient-based GW model that was run for both steady-state (de Graaf et al., 2015) and transient conditions (de Graaf et al., 2017) was driven by GW recharge and SW data of the GHM PCR-GLOBWB (van Beek et al., 2011). However, there is not yet a two-way coupling of a GW flow model and a GHM. This may be due to the way de Graaf et al. (2015, 2017) modelled river-GW interaction. To achieve plausible hydraulic head results, they found it necessary to add an additional drainage flux to GW drainage driven by the hydraulic head difference between GW and river. This additional drainage, which accounts for about 50% of global GW drainage, is simulated as a function of GW storage above the floodplain, the values of which are computed externally by the linear GW reservoir model of PCR-GLOBWB (Equation 3 of de Graaf et al. (2017) – the model component that the gradient-based model was intended to replace. This prevents a full integration of the global GW flow model of de Graaf et al. (2017) into a GHM, as then, the linear GW reservoir model would be replaced by the GW flow model." The section in the discussion read "De Graaf et al. (2015) set their SW head (h_swb) to the land surface elevation of the 6' grid cells minus river depth at bankfull conditions plus water depth at average river discharge. Together with the missing interaction between lakes and wetlands and a different approach to river conductance, this might be a reason for the additional drainage above the floodplain that was necessary to avoid excessive flooding. On the other hand, this adaption allows the drainage of water even if the hydraulic

head is below the SW elevation that might have led to the global underestimation of hydraulic heads. Thus, the difference in model heads seems to be closely related to the sensitivity of SW body elevation."

Changes to manuscript: We modified the section in the discussion on the comparison to the gw model for PCR-GLOBWB by adding (see bold words): "De Graaf et al. (2015) set their SW head (h_swb) to the land surface elevation of the 6' grid cells minus river depth at bankfull conditions plus water depth at average river discharge. Together with the missing interaction between lakes and wetlands and a different approach to river conductance, this might be a reason for the additional drainage above the floodplain that was necessary to avoid excessive flooding, **and that is not needed in G3M**. On the other hand, this adaption allows the drainage of water even if the hydraulic head is below the SW elevation that might have led to the global underestimation of hydraulic heads. Thus, the difference in model heads seems to be closely related to the sensitivity of SW body elevation.

**2.9 The current model is also completely different from the Central Valley model. This strikes me as odd too. Is it water use? Boundary conditions?**

Reply: The presented Central Valley model plot show the initial state of the CVHM model and not computed model results. The initial condition represents the close to natural conditions in the early 1960s in the Central Valley with a very shallow groundwater table and large wetlands. Scale is most likely the main driver for the different results. Except for the scale differences G³M correctly computes shallow conditions close to the values assumed by CVHM with groundwater above the surface in the north and partially in the south of the valley. Furthermore, the depth to groundwater decrease towards the Sierra Nevada. Other differences are likely due to the steady-state and the connected assumptions on surface water bodies.

Changes to manuscript: Page 16, Line 14-17 "G³M correctly computes the shallow conditions with groundwater above the surface in the north, partially in the south of

the valley and decreasing towards the Sierra Nevada. The difference in the extend of flooded area could be due to large wetlands areas still present in the early 60s which are not represented in this extent in the data used by G$^3$M." Page 18, Line 3-6 "The comparison to the initial state (based on historical observations) of the CVHM model presents a first comparison within a data-rich region which provides also the future possibility of comparing transient model results and human impact on a regional scale. G$^3$M is able to reproduce the shallow groundwater table in the early 1960s. Differences are likely due to the steady-state approach and the connected assumptions on surface water bodies."

Please also note the supplement to this comment:
https://www.geosci-model-dev-discuss.net/gmd-2018-120/gmd-2018-120-AC1-supplement.pdf

[Figure]

**Fig. 1.** revised figure 1

[Figure]

**Fig. 2.** revised figure 3

**Supplement:**

[revised manuscript text omitted]

**Supplement**

[Figure]

**Figure S1** Difference [$m$] between mean elevation and $P_{30}$ elevation. Maximum value 1365 m**.**

[Figure]

5    **Figure S2** Land surface elevation [$m$] used in G³M: 5' average of 30"land surface elevation used in(Fan et al., 2013).

[Figure]

**Figure S3** Hydraulic conductivity [$ms^{-1}$] derived from Gleeson et al., 2014) by scaling it with the geometric mean to 5'. Very low values in the northern hemisphere are due to permafrost conditions.

[Figure]

**Figure S4** Mean annual groundwater recharge [$mm\ day^{-1}$] between 1901-2013, from WaterGAP 2.2c.

[Figure]

**Kommentiert [RR20]:** Replaced by old fig03, S5 was moved to new fig03

**Figure S5** Arithmetic mean [*m*] of the 30" land surface elevation per 5' grid cell and simulated equilibrium hydraulic head (simulated depth to GW). Maximum value 2070 m, minimum value -414 m (Extremes included in dark blue and dark red).

[Figure]

**Figure S6** Plots of depth to GW as calculated by G³M (a), difference in surface elevation to neighbouring cells (b), depth to GW as used by the CVHM as the natural state and starting condition (Faunt, 2009) (c), losing and gaining streams as calculated by G³M (d), difference in gradient of hydraulic head and surface elevation (e), losing and gaining lakes and wetlands as calculated by G³M for the Central Valley and the Great Basin.

[Figure]

**Figure S7** Ratio of hydraulic head gradient to 5' mean surface elevation gradient, only computed if the difference in direction of the gradient was smaller than 45°.

[Figure]

**Figure S8** Land surface elevation Difference of 30'' mean land surface elevation in 5' grid cell to mean elevation of neighbouring cells [**m**] to mean elevation of neighboring cells on 5' resolution.

[Figure]

**Figure S9** Comparison between three alternatives for setting $h_{swb}$. Left to right: Fit of simulated hydraulic heads observations if $h_{swb}$ is set (1) to the 30[th] percentile of the 30″ land surface elevations (standard model) , (2) alternatively to the average elevation of all "blue" cells of the 30″ water table results of Fan et al., 2013) or (3) is set to the average of the 30″ land surface elevations. A blue cell has a depth to GW of less than 0.25 m and indicates GW discharge to the surface. If no "blue" cell exists in the 5′ cell, the minimum elevation of the 30″ land surface elevation values within the cell was used.

[Figure]

**Fig. S10** Depth to groundwater [$m$] for SW body elevation at average of 30″ land surface elevations.

[Figure]

**Figure S11** Gaining and losing rivers (lower panel) and wetlands and lakes (upper panel) as flow into/out the GW $[mm\ day^{-1}]$ if $h_{swb}$ is set to average elevation of all "blue" cells of the 30″ water table results of Fan et al., 2013) (right). A blue cell is defined as a depth to groundwater of less than 0.25 m. If no "blue" cell exist in the 5′ cell, the minimum elevation of the 30″ land surface elevation values is used. Red denotes gaining SW bodies.

**Kommentiert [RR21]:** S13 moved to new fig03

---

## Author Comment (AC2) · 24 Jul 2018

**3.1 As explained in https://www.geoscientific-model-development.net/about/manuscript_types.html the preferred reference to code release is through the use of a DOI which is then cited in the paper. As the model version is already published on GitHub a DOI can easily be created using for instance Zenodo, see https://guides.github.com/ activities/citable-code/ for details.**

Reply: Citation of the code in the Open Source Journal was be replaced with a DOI pointing directly to the code.

Changes to manuscript: Page 19, Line 25 ff.

**3.2 As also stated in the guide lines E-mail contact to obtain access is not preferred and simulations and data should be made available as supplement, as DOI or as part of the release.**

Reply: Model output has been uploaded to a public repository and will be referenced in the paper via DOI.

Changes to manuscript: Page 19, Line 25 ff.

Please also note the supplement to this comment:
https://www.geosci-model-dev-discuss.net/gmd-2018-120/gmd-2018-120-AC2-supplement.pdf

**Supplement:**

[revised manuscript text omitted]

**Supplement**

[Figure]

**Figure S1** Difference [$m$] between mean elevation and $P_{30}$ elevation. Maximum value 1365 m**.**

[Figure]

5    **Figure S2** Land surface elevation [$m$] used in G³M: 5' average of 30"land surface elevation used in(Fan et al., 2013).

[Figure]

**Figure S3** Hydraulic conductivity [$ms^{-1}$] derived from Gleeson et al., 2014) by scaling it with the geometric mean to 5'. Very low values in the northern hemisphere are due to permafrost conditions.

[Figure]

**Figure S4** Mean annual groundwater recharge [$mm\ day^{-1}$] between 1901-2013, from WaterGAP 2.2c.

[Figure]

**Kommentiert [RR20]:** Replaced by old fig03, S5 was moved to new fig03

**Figure S5** Arithmetic mean [*m*] of the 30" land surface elevation per 5' grid cell and simulated equilibrium hydraulic head (simulated depth to GW). Maximum value 2070 m, minimum value -414 m (Extremes included in dark blue and dark red).

[Figure]

**Figure S6** Plots of depth to GW as calculated by G³M (a), difference in surface elevation to neighbouring cells (b), depth to GW as used by the CVHM as the natural state and starting condition (Faunt, 2009) (c), losing and gaining streams as calculated by G³M (d), difference in gradient of hydraulic head and surface elevation (e), losing and gaining lakes and wetlands as calculated by G³M for the Central Valley and the Great Basin.

[Figure]

**Figure S7** Ratio of hydraulic head gradient to 5' mean surface elevation gradient, only computed if the difference in direction of the gradient was smaller than 45°.

[Figure]

**Figure S8** Land surface elevation Difference of 30'' mean land surface elevation in 5' grid cell to mean elevation of neighbouring cells [**m**] to mean elevation of neighboring cells on 5' resolution.

[Figure]

**Figure S9** Comparison between three alternatives for setting $h_{swb}$. Left to right: Fit of simulated hydraulic heads observations if $h_{swb}$ is set (1) to the 30[th] percentile of the 30″ land surface elevations (standard model) , (2) alternatively to the average elevation of all "blue" cells of the 30″ water table results of Fan et al., 2013) or (3) is set to the average of the 30″ land surface elevations. A blue cell has a depth to GW of less than 0.25 m and indicates GW discharge to the surface. If no "blue" cell exists in the 5′ cell, the minimum elevation of the 30″ land surface elevation values within the cell was used.

[Figure]

**Fig. S10** Depth to groundwater [$m$] for SW body elevation at average of 30″ land surface elevations.

[Figure]

**Figure S11** Gaining and losing rivers (lower panel) and wetlands and lakes (upper panel) as flow into/out the GW $[mm\ day^{-1}]$ if $h_{swb}$ is set to average elevation of all "blue" cells of the 30″ water table results of Fan et al., 2013) (right). A blue cell is defined as a depth to groundwater of less than 0.25 m. If no "blue" cell exist in the 5′ cell, the minimum elevation of the 30″ land surface elevation values is used. Red denotes gaining SW bodies.

**Kommentiert [RR21]:** S13 moved to new fig03

---

## Author Comment (AC3) · 24 Jul 2018

My apologies for the additional reply and caused inconvenience.

Some of the citations broke after exporting the markup PDF of the manuscript. The supplement of this comment contains a fixed version of the markup. It is otherwise identical to the first submitted changes to the manuscript.

Please also note the supplement to this comment:
https://www.geosci-model-dev-discuss.net/gmd-2018-120/gmd-2018-120-AC3-supplement.pdf

---

## Referee Report (RR1)

**General comments:**

Reinecke et al. established a global gradient-based groundwater model (G3M) that would be integrated to the WaterGAP model. This advanced development should be greatly welcomed as currently there are still few global/large-scale groundwater models having ability to simulate groundwater heads. The geosciences field, particularly hydrology science community, would be benefited by this advanced modeling feature. This is my first time reading the manuscript (as I was not involved in reviewing the earlier version of manuscript) and I read it with great interest. The authors deserve huge credit in taking such huge modeling effort and producing a good manuscript with extensive analyses. I fully support the publication of this study and I have only few comments to the manuscript:

-   P1L20-21, P2L40-P3L4 and other lines related to '… additional drainage above flood plain …' in PCR-GLOBWB-MODFLOW: The 'additional drainage above flood plain' in the PCR-GLOBWB-MODFLOW works (e.g. de Graaf et al., 2015, 2017) was not intended for improving groundwater head simulation performance. Yet, such drainage was introduced to improve/discharge performance of the online coupled PCR-GLOBWB-MODFLOW. In fact, the introduction of the drainage above flood plain was based on the earlier works in Sutanudjaja et al. (2011, 2014). Initially, such drainage was not used in Sutanudjaja et al. (2011), which focused on offline coupling approach of PCR-GLOBWB-MODFLOW. In this offline and one-way coupling approach for modeling spatio-temporal groundwater head dynamics, Sutanudjaja et al. (2011) conceptualized that groundwater discharge/baseflow as merely a function based on groundwater and surface water head differences, via RIV and DRN packages of MODFLOW (McDonald and Harbaugh, 1988; Harbaugh et al., 2000; Harbaugh, 2005). However, as the online two-way coupling approach between PCR-GLOBWB and MODFLOW was established in Sutanudjaja et al. (2014), we realized that flows from RIV and DRN are too slow to satisfy fast/quick-response component of groundwater discharge originating from mountainous regions where many springs tapping groundwater are located higher up in the valleys and feeding tributaries and main rivers. To include such fast groundwater discharge (baseflow) component, it is assumed that groundwater above flood plain is drained based on a linear reservoir concept (for more detailed, see Sutanudjaja et al., 2014 and Sutanudjaja, 2012).

-   Related to the aforementioned comment, I am just wondering how the discharge/flow WaterGAP model will perform when an online two-way coupling/integration between G3M and WaterGAP is used. I know that this is still outside the scope of your current study/manuscript, which still focusses on steady-state (and offline approach) simulation. Yet, could you please speculate about this in the discussion part of your

manuscript? Do you expect that you have to calibrate your parameter values such as river conductances (e.g. c_swb and c_riv in Equations 5 and 6) in order to get good discharge performance? If calibration is required, could you please hypothesize about its consequence to your groundwater head simulation performance?

**Minor comments:**

P1L20-21: What do you mean by "… externally provided values for GW storage …"? Please rephrase. GW storages of PCR-GLOBWB-MODFLOW are always based on (internally) simulated groundwater heads.

P4L18-20: This sentence is not clear for me. Please consider to rephrase. Do you mean that you excluded large mountainous areas in your model simulation? Could you please be more specific about how you defined mountainous areas? It may be helpful for readers if you provide some examples of such mountainous area locations.

P7L20: "Globally constant but different values …" This is hard to read for me. Please consider to rephrase.

P9L32: I suggest providing global flux values in annual unit, e.g. m3 year-1 or km3 year-1 (as commonly done in other hydrological studies, such as Döll et al, 2014; Rodell et al., 2012).

Page 10, Figure 3: Please provide values in annual unit, e.g. m3 year-1 or km3 year-1.

P11L10-13: Could you please share your hypothesis or reason why the model cannot simulate losing rivers in Niger? Is it related to the forcing/input error?

P19L1: … world wide …

P19L24-25: Please give a brief explanation about the method of Morel-Seytoux et al. (2017).

With kind regards,

Edwin Sutanudjaja

**References:**

de Graaf, I. E. M., Sutanudjaja, E. H., van Beek, L. P. H., and Bierkens, M. F. P.: A high-resolution global-scale groundwater model, Hydrol. Earth Syst. Sci., 19, 823–837, https://doi.org/10.5194/hess-19-823-2015, 2015.

de Graaf, I. E., van Beek, L. P. H., Gleeson, T., Moosdorf, N., Schmitz, O., Sutanudjaja, E. H., and Bierkens, M. F. P.: A globalscale two-layer transient groundwater model: Development and application to groundwater depletion, Adv. Water Resour., 102, 53–67, 2017.

Döll, P., Muller Schmied, H., Schuh, Portmann, F. T., and Eicker, A.: Global-scale assessment of groundwater depletion and related groundwater abstractions: Combining hydrological modeling with information from well observations and GRACE satellites, Water Resour. Res., 50, 5698–5720, 2014.

McDonald, M., and A. Harbaugh, A Modular Three-Dimensional Finite-Difference Ground-Water Flow Model: Techniques of Water Resources Investigations, Book 6, U.S. Geol. Surv, Denver, Colorado, http://pubs.water.usgs.gov/twri6a1, 1988.

Harbaugh, A. W., Banta, E. R., Hill, M. C., and McDonald, M. G.: MODFLOW-2000, the U.S. Geological Survey modular groundwater model – User guide to modularization concepts and the Ground-Water Flow Process: U.S. Geological Survey Open-File Report 00-92, 121 pp., 2000.

Harbaugh, A. W.: MODFLOW-2005, the US Geological Survey modular ground-water model: the ground-water flow process, US Department of the Interior, US Geological Survey Reston, 2005.

Rodell, M., Beaudoing, H. K., L'Ecuyer, T. S., Olson, W. S., Famiglietti, J. S., Houser, P. R., Adler, R., Bosilovich, M. G., Clayson, C. A., Chambers, D., Clark, E., Fetzer, E. J., Gao, X., Gu, G., Hilburn, K., Huffman, G. J., Lettenmaier, D. P., Liu, W. T., Robertson, F. R., Schlosser, C. A., Sheffield, J., and Wood, E. F.: The Observed State of the Water Cycle in the Early TwentyFirst Century, J. Climate, 28, 8289–8318, 2012.

Sutanudjaja, E. H., van Beek, L. P. H., de Jong, S. M., van Geer, F. C., and Bierkens, M. F. P.: Large-scale groundwater modeling using global datasets: a test case for the Rhine-Meuse basin, Hydrol. Earth Syst. Sci., 15, 2913–2935, https://doi.org/10.5194/hess-15-2913-2011, 2011.

Sutanudjaja, E. H.: The use of soil moisture remote sensing products for large-scale groundwater modeling and assessment, PhD thesis, Utrecht Univ., Netherlands, 2012.

Sutanudjaja, E. H., van Beek, L. P. H., de Jong, S. M., van Geer, F. C., and Bierkens, M. F. P.: Calibrating a large-extent highresolution coupled groundwater-land surface model using soil moisture and discharge data, Water Resour. Res., 50, 687–705, 2014

---

## Author Response (AR5)

Dear Dr. Kurtz,

Thank you for your patience in the review process. Reviewer #5 provided additional input that was very helpful to include in the final version. Additionally, to his remarks we modified the following:

1) Figure 2 was extended by FigS4.5 (as it was often references in the text).
2) Table 1 and 2 changed their order.
3) Figure 10 contained an error in the data that was fixed and the overall representation improved.
4) Multiple minor spelling mistakes and minor rephrases to streamline the final version.

Our response to the Reviewer comments are in italic.

**Response to Reviewer #5**

**1)**

P1L20-21, P2L40-P3L4 and other lines related to '... additional drainage above flood plain ...' in PCR-GLOBWB-MODFLOW: The 'additional drainage above flood plain' in the PCR-GLOBWB-MODFLOW works (e.g. de Graaf et al., 2015, 2017) was not intended for improving groundwater head simulation performance. Yet, such drainage was introduced to improve/discharge performance of the online coupled PCR-GLOBWB-MODFLOW. In fact, the introduction of the drainage above flood plain was based on the earlier works in Sutanudjaja et al. (2011, 2014). Initially, such drainage was not used in Sutanudjaja et al. (2011), which focused on offline coupling approach of PCR-GLOBWB-MODFLOW. In this offline and one-way coupling approach for modeling spatio-temporal groundwater head dynamics, Sutanudjaja et al. (2011) conceptualized that groundwater discharge/baseflow as merely a function based on groundwater and surface water head differences, via RIV and DRN packages of MODFLOW (McDonald and Harbaugh, 1988; Harbaugh et al., 2000; Harbaugh, 2005). However, as the online two-way coupling approach between PCR-GLOBWB and MODFLOW was established in Sutanudjaja et al. (2014), we realized that flows from RIV and DRN are too slow to satisfy fast/quick-response component of groundwater discharge originating from mountainous regions where many springs tapping groundwater are located higher up in the valleys and feeding tributaries and main rivers. To include such fast groundwater discharge (baseflow) component, it is assumed that groundwater above flood plain is drained based on a linear reservoir concept (for more detailed, see Sutanudjaja et al., 2014 and Sutanudjaja, 2012).

*Thank you very much for the clarification and the interesting references! That helped us a lot to understand your model much better. Of course these statements are then wrong in our manuscript and have been removed from the abstract and modified in introduction, and discussion.*

*Introduction now reads (P.2 L.39 ff):*

"However, to achieve plausible discharge performance, they found it necessary to increase drainage from GW to rivers beyond the drainage driven by the hydraulic head difference between GW and river."

*Discussion now reads (P. 20 L.18 ff):*

"Together with the missing interaction between lakes and wetlands and a different approach to river conductance, this might be a reason for the additional drainage above the floodplain that was necessary to improve the discharge to rivers (Sutanudjaja et al., 2014)."

**2)**

Related to the aforementioned comment, I am just wondering how the discharge/flow

WaterGAP model will perform when an online two-way coupling/integration between

G3M and WaterGAP is used. I know that this is still outside the scope of your current

study/manuscript, which still focusses on steady-state (and offline approach)

simulation. Yet, could you please speculate about this in the discussion part of yourmanuscript? Do you expect that you have to calibrate your parameter values such as

river conductances (e.g. c_swb and c_riv in Equations 5 and 6) in order to get good

discharge performance? If calibration is required, could you please hypothesize about

its consequence to your groundwater head simulation performance?

*That is a very interesting question! We assume that the fully coupled model will help us especially in times of drought to improve streamflow simulation. It is very likely that the conductance will need to be calibrated. Yet we think it is very unclear how this could affect head performance. Future research needs to show how we can achieve a good fit to streamflow as well as to head observations.*

*Now reads (P. 18 L.21 ff):*

"We assume that the fully coupled model will lead to an improved WaterGAP performance during droughts with an increased drop in streamflow due to the now possible switch from gaining to losing conditions. Presumable a calibration of $c_{swb}$ and $c_{riv}$ is necessary to achieve a good discharge performance."

**Minor comments**

P1L20-21: What do you mean by "... externally provided values for GW storage ..."? Please

rephrase. GW storages of PCR-GLOBWB-MODFLOW are always based on (internally) simulated

groundwater heads.

*Thank you very much for the clarification! Of course these statements are then wrong in our
manuscript and have been removed from the abstract, introduction, and discussion.*

P4L18-20: This sentence is not clear for me. Please consider to rephrase. Do you mean that

you excluded large mountainous areas in your model simulation? Could you please be more

specific about how you defined mountainous areas? It may be helpful for readers if you

provide some examples of such mountainous area locations.

*No, they were not excluded. This paragraph speculates about the impacts if not the specific saturated
thickness approach (criticized by the former reviewers) but a head-based transmissivity was to be
used. For clarification the sentence has been revised and now reads (P. 4 L. 10-15):*

"Both approaches have proven to be insufficient to simulate head-based transmissivities (unconfined
contions) on the global scale. Large mountainous areas would be excluded if unconfined conditions
are assumed from the beginning of the solution step, as the head is often far below the deepest
model layer, resulting in a no-flow condition and imposing convergence issues to the matrix solver.
We choose to simulate both layers with a specific saturated thickness even though the upper layer
can be expected to decrease in water level and thus in transmissivity (hydraulic conductivity times
saturated depth)."

P7L20: "Globally constant but different values ..." This is hard to read for me. Please consider

to rephrase.

*Now reads (P. 7 L. 1):*

"Globally constant values are used for $B_{swb}$ for wetlands, local lakes and global lakes (Table 2)."

P9L32: I suggest providing global flux values in annual unit, e.g. m3 year-1 or km3 year-1 (as

commonly done in other hydrological studies, such as Döll et al, 2014; Rodell et al., 2012).

Page 10, Figure 3: Please provide values in annual unit, e.g. m3 year-1 or km3 year-1.

*Figure 3 is now in m³ year-1 and values in the text are reported accordingly. Figure 4, Figure S4.10 in mm/year.*

P11L10-13: Could you please share your hypothesis or reason why the model cannot simulate

losing rivers in Niger? Is it related to the forcing/input error?

*Now reads (P. 11 L. 20 ff):*

"On the other hand, no losing stretches are simulated along the Niger River and its wetlands and almost none in the North-eastern Brazil even though that losing conditions are known to occur there (Costa et al., 2013; FAO, 1997). This is also true when the minimum elevation for SW bodies is assumed (compare Fig. S4.10) leading to the conclusion that the misrepresentation might be linked to an inadequate representation of the local geology."

P19L1: … world wide …

*Fixed.*

P19L24-25: Please give a brief explanation about the method of Morel-Seytoux et al. (2017).

*Has been added (P. 20 L. 13-16):*

[revised manuscript text omitted]
. WaterGAP includes a one layer soil water storage compartment characterized by land cover specific rooting depth, maximum storage capacity and soil texture (Döll et al., 2014). The water content in the soil storage is increased by incoming precipitation and decreased by evapotranspiration and runoff generation (Döll et al., 2014). Capillary rise is not yet implemented in G³M, and SW heads are currently based on land surface elevation.

[Figure]

**Figure S1.1** Conceptual view of the coupling between WGHM and G³M. WGHM provides calculated GW recharge ($R_g$) (Döll and Fiedler, 2008) and if the human impact is considered, net abstraction from GW ($NA_g$) (Döll et al., 2012). G³M spreads this input equally to all 5' grid cells inside a 0.5° cell and calculates hydraulic head and interactions with SW bodies (swb) as well as capillary rise (cap. rise) at the 5' resolution. Grey arrows show information flow that is not yet implemented.

**2 Case study Central Valley**

To evaluate G³M further, its results were analysed for to a well-studied region, the Central Valley in California, USA. The Central Valley is one of the most productive agricultural regions of the world and heavily relies on GW pumpage to meet irrigation demands (Faunt et al., 2016). GW pumping in the valley increased rapidly in the 1960s (Faunt, 2009). Figure S2.1

5   shows simulated WTD for the Central Valley, the coast and the neighboring Sierra Nevada mountainside as well as parts of the Great Basin. The WTD table represents natural conditions without any pumping and is rather small. It roughly resembles the WTD assumed in the Central Valley Hydrological Model (CVHM) as initial condition, representing a natural state (Faunt, 2009) (Fig. S2.1b). G³M correctly computes the shallow conditions with groundwater above the surface in the north, partially in the south of the valley and decreasing towards the Sierra Nevada. The difference in the extend of flooded area could be due

10   to large wetlands areas still present in the early 60s which are not represented in this extent in the data used by G³M. Beyond the CVHM domain, WTD in mountainous regions is probably overestimated by G³M. The elevation of neighboring cells may differ up to a 1000 meter resulting in a large gradient (Fig. S4.5b and S4.5e).

[Figure]

**Figure S2.1** Plots of WTD [*m*] as calculated by G³M for the Central Valley and the Great Basin (a), and as used by CVHM as

15   the natural state and starting condition (Faunt, 2009) (b).

**3 Sensitivity Analysis**

Sensitivities are calculated using forward differences (Poeter et al., 2014).

$$\frac{\Delta y_i{'}}{\Delta b_j} = \frac{y_i{'}(b_j + \Delta b_j - y_i{'}(b_j)}{\Delta b j_j} \tag{S2}$$

where $y_i{'}$ is the simulated hydraulic head at position $i$ from ND number of cells and $b_j$ the perturbed parameter, here a multiplier for grid specific values shown in Table S1, in a vector of all parameter $b$ of length $j$. Based on these values the

20   composite scaled sensitivity is computed as

$$CSS_j = \sqrt{\sum_{i=1}^{ND} \frac{\Delta y_i{'}}{\Delta b_j} ND^{-1}} \tag{S3}$$

The result of the CSS is in units of meters. The higher the CSS, the more sensitive are the computed hydraulic heads to the parameter (Table S1).

**Table S1** Ranges of parameter multipliers used in the local sensitivity analysis and their resulting composite scaled sensitivity values. The multiplier for the wetlands applies to global and local wetlands.

| Parameter | $\Delta b$ | Composite Scaled Sensitivity [m] |
|---|---|---|
| $h_{swb}$ | 0.01 | 39132.1 |
| $K_{aq}$ | 0.01 | 76.8 |
| $R_g$ | 0.1 | 39.8 |
| $c_{Lakes}$ | 0.1 | 3.2 |
| $c_{Wetlands}$ | 0.1 | 0.014 |
| $c_{riv}$ | 0.1 | 0.013 |
| $c_{ocean}$ | 0.1 | 0.013 |

**4 Additional Figures**

[Figure]

**Figure S4.1** Difference [$m$] between 5' average of 30" land surface elevation and P$_{30}$ elevation. Maximum value 1365 m.

[Figure]

5  **Figure S4.2** Land surface elevation [$m$] used in G³M: 5' average of 30" land surface elevation used in Fan et al. (2013).

[Figure]

**Figure S4.3** Hydraulic conductivity [$ms^{-1}$] derived from Gleeson et al. (2014) by scaling it with the geometric mean to 5'. Very low values in the northern hemisphere are due to permafrost conditions.

[Figure]

**Figure S4.4** Mean annual groundwater recharge [$mm\ day^{-1}$] between 1901-2013, from WaterGAP 2.2c.

[Figure]

**Figure S4.5** Plots of WTD as calculated by G³M (a), difference in surface elevation to neighbouring cells (b), WTD as used by the CVHM as the natural state and starting condition (Faunt, 2009) (c), losing and gaining streams as calculated by G³M (d), difference in gradient of hydraulic head and surface elevation (e), losing and gaining lakes and wetlands as calculated by G³M for the Central Valley and the Great Basin.

[Figure]

Different direction  0  0.5  0.9  1.1  1.5  2  3

**Figure S4.6** Ratio of hydraulic head gradient to 5' mean surface elevation gradient, only computed if the difference in direction of the gradient was smaller than 45°.

[Figure]

0  50  100  200  400  1000  2000  3000  4040

**Figure S4.7** Land surface elevation difference between 30" mean land surface elevation in 5' grid cell and mean elevation of neighboring 5' cells [**m**].

[Figure]

**Figure S4.8** Comparison between three alternatives for setting $h_{swb}$. Left to right: Fit of simulated hydraulic heads observations if $h_{swb}$ is set (1) to the 30th percentile of the 30" land surface elevations (standard model) , (2) alternatively to the average elevation of all "blue" cells of the 30" water table results of Fan et al. (2013) or (3) is set to the average of the 30" land surface elevations. A blue cell has a WTD of less than 0.25 m and indicates GW discharge to the surface. If no "blue" cell exists in the 5' cell, the minimum elevation of the 30" land surface elevation values within the cell was used.

[Figure]

**Fig. S4.9** Depth to groundwater [$m$] for SW body elevation $h_{swb}$ at average of 30" land surface elevations.

[Figure]

**Figure S4.10** Gaining and losing rivers (lower panel) and wetlands and lakes (upper panel) as flow into/out the GW [$mm\ year^{-1}$] if $h_{swb}$ is set to average elevation of all "blue" cells of the 30" water table results of Fan et al. (2013) (right). A blue cell is defined as a depth to groundwater of less than 0.25 m. If no "blue" cell exist in the 5' cell, the minimum elevation of the 30" land surface elevation values is used. Red denotes gaining SW bodies.

Dear Dr. Kurtz,

Thank you for your remarks and the possibility to revise the manuscript again.
In the following, we are presenting additional changes to the manuscript to address the issues raised by
the reviewers, as indicated in your letter.
Our comments to the reviewer remarks (numbered) are provided in italics and are marked accordingly
in the manuscript.

**Reviewer #1**

**1.1**

[..]However, authors have not fully addressed reviewers' and editor's comments, and the manuscript still has
some deficiencies that need to be addressed. For example, response to comments 3 and 5 of the Editor is still
not convincing after revising the manuscript.

Comment 3 (of the Editor):
At the current stage, it is not clear what was learned from the modeling exercise and what are the advantages
and differences compared to already existing model set-ups and software packages. As advertised in the
manuscript, a main purpose of this study is to use a steady state groundwater model to gain first insights into
the credibility of the model set-up for future coupled transient simulations. However, this is not streamlined
at the moment, i.e. it is not clear what experience you gained from the steady state simulations to move
forward to transient coupled simulations. The problems outlined by the referees regarding model set-up and
model verification will prevail for transient coupled simulations and therefore need to be properly addressed
at the current stage. It needs to be pointed out more clearly, how this study serves as a basis for the
development of the desired coupled WaterGAP model.

Comment 5 (of the Editor):
The manuscript quite often refers to implementation that are planned in the future (i.e. coupling to WaterGAP,
transient simulations). While it is ok to outline these future plans in order to justify the proposed modelling
approach for the current global steady-state model, the frequency of references to future work is often
misleading (as also acknowledged by referee 2), because it is often not clear which kind of feature is really
implemented and which one is intended to be implemented. Therefore, I would ask you to limit references to
future work to the necessary minimum.

*The abstract and introduction were changed to focus the paper on the steady-state outcome and streamline
it with the presented results*

*Section 2.2 was exchanged with section 2.3 to streamline the paper further. Major parts of the new section
2.3 were moved to the supplement to preserve the information for the interested reader while focusing the
paper on the actual presented model.*
*The new section 2.3 now reads:*

```
In this initial effort, we intend to integrate G³M into WaterGAP 2, i.e. the 0.5°
version of WGHM (for details see S1) to keep computation time low enough for
performing sensitivity analyses and ensemble-based data assimilation and
calibration, instead of integrating it into WaterGAP 3 (Eisner, 2016), which has
the same spatial resolution as G³M. However, data from WaterGAP 3 were used to set
up G³M. Location and area of the 5' grid cells of G³M are the same as in the
landmask of WaterGAP 3. In addition, the percentage of the 5' grid cell area that
is covered by lakes (including reservoirs) and by wetlands, based on Lehner and
Döll (2004), is taken from WaterGAP 3, as well as the length and width of the main
river within each 5' grid cell as estimated by WaterGAP 3 (Table 1).
```

**1.2**

I concur with the editor's comment (3) that it is still not clear what is learned from this modeling exercise and comparison with existing methods are required to show the gradient based approach indeed is superior to the existing approaches.

*See 1.1. A comparison to existing approaches would mean a comparison to the linear storage approach of the global hydrological model as there are no other existing global approaches to this problem (apart from solving the computationally even more complex Richards equation). This is not possible at this stage because the linear storage describes only a transient groundwater storage not a steady-state equilibrium. This is something that is planned for the future transient integration into the GHM.*

**1.3**

Evaluation of the model performance against existing observations and model simulations such as CVHM and ParFlow models are qualitative. A more robust quantitative analysis using various statistical measures are required to show where the new approach works and where it does not.

*Section 3.5 was extended with a Root Mean Squared Errors for plots shown in Fig. 8 and are discussed accordingly (Lines 14-23 P. 14). Furthermore, a new Fig. 9 was added (page 16) that shows observed minus simulated head for different land surface elevation categories of the model including the IQR and Mean as Boxplot. Additionally, this figure shows the RMSE.*

**1.4**

Authors indicate lack of enough observations to validate their modeling approach. Perhaps, they can run the model for a higher resolution and evaluate the impact of coarse simulation on simulated hydrologic fluxes. This could provide valuable insights regarding model development and implementation for the coupling approach as well.

*We extended the study with a spatial scale sensitivity analysis of the model for the region of New Zealand and show 5 arcmin. resolution results in comparison with a model on 30 arcsec and compare results to local observations. It adds evidence to the hypothesis that a more elaborate estimation of the surface water body elevation can improve the 5 arcmin results.*
*See new section 3.6.*

**1.5**

The main contribution of this paper is not entirely clear. Authors have highlighted a number of improvements that the use of a gradient-based groundwater model could have on simulating global hydrological processes but many of those improvements have not been made in this version of the manuscript including the coupling with the global hydrologic model. As pointed out by the reviewers, I suggest authors to focus this paper on the sensitivity analysis of model parameterization and conceptual formulations. This can certainly help with the development of the transient model as well as the coupling method.

*See 1.1. Additionally, we added a local parameter sensitivity analysis of New Zealand (section 3.6). These results are then further reflected in the discussion (Line 23 Page 17, Line 9,18 Page 19).*
*As this paper is an initial model development paper and is already large as it is, we saw fit to submit a separate paper to HESS with a full global sensitivity analysis of the model.*

**1.6**

Could you please further explain how lateral connectivity between neighboring groundwater cells are calculated?

*The lateral connection of cells is based on the cell location which is determined by the landmask which is adapted from WaterGAP3. If the question relates to the lateral flow and hydraulic conductivity between cells, this is described in section 2.1. The hydraulic conductivity is calculated as the harmonic mean between neighboring cells. The calculated value is then used to calculate the hydraulic heads based on the in and out fluxes to the cell.*

*Further information can also be found in the MODFLOW documentation by Harbaugh (2005) also cited in this section.*

**Additional Remarks**

**A1.1**
Page 1 – L18: "as simulation of unsaturated flow and SW body elevation" is not clear. Please describe.

*The abstract was fully revised see #2.1*

**A1.2**
Page 1 - L31-34 : It is not clear.

*The abstract was fully revised see #2.1*

**A1.3**
Page 2- L5: replace recharge from soil with "precipitation"

*Precipitation is an input of the global hydrological model. Recharge from soil is the actual amount of water that infiltrates the groundwater taking into account e.g. evapotranspiration and surface runoff. Thus, recharge is only a certain percentage of precipitation.*

**A1.4**
Page 2- L27: replace "This flow direction" with "losing streams"

*Now reads (Line 14, Page 2):* `Losing streams typically occur in semi-arid and arid but seasonally also in humid regions. In addition, such linear reservoir models provide no information on the location of the GW table and assume that GW flow among grid cells is negligible.`

**A1.5**
Page 3- L16-17: Not clear. Please explain

*Revised and now reads (Line 44 ff., Page 2):*
`This additional drainage, which accounts for about 50% of global GW flow into SW, is simulated as a function of GW storage above the floodplain. The values needed to compute this additional artificial drainage are computed externally by the linear GW reservoir model of PCR-GLOBWB (Equation 3 of de Graaf et al. (2017)) – the model component that the gradient-based model was intended to replace in the first place. This prevents a full integration of the global GW flow model of de Graaf et al. (2017) into a GHM, as then, the linear GW reservoir model would be replaced by the GW flow model.`

**A1.6**
Figure 1. Add further details to the caption. Describe P30 of 30" DEM and why average of 30" DEM is compared with P30?

*Now reads (Description Fig 1.):* `The P`$_{30}$` is used in the presented steady-state model as SW elevation instead of an average or minimum per grid cell.`

**A1.7**
Page 4- Final paragraph needs revision to improve consistency with the rest of the text.

*It now reads (Line 14 ff., Page4):*
```
The simulation of aquifers that contain dry cells and/or cells that oscillate
between wet and dry states poses great challenges to the solving of Eq. (1)
(Niswonger et al. 2011). G³M-f (the framework code used to implement G³M)
implements the traditional wetting approach from Harbaugh (2005) as well as
the approach proposed by Niswonger et al. (2011) along with the proposed
damping scheme.
```

**A1.8**

Page 6 – L6: replace "loosing" with losing

*Now consistently spelled in manuscript.*

**A1.9**

Page 6 – L28: Replace equation 5 with 6

*Has been corrected.*

**A1.10**

Section 2.2. If the coupling is not done yet, why it is discussed in this paper? You could devote this section to assess the impacts of simplifications/improvements you made compared to the linear reservoir GW model. Or this section should come last (future work) if you plan to include the coupling approach in this paper.

*See 1.1. this section has been moved to the supplement and was shortened. For the linear GW model see the answer to 1.2.*

**A1.11**

Page 8 – L4: Could you please justify the choice of ocean conductance value.

*Now reads(Line 9, Page 7):* `[..]to` $10\ m^2\ day^{-1}$ `(Table 1), reflecting a global average conductance based on hydraulic conductivity and lateral surface area.`

**A1.12**

Section 3.1. Why P30 has been considered as the surface elevation of surface water bodies in the model?

*This is discussed in section 2.2 (former 2.3) (Line 24, Page 7):*
*"For the steady-state model, river elevation $h_{riv}$ is set in each grid cell to the same elevation as all other SW bodies, $h_{swb}$. We found that for both gaining and losing conditions, $Q_{swb}$ and thus computed hydraulic heads are highly sensitive to $h_{swb}$. The overall best agreement with the hydraulic head observations of Fan et al. (2013) was achieved if $h_{swb}$ (Eq. 4, 5 and 6) was set to the 30[th] percentile ($P_{30}$) of the 30″ land surface elevation values of Fan et al. (2013) per 5′ cell, e.g. the 30″ elevation that is exceeded by 70% of the thousand 30″ elevation values within one 5′ cell. To decrease convergence time we used $h_{eq}$ derived from the high-resolution steady-state hydraulic head distribution of Fan et al. (2013) as initial guess."*

**A1.13**

Section 3.2. L18 – "the amount of river water that recharges GW is only about a 40th of the drainage to GW," not clear

*Now reads( Line 33, page 9):*
```
According to G³M, the amount of river water that recharges GW is more than one
order of magnitude less than the drainage of GW, and the relative recharge to
GW from lakes and wetlands is even smaller (Fig. 3).
```

**A1.14**

Section 3.2 – L20 – Why outflow from SW body is not limited by water availability? River stage in equation 5 should control this.

*This is not possible in a steady-state model (will be possible with the transient model) as explained in section 2.2 (former 2.3):*
*"A further difficulty in an uncoupled model run is that the water table elevation of SW bodies does not react to the GW-SW exchange flows $Q_{swb}$ and that water supply from SW is not limited by availability. A losing river may in reality dry out due to loss to GW and therefore cease to lose any more water."*

**A1.15**

Section 3.2- Could you verify estimated global water budget against other existing global models?

*We would like to but the global budget of the only other existing model is not available. As we move to the transient model we will compare the transient budget of the gradient-based approach with the linear storage approach.*

**A1.16**

Section 3.3 – No attempt has been made to verify SW-GW exchange rate. This is an important contribution of this work and model validation is required. Perhaps, use ParFlow simulations over CONUS to check these fluxes at steady-state.

*Fluxes between groundwater and surface water are very complicated to validate. Even at the regional scale riverbed conductance is a calibration parameter as fluxes between groundwater and surface water can change on a very small scale and are challenging to measure. Measurements are furthermore only available for very small fractions of local streams which cannot be interpolated as comparison to 9km by 9km gridcells. ParFlow creates rivers naturally without predefining them in their model. Furthermore, as far as the authors are aware no exchange rate data is available for ParFlow at this point.*
*As we move to the transient implementation we certainly should consider to revisit this issue and compare the computed fluxes to existing regional models, if possible.*

**A.1.17**

Section 3.6. Comparison with the Central Valley hydrologic model is purely qualitative. It does not seem this qualitative comparison adds any value to assess G3M model performance.

*This section has been moved to the supplement (now section S2) for the interested reader and was replaced by the New Zealand study (new section 3.6).*

**Reviewer #2**

**2.1**

[..]Any comment or observation on the fully-coupled model therefore remains purely speculative, since the work has yet to be done. I recommend to change the focus to what is actually presented in the paper, which is the steady-state uncoupled application. It would greatly clarify the paper. Section 2.2 should therefore be deleted. The long-term objective could be mentioned in the discussion, as a perspective for future work. It would then be quite acceptable for the authors to justify the choice of using G3M instead of MODFLOW (as raised by reviewers), for tighter integration with WaterGap2.

*See response to 1.1. Section 2.2 has been shortened to a small paragraph and exchanged with section 2.3. The discussion now reflects the long-term perspective as suggested (Line 20 ff., Page 17):*

```
The objective of global gradient-based groundwater flow modeling with G³M is
to better simulate water exchange between SW and GW in the GHM WaterGAP, for
example for improved estimation of GW resources in dry regions of the globe
that are augmented by focused recharge from SW bodies. The presented steady-
state model is the first step in this direction:
```

**2.2**

I agree with the previous reviewers that the lessons learned from the steady-state application are not clear, compared to previous work. Since this paper does not present any results from a coupled model, arguments about coupling strategy as outcome are not valid. A future paper that actually presents results with the coupled model could address this coupling and assess challenges and issues. The other novel elements mentioned relate to scale challenges and equation solved but they are not unique to G3M. The other gradient-based GW flow models mentioned in the paper also face similar issues. The main outcomes of this steady-state application should be much more clearly highlighted and justified, in the context of previous studies.

*See also 1.1.,1.2 and 2.1. The introduction has been changed to reflect that together with the shortened section (new)2.3. Additionally, the discussion has been adapted to reflect that and now reads in Line 20 ff., Page 17:*

```
The objective of global gradient-based groundwater flow modeling with G³M is to
better simulate water exchange between SW and GW in the GHM WaterGAP, for example
for improved estimation of GW resources in dry regions of the globe that are
augmented by focused recharge from SW bodies. The presented steady-state model is
the first step in this direction: (1) It aligns with established GW model
development practices that helped (2) to understand basic model behavior e.g. the
sensitivity to SW body elevation, and the necessary improvement of its
parameterization, before moving to the more complex integrated transient model.
The reduced runtime of the steady-state model in comparison to a fully integrated
transient run (3) supported the investigation of parameter sensitivity and
sensitivity to spatial resolution. Additionally, (4) the presented steady-state
model can be used in future fully integrated transient runs as initial condition.
```

**2.3**

The previous reviewers raised concerns about using the fully-saturated groundwater flow equation, as opposed to an unconfined flow equation, for the upper layer. The justification given in response to these concerns are rather confusing and difficult to understand. I think that a short description of the way WCHM treats the soil "compartment" is needed and could help justify the approach. Right now, the reader has to guess what "soil" refers to.

*As the Editor and all reviewers asked to reduce the description of coupling to WaterGAP a more detailed description of the soil compartment of WaterGAP was added to S1 and is referred to for the interested read in Line 8 Page 4:*

```
WaterGAP includes a one layer soil water storage compartment characterized by
land cover specific rooting depth, maximum storage capacity and soil texture
(Döll et al. 2014). The water content in the soil storage is increased by
incoming precipitation and decreased by evapotranspiration and runoff
generation (Döll et al. 2014).
```

**2.4**

The paper is, in general, not clearly written. The style is often unnecessarily complicated, with very long sentences and repetition. The paper also does not focus on the essential and there is a lot of unnecessary detail. On the other hand, some important information is not presented clearly, such as a short description of how soils are treated in WCHM. Another example is Line 5 on page 8 that states : "It is assumed that there is exchange of water between GW and one river stretch in each 5' grid cell". If I understand correctly, there is a river for every top cell in the model. If it is the case, it is an important assumption and should be stated much more clearly and earlier, and not "buried" on page 8.

*The length of multiple sentences has been shortened wherever possible and repetitions decreased (see also following detailed comments).*

*For soil see 2.3 above.*

*The fact that each cell has a river is stated clearly in Table 2 and is compared to other models. The authors cannot agree with the observation that it is "buried" it appears clearly stated in section 2.2 which describes the steady-state assumptions.*

**2.5**

The terminology used is also very confusing for a reader with a hydrogeology background. For example, groundwater is used to refer to both the contained (the "groundwater" or subsurface compartment) and the content (groundwater that flows according to the governing equation). The paper also uses the term "drainage" to represent fluid exchange between the various compartments (subsurface, rivers, soil, wetlands, etc.). The use of drainage is extremely confusing and a better terminology would greatly clarify the paper. Also, hydrogeologists do not use hydraulic head for surface water or surface water table.

*The use of drainage has been clarified in multiple places (highlighted in the markup document with #2.5).*

*We disagree. The use of hydraulic head in the context of surface water hydrology is necessary in models that rely on hydraulic flow routing and are based on hydraulic head. Furthermore, it is correct, and even more precise and accurate, in this specific context as the flux between SW and GW cannot be computed solely based on a surface water table and the hydraulic head of the groundwater. To calculate an exchange, it is pivotal to know the pressure gradient between the surface water and the groundwater.*

**Detailed comments**

**A2.1**

The abstract provides a good example of the writing style. First, it is much too long. The abstract should be short, precise and to the point. It should not try to explain everything, such as the difference between reservoir and gradient-based GW models. It also contains sentences that are either unclear of very complicated. For example, this excerpt from lines 19-20 : "We identify challenges linked to the coarse resolution, which necessitates the deviation from established processes in regional groundwater modeling as simulation of unsaturated flow and SW body elevation". That sentence is both complicated and unclear, and does not inform the reader.

*The abstract has been adapted to reflect the streamlined paper and shortened accordingly:*
*It now reads:*

In global hydrological models, groundwater (GW) is typically represented by a bucket-like linear groundwater reservoir. Reservoir models, however, can (1) only simulate GW discharge to surface water (SW) bodies but not recharge from SW to GW, (2) provide no information on the location of the GW table and (3) assume that there is no GW flow among grid cells. This may lead, for example, to an underestimation of groundwater resources in semi-arid areas where GW is often replenished by SW or to an underestimation of evapotranspiration where the GW table is close to the land surface. To overcome these limitations, it is necessary to replace the reservoir model in global hydrological models with a hydraulic head gradient-based GW flow model.

We present G³M, a new global gradient-based GW model with a spatial resolution of 5', which is to be integrated into the 0.5° WaterGAP Global Hydrology Model (WGHM). The newly developed model framework enables in-memory coupling to WGHM while keeping overall runtime relatively low, which allows sensitivity analyses, calibration, and data assimilation. This paper presents the G³M concept and model design decisions that are specific to the large grid size required for a global scale model. In contrast to the GW model of de Graaf et al. (2015; 2017), no additional drainage based on externally provided values for GW storage above the floodplain is required in G³M, thus enabling full coupling to a GHM. Model results under steady-state naturalized conditions, i.e. neglecting GW abstractions, are shown. Simulated hydraulic heads show better agreement to observations around the world than the model output of de Graaf et al. (2015). Locations of simulated SW recharge to GW are found, as is expected, in dry and mountainous regions but the areal extent of SW recharge may be underestimated. Globally, GW discharge to rivers is by far the dominant flow component such that lateral GW flows only become a large fraction of total diffuse and focused recharge in case of losing rivers, some mountainous areas and some areas with very low GW recharge. Strong sensitivity of simulated hydraulic heads to the spatial resolution of the model and the related choice of the water table elevation of surface water bodies was found. We suggest to investigate how global-scale groundwater modeling at 5' spatial resolution can benefit from more highly resolved land surface elevation data.

**A2.2**

P2. The first paragraph in the introduction (lines 1-20) presents only generalities and should be deleted. The paper should focus on the model right from the start.

*The first paragraph provides a motivation for the presented research and justifies why modeling of global groundwater resources is worth the effort. Without it, it might be unclear to the reader why a GHM with a complex groundwater model is even necessary. However, the paragraph has been revised to be more succinct.*
*It now reads:*
Groundwater (GW) is the source of about 40% of all human water abstractions (Döll et al. 2014) and is also an essential source of water for freshwater biota in rivers, lakes and wetlands. GW strongly affects river flow regimes and supplies the majority of river water during ecologically and economically critical periods with little precipitation. GW storage and flow dynamics have been altered by human GW abstractions as well as climate change and will continue to change in the future (Taylor et al. 2012). Around the globe, GW abstractions have led to lowered water tables and, in some regions, even GW depletion (Döll et al. 2014; Scanlon et al. 2012; Wada et al. 2012; Konikow 2011). This has resulted in reduced base flows to rivers and wetlands (with negative impacts on water quality and freshwater ecosystems), land subsidence and increased pumping costs (Wada 2016; Döll et al. 2014; Gleeson et al. 2012; 2016). The strategic importance of GW for global water and food security will probably intensify under climate change as more frequent

and intense climate extremes increase variability of SW flows (Taylor et al. 2012). International efforts have been made to promote sustainable GW management and knowledge exchange among countries, e.g., UNESCO's program on International Shared Aquifer Resources Management (ISARM) (http://isarm.org) and the ongoing GW component of the Transboundary Waters Assessment Program (TWAP) (http://www.geftwap.org). To support prioritization for investment among transboundary aquifers as well as identification of strategies for sustainable GW management, information on current conditions and possible trends of the GW systems is required (UNESCO-IHP, IGRAC, WWAP (2012) 2012). In a globalized world, an improved understanding of GW systems and their interaction with SW and soil is needed not only at the local and regional but also at the global scale.

**A2.3**

P2. Line 34 : what are "macro-scale models"?

*Now reads (Line 21, Page 2) :* `large-scale models`.

**A2.4**

P2, Lines 37-38 : what is "the condition of SW"? Be more specific.

*Now reads (Line 24, Page 2):*
`However, they are in most cases not integrated within hydrological models that quantify GW recharge based on climate data and provide information on the condition of SW (e.g. streamflow and storage).`

**A2.5**

P2. Lines 38-39 : the excerpt "Miguez-Macho et al. (2007) linked a land surface model with a two-dimensional gradient-based GW model and computed, with a daily time step, gradient-based GW flow" is one example of unnecessary repetition that does not help the reader. There is no need to repeat that Miguez-Macho et al. used their gradient-based model to compute gradient-based GW flow. Another example of repetition is on page 3, lines 18-19 : "In this study, we present the Global Gradient-based Groundwater Model (G3M) that is to be integrated into the GHM WaterGAP 2" and just a bit further, line 29 is : "G3M is to replace this linear reservoir model in WGHM". Actually, that last repetition is even more confusing because GHM WaterGAP 2 and WGHM are not even the same model. I have noted several such repetitions that I will not list but that the authors should identify and eliminate.

*The sentence has been changed to (Line 25 ff., Page 2):*
`For North America, Fan et al. (2007) and Miguez-Macho et al. (2007) linked a land surface model with a two-dimensional gradient-based GW model and computed, with a daily time step, GW flow, water table elevation, GW–SW interaction, and capillary rise, using a spatial resolution of 1.25 km`

*The second mention of the integration was deleted.*

*To clarify WaterGAP and WGHM the following sentence was added (Line 5, Page 3):*
`In this study, we present the Global Gradient-based Groundwater Model (G³M) that is to be integrated into the GHM WaterGAP 2 (in the following we refer to WGHM (WaterGAP Global Hydrology Model), which is part of the GHM WaterGAP) to improve[..]`

**A2.6**

P3. Line 3 : "GW above the land surface". Check terminology for more clarity. GW above land surface is no longer GW. Should probably write instead something like : groundwater exfiltration.

*The calculated GW head above the land surface elevation does not represent a process like groundwater exfiltration. Such a process is not implemented in the discussed model. GW that is computed to be above*

*the assumed land surface elevation is simply discarded (! To clarify this statement refers to the Model of Fan and Miguez-Macho not the presented model.).*

**A2.7**

P3. Line 12 : The difference with the de Graaf et al paper is that they added "an additional drainage flus to GW drainage". That explanation is given several times in the paper but it is never clearly described and it is difficult to understand what de Graaf et al. did exactly.

*The introduction states that "This additional drainage, which accounts for about 50% of global GW drainage, is simulated as a function of GW storage above the floodplain, the values of which are computed externally by the linear GW reservoir model of PCR-GLOBWB (Equation 3 of de Graaf et al. (2017))[..]" An interested reader can find a detailed description in the original paper of de Graaf and in the according equation.*

**A2.8**

P3, Line 30 : The paper mentions the G3M model and now the G3M-f framework. First, what is the difference between G3M and G3M-f? Second, is it relevant to mention anything else that the model, G3M, since the application here is totally decoupled from a global hydrologic model?

*A framework is an established term that e.g. Wikipedia defines as: "In computer programming, a software framework is an abstraction in which software providing generic functionality can be selectively changed by additional user-written code, thus providing application-specific software. A software framework provides a standard way to build and deploy applications. A software framework is a universal, reusable software environment that provides particular functionality as part of a larger software platform to facilitate development of software applications, products and solutions."*
*This is explained in section 2.4. The goal of G³M-f was to develop a framework that can also be used to build regional models and be reused by the community to also build GW models for the integration into other models easily (because MODFLOW does not offer this capability) rather than only developing a code that can only be used with WaterGAP.*
*The reference in the introduction has been removed to avoid confusion.*

**A2.9**

P3, Lines 33-34 : The formulation "We want to find out whether we can use gradient-based groundwater modelling at the global scale, …, to improve estimation of flows between SW and GW … and capillary rise" is rather surprising. Isn't the working hypothesis that gradient-based models will improve simulations?

*That is the base hypothesis yes, but this is only true if the assumption above holds true and we aim at providing a demonstration of this approach, keeping in mind that this is the first example available. To clarify it now reads (Line 9 ff.,Page 3):*

```
        The objective of this paper is to learn from a steady-state model, a well-
established first step in groundwater model development, to (1) understand the
basic model behaviour by limiting model complexity and degrees of freedom, and
thus (2) providing insights into dominant processes and uncovering potential
model-inherent characteristics difficult to observe in a fully coupled transient
model. A transient model might obfuscate model inherent trends due to the slow
changing nature of groundwater processes e.g. trends towards large over/under-
estimation due to wrong parameterisation. A fully coupled model furthermore adds
complexity and uncertainty to the model outcome. The presented steady-state model
is furthermore used to (3) investigate parameter sensitivity and sensitivity to
spatial resolution. In addition, the steady-state solution can be used as (4)
initial condition for future fully coupled transient runs.
```

**A2.10**

P3, lines 36-40 : This is an example of a very complicated and unclear sentence : "Steady-state simulations are a well-established first step in groundwater model development to understand the basic model

behaviour limiting model complexity and degrees of freedom, thus providing insights into dominant processes and uncovering possible model-inherent characteristics impossible to observe in a fully coupled transient model." For example, what is "the basic model behaviour limiting model complexity and degrees of freedom"? What are the "model-inherent characteristics" that can't be observed in a fully coupled transient model?

*Steady-state models can be easily controlled and run quickly. We are not saying that they are superior to fully transient models, but they are a first step to understand the system and provide information that can be critical for developing an efficient transient model which is expected to require a much longer execution time.*

*Furthermore, in a transient GW model a run of 100 years might contain slight trends that lead to an ever increasing GW head in a specific region. Due to the slow nature of GW this might not be visible to the model developer. A steady-state model on the other hand would possibly show a clear overestimate/flooding due to e.g. the wrong parameterization.*
*A comma has been added to make clear that the model behavior is not limiting the complexity. Furthermore, the following sentence has been extended by an example to clarify what trends might not be visible in a transient model.*
*See A2.9*

**A2.11**
P3, lines 40-41: I don't know what is meant by "A transient model might obfuscate model inherent trends due to the slow changing nature of groundwater processes."

*See A2.10.*

**A2.12**
P3, line 41 : The following statement is quite bold and I am not sure that I agree : "A fully coupled model furthermore adds complexity and uncertainty to the model outcome". If it is the case, what do the authors want to develop a fully-coupled model if its outcome will be more uncertain?

*Because the main purpose of the model is not to build a standalone GW model but to replace the current GW storage model in WaterGAP. For that we need a fully coupled transient model, but the preliminary steady state model will inform us on the key physical processes and it is more controllable as mentioned above.*

**A2.13**
P4. Section 2 on Model description. I suggest to reorganize that section because it is not clear. I suggest to start by presenting the governing equations (equations 2 to 7) and then present the global-scale components. All simplifying hypothesis should be clearly stated. The exact input data originating from the global hydrologic model should also be clearly presented.

*We do not agree with this comment. The governing equations are a consequence of the conceptual nature of the global model and can be misinterpreted without the global-scale components. The data is clearly stated at the beginning of the section and further explained in Table 1.*
*Additionally, see 1.1.*

**A2.14**
P4, lines 10-11 : "G³M differs from traditional local and regional GW models". Is it really the case? I think that the main difference is the scale of application and the use of WCHM output as input.

*Yes, it differs in the exceptional spatial resolution of the grid and the necessary assumptions because of the lack of global data. This is further elaborated in the following sentences (Line 25 ff., Page 3):*
*"[..]These models are generally based on rather detailed information on hydrogeology (including aquifer geometry and properties such as hydraulic conductivity derived from pumping tests), topography, pumping wells, location and shape of SW bodies as well as on observations of hydraulic head in GW and SW. Local observations guide the developer in constructing the model such that local conditions and processes can be*

*properly represented. The lateral extent of individual grid cells of such GW flow models is generally smaller or similar to the depth of the aquifer(s) and the size of the SW bodies that interact with the GW. The global GW flow model G³M, however, covers all continents of the Earth except Greenland and Antarctica. At this scale, information listed above is poor or non-existing, and the lateral extent of grid cells needs to be relatively large due to computational (and data) constraints.*"

**A2.15**

P4, line 17 : "At this scale, information listed above is poor or non-existing". It should be reworded. The information contained for smaller-scale (as mentioned just above) is still available at the large (global) scale. You probably mean something else.

*No, it is unavailable because data available for a specific basin is unusable at the global scale without additional processing and assumptions. Furthermore, it might not even be possible to use this information after processing because it may not be possible to reasonably interpolate the data to an appropriate (global) grid-scale resolution.*
*We assume that the reader is able to understand that the information does not disappear but that it is very challenging to compile global datasets from local information.*

**A2.16**

P4, line 23 : Not clear what is meant by : Due to the lack of the distribution of hydrogeological properties.

*Properties like hydraulic conductivity change vertically and laterally and can be very heterogeneous for a given region. The spatial distribution of these properties is not available on the global scale (or even the regional scale).*
*We added "spatial distribution" (Line 37, Page 3).*

**A2.17**

P4, line 33 : I assume that "groundwater boxes" are actually "groundwater cells". If it's the case, then "cells" should be only consistently.

*Now reads cells.*

**A2.18**

P5. Figure 1 could be improved because it is not clear what is shown exactly. Also, what are "virtual layers"?

*It is unclear to us what needs improvement. The explanation of virtual layers was added to the figure 1:*
```
[..], and the conceptual virtual layers (virtual because at this stage only
confined conditions are computed)
```

**A2.19**

P5. Equation 2 is not the correct partial differential equation (PDE). The cell volumes (delta_x, delta_y, delta_z) only appear when the PDE is integrated over a 3D cell. Also, writing that the partial differential equation is "a function of hydraulic head gradients" is not rigorously correct. The PDE is derived from applying mass conservation to a representative elementary volume, where groundwater flow is described by its mass flux. The hydraulic head gradients appear because the mass flux is expressed with Darcy's Law.

*We clarified that the shown equation is approximated by using the finite element method. Furthermore, the sentence has been revised to be more correct.*
*It now reads(Line 6, Page 5):*
```
Three-dimensional groundwater flow is described by a partial differential
equation (approximated in the model implementation by using the finite
elements method) [equation] where [..]
```

P5, lines 20-21 : It is confusing to write that "Inflows in the groundwater are accounted for as…" because the equation is for both inflow and outflow.

*It now reads In- and outflows.*

P7, line 35 : in the exponential, what is m? what is the value of f?

*m is the unit meter as in the rest of the study. f is described in the sentence before as the e-folding factor*

P8, line 4 : the value of c_ocean is set to 100 m2/day. It appears to be several orders of magnitude greater than other conductances. Based on equation (4), I suspect that this large value is similar to specifying a first-type boundary condition for all cells located on the ocean boundaries. Is it the case?

*No it is set to 10 $m^2$/day. Relative to the other conductances (see Figure 6) it is relatively small.*

P9, lines 11-15. The paragraph is not clear.

*Now reads(Line 26, Page 8):*
```
      Similar to MODFLOW, G³M-f solves Eq. (1) in two nested loops using a Picard
iteration (Mehl, 2006): (1) the outer iteration checks the head and residual
convergence criterion (if the maximum head change is below a given value and/or
the residual norm is below a given value) and adjusts whether external flows have
changed into a different state e.g. from gaining to losing conditions and updates
the system of linear equations if flows are no longer head dependant. (2) The
inner loop primarily consists of the conjugate gradient solver, which runs for a
number of iterations defined by the user or until the residual convergence
criterion is reached (Table 1), solving the current system of linear equations.
```

P12, line 15 : "High conductance values are reached in the tropical zone due to a higher GW recharge". Is it really the case and not the opposite, i.e. because the conductance is large, groundwater recharge is larger?

*Yes, groundwater recharge is an input to the GW model and is used to compute the shown conductance for gaining rivers (see also section 2.1).*

P14, line 11 : what is meant by : comparison to local studies suggested a unit conversion error?

*The dataset of Fan provides the measurements of depth to groundwater in meter. By carefully comparing the values to local studies it seems that some of them haven't been properly converted and appear to be originally in feet. Because the original unit cannot be determined they were discarded.*

*Now reads (Line 15, Page 13):*
```
We selected only observations with known land surface elevation and removed
observations where a comparison to local studies suggested a unit conversion
error. This left total of 1,070,402 depth to GW observations.
```

P15, lines 18-19 : I don't understand the sentence : Plotting hydraulic head instead of depth to GW has the disadvantage that the goodness of fit is dominated by the topography as the observed heads are calculated based on the surface elevation of the model.

*Well observations always measure water table depth not a hydraulic head. For some measurements a surface elevation is available but for many it is not. To make a consistent comparison the surface elevation of the model is used. The values of the surface elevation are much bigger than the water table depth and thus "smooth" out small variations. Thus, the comparison of heads is driven more by topography as by the actual water table depth.*
*This is clarified with an additional sentence(Line 9, Page 15):*

```
Well observations always measure water table depth and only sometimes contain
complementary data specifying the elevation at which the measurements were
taken.
```

P17, lines 29-30. I don't understand the reference to "model decision". What does it mean?

*Now reads (Line 26, Page 18)*:

```
The presented comparison to other large-scale models is based on the
assumption that same model deficiencies e.g. in available data and scale
issues can uncover differences in model decisions e.g. used equations and
spatial resolution.
```

Table 2 : It is incorrect to write that the first 3 models solve the 3D Darcy equation. They solve a 3D mass conservation equation where the fluid flux is expressed with Darcy's law. It is also the case for ParFlow, which uses Darcy's Law to represent fluid fluxes in Richards' equation.

*Table 2 was changed to indicate that the flow representation is either 2D or 3D and either saturated or unsaturated.*

*It now reads:*

[revised manuscript text omitted]
. WaterGAP includes a one layer soil water storage compartment characterized by land cover specific rooting depth, maximum storage capacity and soil texture (Döll et al., 2014). The water content in the soil storage is increased by incoming precipitation and decreased by evapotranspiration and runoff generation (Döll et al., 2014). Capillary rise is not yet implemented in G³M, and SW heads are currently based on land surface elevation.

[Figure]

**Figure S1.1** Conceptual view of the coupling between WGHM and G³M. WGHM provides calculated GW recharge ($R_g$) (Döll and Fiedler, 2008) and if the human impact is considered, net abstraction from GW ($NA_g$) (Döll et al., 2012). G³M spreads this input equally to all 5' grid cells inside a 0.5° cell and calculates hydraulic head and interactions with SW bodies (swb) as well as capillary rise (cap. rise) at the 5' resolution. Grey arrows show information flow that is not yet implemented.

Commented [RR37]: #2.3

Commented [o38]: #A1.10

**2 Case study Central Valley**

To evaluate G³M further, its results were analyzed for to a well-studied region, the Central Valley in California, USA. The Central Valley is one of the most productive agricultural regions of the world and heavily relies on GW pumpage to meet irrigation demands (Faunt et al., 2016). GW pumping in the valley increased rapidly in the 1960s (Faunt, 2009). Figure S2.1

5    shows simulated depth to GW for the Central Valley, the coast and the neighboring Sierra Nevada mountainside as well as parts of the Great Basin. The depth to GW table represents natural conditions without any pumping and is rather small. It roughly resembles the depth to GW assumed in the Central Valley Hydrological Model (CVHM) as initial condition, representing a natural state (Faunt, 2009) (Fig. S2.1b). G³M correctly computes the shallow conditions with groundwater above the surface in the north, partially in the south of the valley and decreasing towards the Sierra Nevada. The difference in the

10   extent of the flooded area could be due to large wetlands areas still present in the early 60s which are not represented in this extent in the data used by G³M. Beyond the CVHM domain, depth to GW in mountainous regions is probably overestimated by G³M. The elevation of neighboring cells may differ up to a 1000 meter resulting in a large gradient (Fig. S4.6b and S4.6e).

[Figure]

**Figure S2.1** Plots of depth to GW [$m$] as calculated by G³M for the Central Valley and the Great Basin (a), and as used by

15   CVHM as the natural state and starting condition (Faunt, 2009) (b).

**3 Sensitivity Analysis**

Sensitivities are calculated using forward differences

$$\frac{\Delta y'_i}{\Delta b_j} = \frac{y'_i(b_j + \Delta b_j) - y'_i(b_j)}{\Delta b_j} \qquad (S2)$$

where $y'_i$ is the simulated hydraulic head at position $i$ from ND number of observations and $b_j$ the perturbed parameter (Table S1) in a vector of all parameters of length j. Based on these values the composite scaled sensitivity is computed as

$$CSS_j = \sqrt{\sum_{i=1}^{ND} \left(\frac{\Delta y'_i}{\Delta b_j}\frac{b_j}{\sigma_{y_i}}\right) ND^{-1}} \qquad (S3)$$

20   where $\sigma_{y_i}$ the standard deviation of an observation at this position. Because the observations are only available in a very small part of New Zealand "artificial observations" for each cell are assumed with 1 m hydraulic head each. Thus, $\sigma_{y_i}$ is 1 m. The result of CSS is dimensionless and can be compared directly between parameter multipliers (Table S1).

**Table S1** Ranges of parameter multipliers used in the local sensitivity analysis and their resulting composite scaled sensitivity values. The multiplier for the wetlands applies to global and local wetlands.

| Parameter | Multiplier Range | Composite Scaled Sensitivity |
|---|---|---|
| $h_{swb}$ | 1.01 | 39132.1 |
| $K_{aq}$ | 1.01 | 76.8 |
| $R_g$ | 1.1 | 39.8 |
| $c_{Lakes}$ | 1.1 | 3.2 |
| $c_{Wetlands}$ | 1.1 | 0.014 |
| $c_{riv}$ | 1.1 | 0.013 |
| $c_{ocean}$ | 1.1 | 0.013 |

**4 Additional Figures**

[Figure]

**Figure S4.1** Difference [$m$] between mean elevation and $P_{30}$ elevation. Maximum value 1365 m.

[Figure]

**Figure S4.2** Land-surface elevation [$m$] used in G³M: 5' average of 30" land surface elevation used in Fan et al. (2013).

[Figure]

**Figure S4.3** Hydraulic conductivity [$ms^{-1}$] derived from Gleeson et al. (2014) by scaling it with the geometric mean to 5'. Very low values in the northern hemisphere are due to permafrost conditions.

[Figure]

**Figure S4.4** Mean annual groundwater recharge [$mm\ day^{-1}$] between 1901-2013, from WaterGAP 2.2c.

[Figure]

**Figure S4.5** Arithmetic mean [*m*] of the 30" land surface elevation per 5' grid cell minus simulated equilibrium hydraulic head (simulated depth to GW). Maximum value 2070 m, minimum value -414 m (Extremes included in dark blue and dark red).

[Figure]

**Figure S4.6** Plots of depth to GW as calculated by G³M (a), difference in surface elevation to neighbouring cells (b), depth to GW as used by the CVHM as the natural state and starting condition (Faunt, 2009) (c), losing and gaining streams as calculated by G³M (d), difference in gradient of hydraulic head and surface elevation (e), losing and gaining lakes and wetlands as calculated by G³M for the Central Valley and the Great Basin.

[Figure]

**Figure S4.7** Ratio of hydraulic head gradient to 5' mean surface elevation gradient, only computed if the difference in direction of the gradient was smaller than 45°.

[Figure]

**Figure S4.8** Land-surface elevation Difference of 30" mean land surface elevation in 5' grid cell to mean elevation of neighboring cells [**m**] to mean elevation of neighboring cells on 5' resolution.

[Figure]

**Figure S4.9** Comparison between three alternatives for setting $h_{swb}$. Left to right: Fit of simulated hydraulic heads observations if $h_{swb}$ is set (1) to the 30th percentile of the 30″ land surface elevations (standard model) , (2) alternatively to the average elevation of all "blue" cells of the 30″ water table results of Fan et al. (2013) or (3) is set to the average of the 30″ land surface elevations. A blue cell has a depth to GW of less than 0.25 m and indicates GW discharge to the surface. If no "blue" cell exists in the 5′ cell, the minimum elevation of the 30″ land surface elevation values within the cell was used.

[Figure]

**Fig. S4.10** Depth to groundwater [$m$] for SW body elevation at average of 30″ land surface elevations.

[Figure]

**Figure S4.11** Gaining and losing rivers (lower panel) and wetlands and lakes (upper panel) as flow into/out the GW [$mm\ day^{-1}$] if $h_{swb}$ is set to average elevation of all "blue" cells of the 30" water table results of Fan et al. (2013) (right). A blue cell is defined as a depth to groundwater of less than 0.25 m. If no "blue" cell exist in the 5' cell, the minimum elevation of the 30" land surface elevation values is used. Red denotes gaining SW bodies.

Dear Dr. Kurtz,

Thank you for your remarks and the offer to provide additional guidance for revising the manuscript. In the following we are presenting additional changes to the manuscript to address the issues raised by the reviewers, as indicated in your letter.
Our comments to your remarks (numbered) are provided in italics.

(1) Both reviewers question the choice of a confined aquifer for the simulation of river-aquifer exchange. Evidence needs to be provided that the chosen assumption does not bias model results, e.g. by comparison with an unconfined representation of the upper model layer (as used in other available large-scale groundwater models), or by conducting a sensitivity study as suggested by referee 2.

*We acknowledge that the approach is counterintuitive and needs to be justified more extensively. Assuming that hydraulic conductivity/transmissivity is independent of the hydraulic head is an established practice in other large-scale GW models e.g. the Rhine Meuse basin model of Sutanudjaja et al. (2011), the Death Valley Regional Flow Model of Belcher (2004) and is applied in the global-scale groundwater flow model of de Graaf et al. (2015; 2017). Furthermore, this assumption, which should more appropriately be called a constant saturated thickness assumption, has been previously investigated by Faunt et al. (2011) and Sheets et al. (2015) showing reasonable performance for a large-scale regional GW model.*

*We extend section 2.1 (page 5, lines 1-14) with an additional paragraph:*

*"[..]The simulation of aquifers that contain dry cells and/or cells that oscillate between wet and dry states pose great challenges to the solver (Niswonger et al. 2011). G³M-f implements the traditional wetting approach of Harbaugh (2005) as well as the approach prosed in Niswonger et al. (2011) along with the proposed damping scheme. However, both approaches have proven to be insufficient to simulate hydraulic head-dependent transmisivities (i.e. unconfined conditions) on the global scale. Large mountainous areas would be excluded from the solution as the head is often far below the deepest model layer, resulting in a no-flow condition and causing convergence issues to the matrix solver. Therefore, we chose to simulate both layers with a constant saturated thickness as experiments with large-scale groundwater flow models have shown that the constant saturated thickness assumption is appropriate for large, complex groundwater models (Faunt et al. 2011; Sheets et al. 2015). Given the large uncertainties regarding hydraulic conductivities (possibly an order of magnitude), it is appropriate to choose the computationally more efficient assumption of specified saturated thickness. This approach was also chosen in recent large-scale groundwater flow studies, e.g for the Rhine Meuse basin (Sutanudjaja et al. 2011) (using one confined layer), the Death Valley Regional Flow Model (Belcher 2004), and the global groundwater model of de Graaf et al. (2017) (two layers and partially unconfined conditions are simulated by parameterization of the storage coefficients and not by a head-dependant transmissivity)."*

(2) The verification of the model shows considerable deficiencies. There is a significant mismatch between modelled and observed water table depths which is mainly justified by scale effects. Referee 1 rightfully points out that these discrepancies undermine the credibility of the modelling approach, also with respect to future model extensions and asks for a regional-scale validation of the model (which also avoids the problems of comparison of hydraulic heads over large ranges). The provided comparison with the Central Valley model only provides a rough qualitative comparison whereas a more detailed analysis would be needed for validation. If scale is a general obstacle for a proper model validation, then a logical consequence would be to increase model resolution to an appropriate level.

*Increasing the model resolution on the global scale is not possible not only due to insufficient data but in particular computational demand hindering the goal of having a model that can be further evaluated,*

*e.g. with a sensitivity analysis, and calibrated. We are currently investigating these scale impacts (see next remark).*
*To reflect the challenges that the proposed approach faces we have changed the paper title and abstract:*

*"Challenges in developing a global gradient-based groundwater model (G³M v1.0) for integration into a global hydrological model.*

*To quantify water flows between groundwater (GW) and surface water (SW) as well as the impact of capillary rise on evapotranspiration by global hydrological models (GHMs), it is necessary to replace the bucket-like linear GW reservoir model typical for hydrological models with a fully integrated gradient-based GW flow model. Linear reservoir models can only simulate GW discharge to SW bodies, provide no information on the location of the GW table, and assume that there is no GW flow among grid cells. A gradient-based GW model simulates not only GW storage but also hydraulic head, which together with information on SW table elevation enables the quantification of water flows from GW to SW and vice versa. In addition, hydraulic heads are the basis for calculating lateral GW flow among grid cells and capillary rise.*

*Global gradient-based modelling of GW is limited to certain resolutions due to available data and computational demands. We identify challenges linked to the coarse resolution, which necessitates the deviation from established processes in regional groundwater modelling as simulation of unsaturated flow and surface water body elevation. We present G³M, a global gradient-based GW model with a spatial resolution of 5' intended to replace the current linear GW reservoir in the 0.5° WaterGAP Global Hydrology Model (WGHM). The newly developed model framework enables in-memory coupling to WGHM while keeping overall runtime relatively low, allowing sensitivity analyses and data assimilation. This paper presents the G³M concept and specific model design decisions together with results under steady-state naturalized conditions (neglecting GW abstractions) that can later be used as initial conditions for the fully-coupled WGHM-G³M runs. Cell-specific conductances of river beds, which govern GW-SW interaction, were determined based on the 30" steady-state water table computed by Fan et al. (2013). Together with an appropriate choice for the effective elevation of the SW table within each grid cell, this enables a reasonable simulation of drainage from GW to SW such that, in contrast to the GW model of de Graaf et al. (2015; 2017), no additional drainage based on externally provided values for GW storage above the floodplain is required in G³M allowing a full coupling to a GHM. Comparison of simulated hydraulic heads to observations around the world shows better agreement than de Graaf et al. (2015). In addition, G³M output is compared to the output of two established macro-scale models for the Central Valley, California, and the continental United States, respectively. A first analysis of losing and gaining rivers and lakes/wetlands indicates that GW discharge to rivers is by far the dominant flow, draining diffuse GW recharge, such that lateral flows only become a large fraction of total diffuse and focused recharge in case of losing rivers and some areas with very low GW recharge. G³M does not represent losing rivers in some dry regions. This study clarifies the conceptual approach to gradient-based groundwater modelling that is necessary for global-scale modelling with a coarse spatial resolution. It presents the first steps towards replacing the linear GW reservoir model in a 0.5° GHM with a 5' gradient-based groundwater model, improving on recent efforts (fit to observations, model coupling), while explicating the major challenges related to the model resolution and the need for analysing the applicability of available higher-resolution land surface elevation data to overcome these challenges in the future."*

(3) At the current stage, it is not clear what was learned from the modelling exercise and what are the advantages and differences compared to already existing model set-ups and software packages. As advertised in the manuscript, a main purpose of this study is to use a steady state groundwater model to gain first insights into the credibility of the model set-up for future coupled transient simulations. However, this is not streamlined at the moment, i.e. it is not clear what experience you gained from the

steady state simulations to move forward to transient coupled simulations. The problems outlined by the referees regarding model set-up and model verification will prevail for transient coupled simulations and therefore need to be properly addressed at the current stage. It needs to be pointed out more clearly, how this study serves as a basis for the development of the desired coupled WaterGAP model.

*In our view the main contributions of that paper to the scientific community apart from the model itself are:*
- *The coupling strategy of a global groundwater model to a GHM on a different spatial resolution*
- *The **first time** that it is possible to **fully** couple a GHM to a GW model*
- *Provide insights into the scale challenges (SWB choices, observation comparison, choice in equations)*
- *Largely improve on recent efforts in comparison to observations*
- *We found that a conceptual approach to gradient-based groundwater flow modelling, with virtual layers, is necessary to deal with the large differences in land surface elevations.*
- *We determined that the large horizontal extent of grid cells required at the global scale is a major problem and developed a research plan involving simulation of the steady-state groundwater flow in New Zealand at three different spatial resolutions using globally available land surface elevation data; to identify how to parameterize the 5 min model using higher-resolution land surface elevation data.*

*To streamline the innovation of the paper and the gained insights we added the following paragraphs to the conclusion (together with the changed abstract and title in 2):*

*"[..] Furthermore, we provided insights into how the choice of surface water body elevation affects model outcome and plan to further investigate how we can use higher resolution topographic data to overcome these challenges by comparing simulation results of a 5', 30", and 3" GW model of New Zealand."*

(4) You should more clearly explain the advantages and differences of your modelling approach compared to already existing infrastructure as requested by both referees. For example, it is still not clear why G3M would be technically more suitable for in-memory coupling than already existing groundwater models like Modflow, i.e. why is it easier to access and exchange model variables with G3M. As pointed out by referee 1, the global-scale model set-up itself could be implemented with already existing groundwater modelling software. Following referee 2 also a more in-depth analysis on the differences between the proposed model and the model of de Graaf et al. is needed.

*We agree that we should point out the advantages of G³M-f over an existing modelling software. We added the following paragraph to 2.4 (page 8, lines 29-41):*

*"[..]G³M-f has the following advantages over using an established GW modelling software such as MODFLOW. G³M-f enables an improved coupling capability: (1) as it is intended to be used as a library-like module (unlike MODFLOW it provides a clear development interface to the programmer coupling a model to G³M-f, can be easily compiled as a library, and provides a clearly separated logic between computation and data read-in/write-out), (2) is written in the same language as the target GHM enabling a straight-forward in-memory access to arrays without the need to write data to disk, required by other global models (a very expensive operation even if that disk is a RAM-disk). Even though, it is possible to call FORTRAN functions from C++ it is complicated to pass file pointers properly, as the I/O implementation of both languages differ substantially and it is widely considered bad practice to handle I/O in two different languages at once. As MODFLOW was never designed to be coupled to other models, it is not possible to separate the I/O logic fully from the computational logic without substantial code changes that are hard to test. To this end, G³M-f provides a highly modularized framework that is written with extensibility as design goal while implementing all required groundwater mechanisms."*

*Concerning the differences to de Graaf et al., we think that we had already addressed the differences to a large extent.*

1) We chose to address this difference already in the abstract: "*Together with an appropriate choice for the effective elevation of the SW table within each grid cell, this enables a reasonable simulation of drainage from GW to SW such that, in contrast to the GW model of de Graaf et al. (2015; 2017), no additional drainage based on externally provided values for GW storage above the floodplain is required in G³M allowing a full coupling to a GHM. Comparison of simulated hydraulic heads to observations around the world shows better agreement than de Graaf et al. (2015).*"

2) In the introduction:
   "*The first global gradient-based GW model that was run for both steady-state (de Graaf et al. 2015) and transient conditions (de Graaf et al. 2017) was driven by GW recharge and SW data of the GHM PCR-GLOBWB (van Beek et al. 2011). However, there is not yet a two-way coupling of a GW flow model and a GHM. This may be due to the way de Graaf et al. (2015; 2017) modelled river-GW interaction. To achieve plausible hydraulic head results, they found it necessary to add an additional drainage flux to GW drainage driven by the hydraulic head difference between GW and river. This additional drainage, which accounts for about 50% of global GW drainage, is simulated as a function of GW storage above the floodplain, the values of which are computed externally by the linear GW reservoir model of PCR-GLOBWB (Equation 3 of de Graaf et al. (2017)) – the model component that the gradient-based model was intended to replace. This prevents a full integration of the global GW flow model of de Graaf et al. (2017) into a GHM, as then, the linear GW reservoir model would be replaced by the GW flow model.*"

3) Furthermore, we addressed the differences in the discussion:
   "*de Graaf et al. (2015) set their SW head ($h_{swb}$) to the land surface elevation of the 6' grid cells minus river depth at bankfull conditions plus water depth at average river discharge. Together with the missing interaction between lakes and wetlands and a different approach to river conductance, this might be a reason for the additional drainage above the floodplain that was necessary to avoid excessive flooding, and that is not needed in G³M. On the other hand, this adaption allows the drainage of water even if the hydraulic head is below the SW elevation that might have led to the global underestimation of hydraulic heads. Thus, the difference in model heads seems to be closely related to the sensitivity of SW body elevation.*"

*For clarity, to describe the differences of G³M to three previous global/continental-scale groundwater modelling approach referred to in our manuscript better we add an additional table in the discussion.*

**"Table 2 Comparison of global- and continental-scale groundwater models**

| Aspect | G³M | de Graaf et al. (2015; 2017) | Fan et al. (2013) | ParFlow |
|---|---|---|---|---|
| Extent | Global | Global | Global | Continental USA |
| Resolution | 5' | 6' | 30" | 1 km |
| Software | G³M-f | MODFLOW | Unnamed | ParFlow |
| Computational expense | Medium | Medium | High | Very high |
| Equation | 3d Darcy | 3d Darcy | 2d Darcy | 3d Richards |
| Time scale | Steady-state/(transient) | Steady-state/transient | Steady-state | Steady-state |
| Vertical layers | 2 | 2 | 1 | 5 |
| Full coupling possible | Yes | No (Conceptual issue) | No | Yes (already coupled) |
| In-memory coupling | Yes | No | N/A | Yes |

| | | | | |
|---|---|---|---|---|
| Constant saturated thickness | Yes | Yes | No | No |
| Impermeable bottom | No | No | No | Yes |
| Surface water body location | In every cell | In almost every cell | No surface water | Created during simulation |
| Surface water body elevation | $P_{30}$ of 30" DEM | Avg. of 30" DEM | N/A (outflow if depth to GW < 0.25 m) | N/A |
| Deviation from observations | Large | Very large | Medium | Medium |

"

(5) The manuscript quite often refers to implementation that are planned in the future (i.e. coupling to WaterGAP, transient simulations). While it is ok to outline these future plans in order to justify the proposed modelling approach for the current global steady-state model, the frequency of references to future work is often misleading (as also acknowledged by referee 2), because it is often not clear which kind of feature is really implemented and which one is intended to be implemented. Therefore, I would ask you to limit references to future work to the necessary minimum.

*We acknowledge that remark and think that providing a vision for a newly presented model is necessary to understand the presented decisions. We revised the presented document extensively and reduced references to future development to a minimum.*

Kind regards,
Robert Reinecke

[revised manuscript text omitted]

5    **Figure S2** Land surface elevation [*m*] used in G³M: 5' average of 30" land surface elevation used in Fan et al. (2013).

[Figure]

**Figure S3** Hydraulic conductivity [$ms^{-1}$] derived from Gleeson et al. (2014) by scaling it with the geometric mean to 5'. Very low values in the northern hemisphere are due to permafrost conditions.

[Figure]

**Figure S4** Mean annual groundwater recharge [$mm\ day^{-1}$] between 1901-2013, from WaterGAP 2.2c.

[Figure]

**Figure S5** Arithmetic mean [*m*] of the 30" land surface elevation per 5' grid cell and simulated equilibrium hydraulic head (simulated depth to GW). Maximum value 2070 m, minimum value -414 m (Extremes included in dark blue and dark red).

[Figure]

**Figure S6** Plots of depth to GW as calculated by G³M (a), difference in surface elevation to neighbouring cells (b), depth to GW as used by the CVHM as the natural state and starting condition (Faunt, 2009) (c), losing and gaining streams as calculated by G³M (d), difference in gradient of hydraulic head and surface elevation (e), losing and gaining lakes and wetlands as calculated by G³M for the Central Valley and the Great Basin.

[Figure]

**Figure S7** Ratio of hydraulic head gradient to 5' mean surface elevation gradient, only computed if the difference in direction of the gradient was smaller than 45°.

[Figure]

**Figure S8** Land surface elevation Difference of 30" mean land surface elevation in 5' grid cell to mean elevation of neighbouring cells [**m**] to mean elevation of neighboring cells on 5' resolution.

[Figure]

**Figure S9** Comparison between three alternatives for setting $h_{swb}$. Left to right: Fit of simulated hydraulic heads observations if $h_{swb}$ is set (1) to the 30[th] percentile of the 30″ land surface elevations (standard model) , (2) alternatively to the average elevation of all "blue" cells of the 30″ water table results of Fan et al. (2013) or (3) is set to the average of the 30″ land surface elevations. A blue cell has a depth to GW of less than 0.25 m and indicates GW discharge to the surface. If no "blue" cell exists in the 5' cell, the minimum elevation of the 30″ land surface elevation values within the cell was used.

[Figure]

**Fig. S10** Depth to groundwater [$m$] for SW body elevation at average of 30″ land surface elevations.

[Figure]

**Figure S11** Gaining and losing rivers (lower panel) and wetlands and lakes (upper panel) as flow into/out the GW [$mm\ day^{-1}$] if $h_{swb}$ is set to average elevation of all "blue" cells of the 30" water table results of Fan et al. (2013) (right). A blue cell is defined as a depth to groundwater of less than 0.25 m. If no "blue" cell exist in the 5' cell, the minimum elevation of the 30" land surface elevation values is used. Red denotes gaining SW bodies.

**Author comment**

We thank both reviewers for the thoughtful comments and questions. They helped us in particular to improve our explanation of the conceptual approach to gradient-based groundwater modeling that is necessary for global-scale groundwater modeling with a coarse spatial resolution, including the choice of simulating unconfined conditions and the conceptual difficulties in defining depth to groundwater table.

Our answers to the referees' comments are written in italics.

**Referee #1**

**#1.1**

A fundamental problem with these models is that they are difficult to verify, but this is not at all reflected in the discussion of the results.

*We have added a new paragraph to section 4 Discussion.*

**Changes to manuscript**

We added the following paragraph to section 4 Discussion (second paragraph)

"It is difficult to assess performance of the presented steady-state G³M results. Model performance assessment is hindered by data availability and the coarse model resolution. (1) To our knowledge the data collection of depth to groundwater by Fan. et al (2013) is unique. However, they do not represent steady-state values. Apart from depth to groundwater observations, hardly any relevant data is available at the global scale. Especially exchange between surface water and groundwater is difficult to measure even at the local scale. Therefore, we compared G³M results with the results from other large-scale models. Comparison to the results of catchment-scale groundwater flow models is planned for transient runs that will be possible after the integration into WaterGAP. (2) Scale differences make the comparison to point observations of depth to groundwater difficult. Multiple local observations within a 5' cell may strongly vary, maybe just due to land surface elevation variations within the approximately 80 km² large cells (compare Fig. S1 and S8). Often, observations are biased towards alluvial aquifers in valleys. The calculated hydraulic head of the grid cell may represent the average groundwater level per grid cell correctly but can be still far off the local observations of depth to groundwater. As the current model only presents an uncalibrated natural steady-state, a comparison to observations only provides a first indicator where the model and the performance measurements needs to be improved as we move to a fully transient model."

**#1.2**

The authors state on line 24 page 1 that these models are useful in areas with little or no data, as they allow to generate robust information. How can anything robust be generated (and how do we know its true?) in the absence of data. The hydrogeological literature is full of examples where even in the data -rich regions different models produce different outcomes.

*In the revised version, we have deleted the statement about robust information, and explain now in more detail (on page 2 lines 24 ff) the purpose of our research effort, i.e. global-scale gradient-based groundwater flow modelling.*

**Changes to manuscript**

Page 3, Line 24 ff now reads:

"Our model development approach was to learn from existing large-scale regional models (Faunt, 2009; Vergnes et al., 2014, Maxwell et al., 2015; Dogrul et al., 2016) to gain insights into how the coarse spatial resolution, incomplete data, and conceptual model design effects model outcome. We want to find out whether we can use gradient-based groundwater modelling at the global scale, when later integrated into a global hydrological model, to improve estimation of flows between SW and GW (affecting both e.g. streamflow and groundwater recharge and thus water availability for humans and ecosystems) and capillary rise (affecting evapotranspiration)."

**#1.3**

We are presented with plots, numbers and graphs and some interpretation, but there is no credible discussion on the reliability of the result obtained. The only indication of model performance is that there is essentially no correlation between simulated and observed depth to groundwater. To me this means simply that the model cannot be used to make these types of predictions.

*Comparison between depth to GW derived from simulated steady-state hydraulic head and point-scale observations of (non-steady state) depth to GW is not straightforward at all, such that clear conclusions about the model performance are difficult. The model performance assessment is hindered by two factors: data availability and scale.*

*(1) To our knowledge the data collection of depth to groundwater by Fan. 2013 is unique. We try to extend that picture with large scale regional models as base for our comparison. We do acknowledge that comparison to a model is not the same as a comparison to observations.*

*Apart from depth to groundwater observations hardly any relevant data is available. Especially exchange between surface water and groundwater is inherently hart to measure even at the local scale and thus often a calibration parameter in small scale models.*

*(2) Scale differences make the comparison to depth to groundwater observations difficult. (1) Multiple local observations within a 5 arcmin cell can vary by a large range (2) and they may have been observed at location very different from the average groundwater characteristic within a grid cell - often biased towards alluvial aquifers in valleys. The calculated hydraulic head of the grid cell may represent the average groundwater level per grid cell but can be still far of the local observation.*

*Furthermore, we observe that the depth to groundwater is highly influenced by the location of the surface water bodies (swb) and perception of depth to groundwater changes if calculated heads are compared to swb elevation and not to the average cell elevation.*

**Changes to manuscript**

To respond to this comment, we added the following paragraph to section 4 Discussion (second paragraph) (refer to comment #1.1)

To show the conceptual difficulty of calculating "simulated" depth to GW from the simulated 5' grid cell hydraulic head and an effective or mean land surface elevation at this scale, we revised section 3.1 and added, as Fig. 3b, a map showing the difference between P30 (the $30^{th}$ percentile of the 30'' land surface elevations, the assumed elevation of the surface water body water table) and the computed hydraulic head.

**#1.4**

It is not useful to plot observed and simulated hydraulic heads over such large scales, even if its just for the sake of model comparison. It is true that other authors have also presented simulated vs observed hydraulic heads over such large scales, but this is simply misleading. Depth to groundwater is the variable that counts for calculating exchanges with surface, amongst many other processes. In this sense none of the available models on a global scale is ready yet. This must not necessarily be a problem, as long as the results are not oversold, as is unfortunately rather often the case.

*Hydraulic head is the main model output which is a good reason for showing it as such. And while the simulated heads might not match the observed very well in terms of absolute quantities, there are insights to be gained by looking at trends. Even local-scale models often do not match heads very well, but can be useful to understand the system response (i.e., how/where do aquifer heads change with other changes in the system stresses (pumping increases, recharge decreases, stream flows change, etc.).*

*Depth to groundwater table is only derived from the model output using some estimate of a representative land surface elevation (see response to the above comment and the ensuing revisions of the manuscript). Calculated depth to groundwater highly depends on how a DEM is used to account for inter cell variability. On the other hand, this is also true for the derived head observations. Plots of simulated head vs. observed head are heavily influenced by the DEM signal and deviations due to difference in depth to water table are obfuscated due to the plot scales.*

*Furthermore, the interaction of surface water bodies and the groundwater is driven by the gradients between heads. We do agree, however, that simulation of capillary rise requires a good estimate of local-scale depth to GW. Currently the model outcomes are not suitable to perform such a calculation.*

*We already stated in the original manuscript (p.14 line 19-20) that there is almost no correlation between depth to GW observations and simulated values. So we are transparent about this and think that we do not "oversell" our results.*

**Changes to manuscript**

none

**#1.5**

The formulation of the equation 2 is for a confined aquifer. The authors justify this conceptually wrong choice on line 20, page 6: "Flow equations are for confined aquifer because it reduces convergence time. "This is a very poor argument, purely based on convenience. To what extent the model should capture the relevant physics should cannot be a question on how difficult it is to solve

equations. The goal of this modelling approach is to advance the interaction between the surface and the subsurface across very large scales. Given that the direct interaction with the surface always happens with unconfined aquifers the fundamental basis of the approach is flawed on the most

basic level. While for steady state simulations the term falls out of the equation it is still very concerning that that a model is developed with inadequate flow equations.

*The paper presents a conceptual model that differs in many aspects to traditional regional GW models due to the required coarse spatial resolution. Using the flow equation for unconfined conditions, which is typically done for the upper layer of groundwater models (unless confined by aquitards) is done to represent that in case of the same hydraulic gradient, less water can be transported if the hydraulic head and thus the saturated thickness drops. When looking at depth of GW in Fig. (old)3, one may think that in particular in mountainous terrain, the 100 m thick upper layer of the aquifer has fallen (almost) dry and does therefore in reality transfer no more groundwater. However, as shown in section 3.1, the high depth to GW is mainly related to the large land surface elevation differences within the 5' grid cell, in in almost all cells, the groundwater table is above the elevation of the water table in the surface water bodies (while land surface elevation per se is not part of the flow equation). Thus, given the high uncertainty of assumed hydraulic conductivity values and unknown actual aquifer depth, the assumption of fixed transmissivities seems to be appropriate for our global 5' model. Using the equation of unconfined conditions cannot be expected to improve the simulations significantly.*

*Conceptually, at the applied coarse spatial resolution of the GW model, model layers should not be considered to be fixed to a land surface elevation. The model layers can be rather thought to be vertically (somewhat) aligned with the elevation of the surface water body table, and the flow equation rather governs the lateral and vertical fluxes over a thickness of 200 m.*

**Changes to manuscript**

To clarify the difficult but important aspect of the relation between model layers and surface elevation in steep terrain at the spatial resolution of 5', we revised Figure 1 and added to section 2.1:

"In addition, due to the coarse spatial scale and the possible large variations of land surface elevations within each grid cell, the upper model layers should not be considered to aligned with an average land surface elevation. The model layers can be rather thought to be vertically aligned with the elevation of the surface water body table, as this prescribed elevation is, together with the sea level, the only elevation included in the groundwater flow equation (Eq. 2)."

We added to the second paragraph of section 2.3:

"We choose to simulate confined flow conditions in both layers even though the upper layer can be expected to decrease in depth and thus in transmissivity (hydraulic conductivity times saturated depth). Every unconfined aquifer can have an equivalent confined representation assuming a correct saturated thickness (Sheets et al., 2015). However, given the large uncertainties regarding hydraulic conductivities (possibly an order of magnitude) and the lack of knowledge about aquifer thickness, it is appropriate to choose the computationally more efficient assumption of confined conditions."

**#1.6**

I did not understand why the authors develop a new model in the first place. They rightfully acknowledge that models such as MODFLOW exist, and these model could potentially do the job. Their argument is that MODFLOW models typically integrate geological data that is not available on a global scale. Therefore, a simplified model is developed. But this is a strange way of reasoning, as with MODFLOW one is not obliged to integrate all the geological complexity. It would have been perfectly possible to use MODFLOW for this project , with several significant advantages: for example, an unconfined aquifer (see below) could have been simulated. In this sense the novelty of

the aspects concerning model development is questionable.

*The main reason for not using MODFLOW directly but just implementing the MODFLOW approach is an efficient coupling to the existing global hydrological model. The structure of MODFLOW does not allow an efficient in memory coupling that also account for the two different scales without too much computational overhead.*

*Furthermore, the new model allows a more flexible extension of new components and adaptions to the conceptual nature of the model like an alternative capillary rise or dynamic recalculation of surface waterbody conductance.*

**Changes to manuscript**

Page 8, Line 25-27

"The main motivation to develop a new model framework is the efficient in-memory coupling to the GHM and more flexible adaptation to the specific requirements of global-scale modelling."

**#1.7**

There are many other problems working on a global scale which are not even mentioned here but will even further undermine the credibility of the model. The three most important ones are: (1) Elevation is the wrong parameter for such a model. The data that should be used is not an ellipsoid-DEM but rather a geoid as the geoundulation is significant. (2) The density of sea-water is different, therefore there should be a density correction. (3) Steady-state conditions are inappropriate assumption that is not justified sufficiently well.

*(1) As far as I understand the SRTM-based DEM it is based on a reference ellipsoid (WGS84) and a reference geoid that should already account for geoundulation. We assume that on a 5 arcmin resolution differences in the gravitational field are negligible. Furthermore, other inputs present a much higher uncertainty.*

*(2) As the model is not intended to be used for studying specifically groundwater-ocean interactions, and given the cell size of 9 km, we assume that the difference in density can be neglected at this scale as other parameterization introduce a higher level of uncertainty.*

*(3) Presenting a steady-state model is one of the first steps to understand model behaviour before moving towards a fully transient and fully coupled model. This represents a well-established method in developing groundwater models - regional as well as large-scale models.*

*A steady-state model (1) limits the degrees of freedom and thus model complexity as no time-variation needs to be taken into account and no storage changes need to be tracked. (2) A steady-state uncovers dominant processes and trends clearly that otherwise might have been obfuscated in a transient model due to the slow changing nature of groundwater. It is evident that not all processes can be observed that way and model behaviour will change as we move towards a fully transient model. It represents a first step in the model development process. Furthermore, (3) generated steady-state hydraulic heads can be used as initial state for a transient model spin-up phase in a fully coupled model.*

*It is true however that surface water bodies do not have a steady-state and that aquifers are ever changing. This is why the presented steady-state represents a first step into the model development as we move towards fully transient and coupled model. We think that it is not meaningful to move to a transient model directly with a completely new model without looking at the steady-state behaviour first.*

**Changes to manuscript**

We added to the last paragraph of the introduction:

**"**Steady-state simulations are a well-established first step in groundwater model development to understand the basic model behavior limiting model complexity and degrees of freedom, thus providing insights into dominant processes and uncovering possible model-inherent characteristics impossible to observe in a fully coupled transient model. A transient model might obfuscate model inherent trends due to the slow changing nature of groundwater processes. In addition, the steady-state solution can be used as initial condition for future fully coupled transient runs.**"**

(2) Page 5, Line 16

**1.8**

Validation is done with other macro-scale models. This is a not an ideal strategy, as these large-scale models suffer from similar deficiencies (even though on less fundamental level). For a solid assessment of model performance a detailed, catchment scale hydrogeological model should be used for a benchmark comparison.

*Validation has been achieved by a comparison to global groundwater observations, assumed naturalized conditions in a well-studied area (Central Valley) and by an additional comparison to other large-scale models. Goal of the model development was not the replication of regional groundwater characteristics - at this scale this is not a reasonable goal. Comparison is furthermore likely to be very challenging or impossible as a catchment might span only a couple of cells of the global model. The comparison to other large-scale models however enable a comparison based on similar input data (and input data deficiencies) uncovering how model decisions at this scale affect model outcome.*

**Changes to manuscript**

See changes in response to comment #1.3

Page 17, Line 29 ff.

"The presented comparison to other large-scale models is based on the assumption that same model deficiencies e.g. in available data and scale issues can uncover differences in model decision. A comparison to catchment scale models is challenging as scales can differ by multiple magnitudes. As the model is further developed towards a transient model the presented comparison to simulations in data-rich regions need to be extended and temporal changes in interactions with surface water investigated."

**#1.9**

On line 28,page 7 the authors highlight that this is ok –" . . .without losing important model behavior. " Transient and steady state is significantly different in both spatial and temporal dynamics.

*Reviewer refers to line 28 on page 3. We agree with the reviewer*

**Changes to manuscript**

We revised the sentence. See changes in response to comment #1.2 and #1.7.

**#1.10**

The description of the conductance is confusing. In MODFLOW L is not the length of the river itself, but the length of the river within a grid cell. But this might just be an imprecise formulation.

*This is correct (See table 1). Manuscript has been changed accordingly.*

**Changes to manuscript**

Page 6, Line 5

**#1.11**

Other aspects also require more justification and discussion. Why only 8 % of wetland surfaces? Where does this number come from? What are the numerical convergence criteria, as well as a wide range of additional model parameters?

*Manuscript is describing 80% of wetland area. Available maps of wetland areas show the maximum spatial extent of surface water bodies. As the maximum extent is seldom reached we reduce the extent for the steady-state model to 80% of the area shown in maps. In the fully transient model the wetland area will be adjusted in each time step as a function of wetland water storage.*

*It is not clear to us what the referee meant by "as well as a wide range of additional parameters". Parameters including convergence criteria are shown in Table 1.*

**Changes to manuscript**

Page 8, Line 12

**Referee #2**

**#2.1**

Is 5' an appropriate resolution at which to simulate groundwater flow? The analysis by Krakauer et al may be useful in determining the appropriate resolution.

*Kraukauer et al. (2014) suggests that a grid spacing smaller than 0.1° (6') for lateral groundwater processes is favourable for models running at a finer resolution than 1°. Thus a 5' seems to be reasonable even though our results suggest that the scale properties of surface water elevation need to be investigated further and that information from subgrid scales might need to be accounted for to improve overall results.*

**Changes to manuscript**

Page 4, Line 12,13

**#2.2**

The work is coupled to WaterGap at 0.5deg, this is a really large scale discrepancy. How do you think this might alter the model results?

*As groundwater recharge is mainly driven by climate inputs that are only available at coarse scales the presented steady-state model is not affected by the scale differences. Moving towards a fully coupled model scale differences between the two models play an important role especially for surface water body coupling. For example, it is not reasonable to calculate a river head change in the 0.5° model and apply that change equally to all 5' grid cells to recalculate the interaction between the surface water and the groundwater. The (future) presentation of a fully transient coupled model needs to discuss this more extensively.*

**Changes to manuscript**

none

**#2.3**

The comparisons between this study and Fan et al and Maxwell et al are interesting. While pressure head is important, I think the bias from these scatterplots, basically water table depth, is more meaningful (as plotted in Fan et al / Maxwell et al too). The statistics will really be driven by topography which can occlude model performance and differences.

*Please refer to our responses and changes to manuscript in response to comments #1.3 and #1.4.*

The diagram for how the model handles topographic breaks (Fig 1) is super confusing. Basically is water moved between cells even if there is a disconnect?

*Yes, this is due to the coarse lateral discretization where in a 5' grid cell with approx. 80 km² area, the elevation differences can be larger than 200 m (as described in the text). Lateral interaction between neighbouring cells is always calculated in the model even if large topographic breaks are present. In order to avoid confusion, we modified Fig. 1 and text in section 2.1 to clarify that the top layer in the model should not be thought of as being located right at the land surface elevation.*

**Changes to manuscript**

To clarify the difficult but important aspect of the relation between model layers and surface elevation in steep terrain at the spatial resolution of 5', we revised Figure 1 and added to section 2.1:

"In addition, due to the coarse spatial scale and the possible large variations of land surface elevations within each grid cell, the upper model layers should not be considered to aligned with an average land surface elevation. The model layers can be rather thought to be vertically aligned with the elevation of the surface water body table, as this prescribed elevation is, together with the sea level, the only elevation included in the groundwater flow equation (Eq. 2)."

The assumption of confined conditions really seems hard to justify. This is effectively what de Graaf et al (2015, 2017) do with their two layer MODFLOW model with a stream package connection to PCRGLOB. There are so many assumptions present I think more careful discussion of how sensitivities in these assumptions (e.g. parameters in what amounts to the stream package used here) and feedback back to the WaterGap (which I think is just one-way at this point) would be really important.

*Regarding the assumption of confined conditions, we now explain the rationale for it (see changes to manuscript). A sensitivity analysis is beyond the scope of this paper. We are currently preparing a paper that presents an extensive sensitivity analysis of the steady-state G³M presented here.*

**Changes to manuscript**

Regarding the assumption of confined conditions, we added to the second paragraph of section 2.3:

"We choose to simulate confined flow conditions in both layers even though the upper layer can be expected to decrease in depth and thus in transmissivity (hydraulic conductivity times saturated depth). Every unconfined aquifer can have an equivalent confined representation assuming a correct saturated thickness (Sheets et al., 2015). However, given the large uncertainties regarding hydraulic conductivities (possibly an order of magnitude) and the lack of knowledge about aquifer thickness, it is appropriate to choose the computationally more efficient assumption of confined conditions."

From Figure 2 it appears that not all the features are implemented in this model, or perhaps not all the features are activated except for recharge. Since the abstract discusses capillary subsides for plant water use but this feature is not described (nor is it entirely clear how that would be implemented as a simple flux), I think a thorough re-working of this discussion and assumptions are needed. Unfortunately, this figure begs the question why is a methods paper in GMD incomplete and not presenting all the model features?

*The intention of Fig. 2 was to show how the gradient-based groundwater model G³M is planned to be coupled with/integrated into the global hydrological model WaterGAP. This information is necessary to understand the modelling choices made for the steady-state G³M presented in the manuscript, as a first step towards a fully coupled transient model. We think that a steady-state model is an important first step to justify a newly developed groundwater model and needs to be presented to the scientific community before moving further along to a fully coupled transient model. The steady-state model alone shows the difficulties of simulating groundwater flows at the coarse spatial resolution required for global-scale modelling. The model feature capillary rise is not presented as it cannot work without coupling to the soil compartment of WaterGAP.*

**Changes to manuscript**

We added the following sentence to the last paragraph of the 1 Introduction:

"Capillary rise is not included in the presented steady-state simulation as simulation of capillary rise requires information of soil moisture that is only available when G³M is fully integrated into WGHM."

**#2.7**

The maps of water table depth seem to have a tremendous shallow bias. It is hard to say because of low figure resolution, but perhaps most of Eastern N America, most of Australia, half of Europe and all of Tropical Africa are under water. I think additional discussion is needed here at least. Could this be due to the steady state assumptions? Confined conditions? The stream aquifer package? Resolution and slope? ET feedbacks?

*So your visual impression is wrong, only the darkest blue means "under water", and this happens only in 2.1% of all cells. As we write (already in the first manuscript version) in section 3.1, "In 2.1 % of all cells, GW head is simulated to be above the land surface elevation, by more than 1 m in 0.3 % and by more than 100 m in 0.004 % of the cells.". Still, areas in Eastern N-America, Australia, Europe and tropical Africa present very shallow groundwater tables. This is mainly due to large wetland extends in these areas in connection with the steady-state approach. The extent of all wetlands (global already reduced by 20%) likely is overestimated as the data represents a maximum extend that is rarely reached in reality. Additionally, wetlands don't have a steady-state (or rather no surface water body) thus the interaction with the groundwater is likely overestimated and leads to the observed flooding.*

**Changes to manuscript**

none

It's hard to tell what the difference is here between the PRCGlob-MODFLOW model and this current model. More discussion is needed to clarify this distinction. I actually feel it's okay if there are many similar models out there (and both can be good models or bad models, it's not a competition), I would like more dissection of the differences in approach.

*Already in the first version, we wrote in the abstract*

*"Together with an appropriate choice for the effective elevation of the SW table within each grid cell, this enables a reasonable simulation of drainage from GW to SW such that, in contrast to the GW model of de Graaf et al. (2015, 2017), no additional drainage based on externally provided values for GW storage above the floodplain is required in G³M. Comparison of simulated hydraulic heads to observations around the world shows better agreement than de Graaf et al. (2015)."*

*More explanation about this additional drainage required by PCR-GLOBWB but not G³M is given in the introduction:*

*"The first global gradient-based GW model that was run for both steady-state (de Graaf et al., 2015) and transient conditions (de Graaf et al., 2017) was driven by GW recharge and SW data of the GHM PCR-GLOBWB (van Beek et al., 2011). However, there is not yet a two-way coupling of a GW flow model and a GHM. This may be due to the way de Graaf et al. (2015, 2017) modelled river-GW interaction. To achieve plausible hydraulic head results, they found it necessary to add an additional drainage flux to GW drainage driven by the hydraulic head difference between GW and river. This additional drainage, which accounts for about 50% of global GW drainage, is simulated as a function of GW storage above the floodplain, the values of which are computed externally by the linear GW reservoir model of PCR-GLOBWB (Equation 3 of de Graaf et al. (2017) – the model component that the gradient-based model was intended to replace. This prevents a full integration of the global GW flow model of de Graaf et al. (2017) into a GHM, as then, the linear GW reservoir model would be replaced by the GW flow model."*

*The section in the discussion read*

*"De Graaf et al. (2015) set their SW head ($h_{swb}$) to the land surface elevation of the 6' grid cells minus river depth at bankfull conditions plus water depth at average river discharge. Together with the missing interaction between lakes and wetlands and a different approach to river conductance, this might be a reason for the additional drainage above the floodplain that was necessary to avoid excessive flooding. On the other hand, this adaption allows the drainage of water even if the hydraulic head is below the SW elevation that might have led to the global underestimation of hydraulic heads. Thus, the difference in model heads seems to be closely related to the sensitivity of SW body elevation."*

**Changes to manuscript**

We modified the section in the discussion on the comparison to the gw model for PCR-GLOBWB by adding (see bold words): "De Graaf et al. (2015) set their SW head ($h_{swb}$) to the land surface elevation of the 6' grid cells minus river depth at bankfull conditions plus water depth at average river discharge. Together with the missing interaction between lakes and wetlands and a different approach to river conductance, this might be a reason for the additional drainage above the floodplain that was necessary to avoid excessive flooding, **and that is not needed in G³M**. On the other hand, this adaption allows the drainage of water even if the hydraulic head is below the SW elevation that might have led to the global underestimation of hydraulic heads. Thus, the difference in model heads seems to be closely related to the sensitivity of SW body elevation.

The current model is also completely different from the Central Valley model. This strikes me as odd too. Is it water use? Boundary conditions?

*The presented Central Valley model plot show the initial state of the CVHM model and not computed model results. The initial condition represents the close to natural conditions in the early 1960s in the Central Valley with a very shallow groundwater table and large wetlands. Scale is most likely the main driver for the different results. Except for the scale differences G³M correctly computes shallow conditions close to the values assumed by CVHM with groundwater above the surface in the north and partially in the south of the valley. Furthermore, the depth to groundwater decrease towards the Sierra Nevada. Other differences are likely due to the steady-state and the connected assumptions on surface water bodies.*

**Changes to manuscript**

Page 16, Line 14-17

"G³M correctly computes the shallow conditions with groundwater above the surface in the north, partially in the south of the valley and decreasing towards the Sierra Nevada. The difference in the extend of flooded area could be due to large wetlands areas still present in the early 60s which are not represented in this extend in the data used by G³M."

Page 18, Line 3-6

"The comparison to the initial state (based on historical observations) of the CVHM model presents a first comparison within a data-rich region which provides also the future possibility of comparing transient model results and human impact on a regional scale. G³M is able to reproduce the shallow groundwater table in the early 1960s. Differences are likely due to the steady-state approach and the connected assumptions on surface water bodies."

**Short comment L. Gross**

**#3.1**

As explained in https://www.geoscientific-model-development.net/about/manuscript_types.html the preferred reference to code release is through the use of a DOI which is then cited in the paper. As the model version is already published on GitHub a DOI can easily be created using for instance Zenodo, see https://guides.github.com/ activities/citable-code/ for details.

*Citation of the code in the Open Source Journal was be replaced with a DOI pointing directly to the code.*

**Changes to manuscript**

Page 19, Line 25 ff.

As also stated in the guide lines E-mail contact to obtain access is not preferred and simulations and data should be made available as supplement, as DOI or as part of the release.

*Model output will be added to the supplementary material.*

**Changes to manuscript**

Page 19, Line 25 ff.

[revised manuscript text omitted]

**Supplement**

[Figure]

**Figure S1** Difference [$m$] between mean elevation and $P_{30}$ elevation. Maximum value 1365 m.

[Figure]

5    **Figure S2** Land surface elevation [$m$] used in G³M: 5' average of 30"land surface elevation used in Fan et al. (2013).

[Figure]

**Figure S3** Hydraulic conductivity $[ms^{-1}]$ derived from Gleeson et al. (2014) by scaling it with the geometric mean to 5'. Very low values in the northern hemisphere are due to permafrost conditions.

[Figure]

**Figure S4** Mean annual groundwater recharge $[mm\ day^{-1}]$ between 1901-2013, from WaterGAP 2.2c.

[Figure]

[Figure]

**Kommentiert [RR20]:** Replaced by old fig03, S5 was moved to new fig03

**Figure S5** Arithmetic mean [*m*] of the 30" land surface elevation per 5' grid cell and simulated equilibrium hydraulic head (simulated depth to GW). Maximum value 2070 m, minimum value -414 m (Extremes included in dark blue and dark red).

[Figure]

**Figure S6** Plots of depth to GW as calculated by G³M (a), difference in surface elevation to neighbouring cells (b), depth to GW as used by the CVHM as the natural state and starting condition (Faunt, 2009) (c), losing and gaining streams as calculated by G³M (d), difference in gradient of hydraulic head and surface elevation (e), losing and gaining lakes and wetlands as calculated by G³M for the Central Valley and the Great Basin.

[Figure]

**Figure S7** Ratio of hydraulic head gradient to 5' mean surface elevation gradient, only computed if the difference in direction of the gradient was smaller than 45°.

[Figure]

**Figure S8** Land surface elevation Difference of 30'' mean land surface elevation in 5' grid cell to mean elevation of neighbouring cells [**m**] to mean elevation of neighboring cells on 5' resolution.

[Figure]

**Figure S9** Comparison between three alternatives for setting $h_{swb}$. Left to right: Fit of simulated hydraulic heads observations if $h_{swb}$ is set (1) to the 30[th] percentile of the 30″ land surface elevations (standard model) , (2) alternatively to the average elevation of all "blue" cells of the 30″ water table results of Fan et al. (2013) or (3) is set to the average of the 30″ land surface elevations. A blue cell has a depth to GW of less than 0.25 m and indicates GW discharge to the surface. If no "blue" cell exists in the 5' cell, the minimum elevation of the 30″ land surface elevation values within the cell was used.

[Figure]

**Fig. S10** Depth to groundwater [$m$] for SW body elevation at average of 30″ land surface elevations.

[Figure]

**Figure S11** Gaining and losing rivers (lower panel) and wetlands and lakes (upper panel) as flow into/out the GW [$mm\ day^{-1}$] if $h_{swb}$ is set to average elevation of all "blue" cells of the 30" water table results of Fan et al. (2013) (right). A blue cell is defined as a depth to groundwater of less than 0.25 m. If no "blue" cell exist in the 5' cell, the minimum elevation of the 30" land surface elevation values is used. Red denotes gaining SW bodies.

**Kommentiert [RR21]:** S13 moved to new fig03